

# Isotopomer labeling and oxygen dependence of hybrid nitrous oxide production

Colette L. Kelly,[1,2*] Nicole M. Travis,[1] Pascale Anabelle Baya,[1] Claudia Frey,[3] Xin Sun,[4] Bess B. Ward,[5] and Karen L. Casciotti[1]

[1]Department of Earth System Science, Stanford University, Stanford, CA 94305, U.S.A.
[2]Department of Marine Chemistry and Geochemistry, Woods Hole Oceanographic Institution, Woods Hole, MA 02543, U.S.A.
[3]Department of Environmental Science, University of Basel, Basel, Switzerland
[4]Department of Global Ecology, Carnegie Institution for Science, Stanford, CA 94305, U.S.A.
[5]Department of Geosciences, Princeton University, Princeton, NJ 08544, U.S.A.

*Correspondence to:* Colette L. Kelly (email: colette.kelly@whoi.edu).

**Keywords:** Ammonia-oxidizing archaea, nitrous oxide, isotopomers, isotopocules, oxygen deficient zones, ammonia oxidation, hybrid nitrous oxide production, nitrous oxide production, Eastern Tropical North Pacific, nitrous oxide production from nitrate

**Abstract**

Nitrous oxide ($N_2O$) is a potent greenhouse gas and ozone depletion agent, with a significant natural source from marine oxygen deficient zones (ODZs). Open questions remain, however, about the microbial processes responsible for this $N_2O$ production, especially hybrid $N_2O$ production when ammonia-oxidizing archaea are present. Using [15]N-labeled tracer incubations, we measured the rates of $N_2O$ production from ammonium ($NH_4^+$), nitrite ($NO_2^-$), and nitrate ($NO_3^-$) in the

Eastern Tropical North Pacific ODZ, as well as the isotopic labeling of the central ($\alpha$) and terminal ($\beta$) nitrogen atoms of the $N_2O$ molecule. We observed production of both doubly- and singly labeled $N_2O$ from each tracer, with the highest rates of labeled $N_2O$ production at the same depths as the near-surface $N_2O$ concentration maximum. At most stations and depths, the production of ${}^{45}N_2O^\alpha$ and ${}^{45}N_2O^\beta$ were statistically indistinguishable, but at a few depths, there were significant differences in the labelling of the two nitrogen atoms in the $N_2o$ molecule. Implementing the rates of labeled $N_2O$ production in a

forward-running model, we found that $N_2O$ production from $NO_3^-$ dominated at most stations and depths, with rates as high as 1.6±0.2 nM $N_2O$/day. Hybrid $N_2O$ production, one of the mechanisms by which ammonia-oxidizing archaea produce $N_2O$, had rates as high as 0.23±0.08 nM $N_2O$/day that peaked in both the near-surface and deep $N_2O$ concentration maxima. We inferred from the ${}^{45}N_2O^\alpha$ and ${}^{45}N_2O^\beta$ data that hybrid $N_2O$ production by ammonia-oxidizing archaea may have a variable site preference that depends on the [15]N content of each substrate. We also found that the rates and yields of hybrid

$N_2O$ production exhibited a clear $[O_2]$ inhibition curve, with the hybrid $N_2O$ yields as high as 20% at depths where dissolved $[O_2]$ was 0 µM but nitrification was still active. Finally, we identified a few incubations with dissolved $[O_2]$ up to 20 µM



where N$_2$O production from NO$_3^-$ was still active. A relatively high O$_2$ tolerance for N$_2$O production via denitrification has implications for the feedbacks between marine deoxygenation and greenhouse gas cycling.

**1. Introduction**

Nitrous oxide (N$_2$O) is one of the lesser-known greenhouse gases, yet its potential to warm the environment, on a per-molecule basis, is immense. N$_2$O has a global warming potential 273 times that of carbon dioxide (Smith et al. 2021), and its atmospheric mixing ratio is increasing at a rate of 0.85±0.003 ppb/year (Tian et al. 2020). In the ocean, hotspots of N$_2$O production and flux to the atmosphere occur in marine oxygen deficient zones (ODZs), where steep redox gradients allow

for multiple, overlapping N$_2$O production processes to occur (Codispoti and Christensen 1985). ODZs have expanded over the last 60 years (Stramma et al. 2008; Breitburg et al. 2018) and will likely continue to do so as the oceans warm (Oschlies et al. 2018), although fate of the anoxic cores of ODZs ([O$_2$] ≤ 20 mmol/kg) remains uncertain (Cabré et al. 2015; Bianchi et al. 2018; Busecke et al. 2022). Without a clear picture of N$_2$O cycling in these regions, it is impossible to predict how climate change will impact marine production of this powerful greenhouse gas.

Ammonia-oxidizing archaea are one of the most abundant organisms in the ocean (Schleper et al. 2005; Francis et al. 2005; Wuchter et al. 2006; Santoro et al. 2010; Newell et al. 2011) and one of the primary sources of marine N$_2$O (Santoro and Casciotti 2011; Löscher et al. 2012; Toyoda et al. 2019). Our understanding of the mechanisms by which ammonia-oxidizing archaea produce N$_2$O is rapidly evolving. Isotopic evidence continues to suggest that N$_2$O production by ammonia-oxidizing

archaea proceeds — at least in part — via a hybrid mechanism that combines nitrogen (N) derived from nitrite (NO$_2^-$) and ammonium (NH$_4^+$) to form the N$_2$O molecule (Stieglmeier et al. 2014; Trimmer et al. 2016; Frame et al. 2017; Frey et al. 2020, 2023). New evidence indicates that in aerobic conditions, ammonia-oxidizing archaea can produce N$_2$O both as a by-product of hydroxylamine oxidation and via hybrid N$_2$O production, and that the ratio of these processes depends on the ratio of NH$_4^+$ to NO$_2^-$ available to the archaea (Wan et al. 2023). Hydroxylamine and nitric oxide (NO) likely occur as

intermediates during archaeal ammonia (NH$_3$) oxidation (Vajrala et al. 2013; Martens-Habbena et al. 2015; Kozlowski et al. 2016; Lancaster et al. 2018), but the exact mechanism and enzymology of archaeal N$_2$O production remains unknown (Carini et al. 2018; Stein 2019). In anaerobic conditions, ammonia-oxidizing archaea are capable of NO dismutation to O$_2$ and N$_2$, which may involve N$_2$O as an intermediate (Kraft et al. 2022). Ammonia-oxidizing bacteria, more common in regions that are nutrient replete, produce N$_2$O as a byproduct of hydroxylamine oxidation (Cohen and Gordon 1979), and via

nitrifier-denitrification as oxygen concentrations decline (Goreau et al. 1980; Wrage et al. 2001; Stein and Yung 2003) and nitrite concentrations rise (Frame and Casciotti, 2010).



In low-oxygen waters, denitrifying organisms produce $N_2O$ as an intermediate during organic matter remineralization (Zumft 1997; Naqvi et al. 2000; Dalsgaard et al. 2014). A growing body of evidence suggests that most $N_2O$ production from

denitrification occurs via nitrate ($NO_3^-$) reduction to $NO_2^-$ and $N_2O$ entirely within the cell, i.e., without exchange with an extracellular $NO_2^-$ pool (Ji et al. 2015, 2018; Casciotti et al. 2018; Frey et al. 2020; Kelly et al. 2021; Monreal et al. 2022; Toyoda et al. 2023). $N_2O$ production from extracellular $NO_2^-$ tends to occur at lower rates (Ji et al. 2015, 2018; Frey et al. 2020). Historically, $N_2O$ production from denitrification was thought to cease at dissolved oxygen concentrations above 2-3 µM (Dalsgaard et al. 2014), but more recent data suggest that $N_2O$ production from $NO_3^-$ can occur at oxygen levels as high

as 30 µM (Ji et al. 2018; Frey et al. 2020). Denitrification is also the primary microbial $N_2O$ sink (Sun et al. 2021b). $N_2O$ consumption via denitrification is more sensitive to oxygen than $N_2O$ production via denitrification (Farías et al. 2009; Dalsgaard et al. 2014; Babbin et al. 2015; Frey et al. 2020), although the oxygen inhibition constant for $N_2O$ consumption remains difficult to define (Sun et al. 2021b). $N_2O$ may also be consumed through $N_2O$ fixation, although the importance of $N_2O$ fixation in the ocean has yet to be determined (Farías et al. 2013; Si et al. 2023).


A clear picture of the complex $N_2O$ dynamics in ODZs requires the identification and quantification of each $N_2O$ cycling process. The stable nitrogen and oxygen isotopes of $N_2O$ can provide quantification of — and distinction among — the potential mechanisms (Kim and Craig 1990; Toyoda and Yoshida 1999; Rahn and Wahlen 2000). The isotopic content of the individual nitrogen and oxygen atoms in the $N_2O$ molecule are expressed in delta notation, defined as $\delta(^{15}N)$ or $\delta(^{18}O) =$

$(R_{sample}/R_{standard}-1) \times 1{,}000$, where $R_{standard}$ for $\delta(^{15}N)$ and $\delta(^{18}O)$ are the ratios $^{15}N/^{14}N$ of air and $^{18}O/^{16}O$ of Vienna Standard Mean Ocean Water (VSMOW), respectively (Kim and Craig 1990; Toyoda and Yoshida 1999; Rahn and Wahlen 2000). In addition to the bulk nitrogen and oxygen isotope ratios in $N_2O$, we can measure the isotopic content of the two nitrogen atoms that occur in unique chemical environments: an inner (α) nitrogen atom, and an outer (β) nitrogen atom (Brenninkmeijer and Röckmann 1999; Toyoda and Yoshida 1999). The difference in the $^{15}N$ content of these two atoms is

often referred to as the 'site preference' and is defined as $\delta(^{15}N^{sp}) = \delta(^{15}N^\alpha) - \delta(^{15}N^\beta)$. $\delta(^{15}N^{sp})$ exhibits distinct values for different $N_2O$ production processes, unlinked to the isotopic value of the substrate, for processes that draw both nitrogen atoms from the same substrate pool (Toyoda et al. 2002, 2005; Sutka et al. 2003, 2004, 2006; Frame and Casciotti 2010). Because they serve as additional tracers, natural abundance $N_2O$ isotopocules ($\delta(^{15}N^{sp})$, $\delta(^{15}N^\alpha)$, $\delta(^{15}N^\beta)$, and $\delta(^{18}O)$) have been used extensively to quantify $N_2O$ cycling in soils (Pérez et al. 2001; Yamulki et al. 2001; Lewicka-Szczebak et al.

2017; Verhoeven et al. 2019), the atmosphere (Rahn and Wahlen 2000; Yoshida and Toyoda 2000; Prokopiou et al. 2017; Yu et al. 2020), and the ocean (Toyoda et al. 2002, 2019, 2021, 2023; Popp et al. 2002; Yamagishi et al. 2005, 2007; Westley et al. 2006; Farías et al. 2009; Bourbonnais et al. 2017, 2023; Casciotti et al. 2018; Kelly et al. 2021).

Previous studies have used $^{15}N$ tracer experiments to measure $N_2O$ production rates in ODZs (Ji et al. 2015, 2018; Frey et al.

2020, 2023). These studies used the accumulation of $^{45}N_2O$ and $^{46}N_2O$ resulting from the addition of $^{15}N$-labeled substrates such as $^{15}NH_4^+$ and $^{15}NO_2^-$ to measure $N_2O$ production rates. To our knowledge, however, the isotopomer measurement has



never been applied to $^{15}$N-tracer experiments to track $^{15}$N from different substrates into the α and β positions of the N$_2$O molecule. Here, we present data from $^{15}$N-tracer experiments where we measured the production of N$_2$O isotopomers with $^{15}$N in the α position ($^{45}$N$_2$O$^\alpha$) and $^{15}$N in the β position ($^{45}$N$_2$O$^\beta$) from $^{15}$N-labeled NH$_4^+$, NO$_2^-$, and NO$_3^-$. We used these

measurements to quantify the rates and oxygen dependence of N$_2$O production from solely NH$_4^+$, hybrid production, and denitrification. Based on these measurements, we inferred a potentially variable site preference ($\delta(^{15}N^{sp})$) for hybrid N$_2$O.

## 2. Methods

### 2.1 Sampling sites

Experiments were performed at three stations in the eastern tropical North Pacific on the R/V Sally Ride in March-April

2018 (Fig. 1). Station PS1 (113º W, 10º N) was on the edge of the oxygen deficient region, station PS2 (105º W, 16º N) was near the geographic center of the ODZ, and station PS3 (102º W, 18º N) was 12 miles from the coast of Mexico. Samples were collected from 30 L Niskin bottles mounted on a 12-place rosette with a conductivity-temperature-depth profiler and sensors for chlorophyll *a* fluorescence and dissolved O$_2$ (Sea-Bird SBE 43 oxygen sensor). A switchable trace oxygen (STOX) sensor with a detection limit of 10 nM (Revsbech et al. 2009) was also mounted on the rosette for at least one cast

per station (Table S1). The cruise took place during a weak La Niña event (Ocean Niño Index = −0.6°C; NOAA Climate Prediction Center).

Ambient [NO$_2^-$] and [NH$_4^+$] were measured shipboard with standard colorimetric (Grasshoff et al. 1999) and fluorometric methods (Grasshoff et al. 1999; Holmes et al. 1999), respectively. Ambient [NO$_3^-$] was measured at Stanford University

using a Westco SmartChem 200 Discrete Analyzer (detection limit = 83 nM, precision = 0.6 μM). Ambient [N$_2$O] was measured via an isotope ratio mass spectrometer (IRMS) as part of a prior study (Kelly et al. 2021).



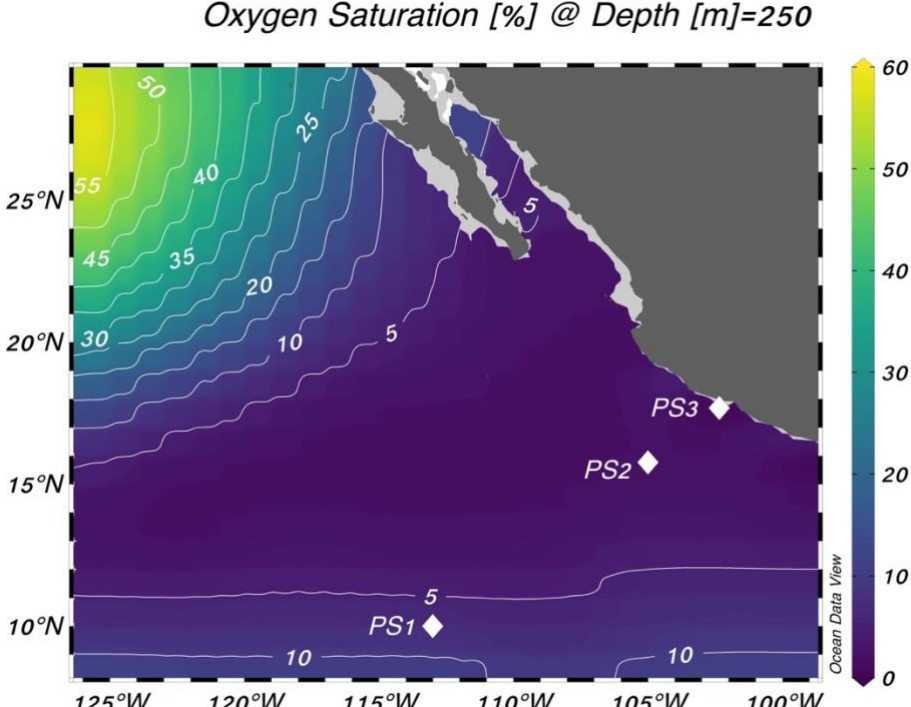

**Figure 1. Locations of the three stations sampled for this study. Stations are plotted on top of World Ocean Atlas oxygen saturation (%) at 250 m depth (World Ocean Atlas, 2013).**

**2.2 Sample collection**

Incubation depths were chosen to target prominent hydrographic features: the primary nitrite maximum, shallow and deep oxyclines, oxic-anoxic interfaces above and below the ODZ, secondary chlorophyll maximum, and secondary nitrite maximum (Table S1). Incubation samples were filled directly from Niskin bottles into 160 mL glass serum bottles (Wheaton) using Tygon tubing. Incubation bottles were overflowed three times before being capped and sealed with no

headspace using gray butyl rubber septa (National Scientific) and aluminum crimp seals. To minimize oxygen contamination during sampling, incubation bottles were overflowed in a secondary container filled with suboxic water from the same depth, and Niskin bottles were vented with carbon dioxide gas to displace the withdrawn water. The butyl rubber stoppers were deoxygenated in a He-flushed anaerobic chamber for ~1 week prior to sampling.

After sample collection, a 2 mL He headspace was created in each bottle by displacing 2 mL sample from the bottle with He. At most (all but two) anoxic depths at stations PS2 and PS3, samples were sparged with He gas for 90 minutes at a flow rate of at least 100 mL/min, equivalent to 56 volume exchanges, to remove potential oxygen contamination introduced during sampling. After sparging, 100 µL of 1030 ppm $N_2O$ in He (4 nmol $N_2O$) was introduced back into each bottle for a final concentration of 26 nM to provide a constant background of $N_2O$ for later isotopic analysis (Fig. S4a). The isotopic content



of this N$_2$O carrier, measured independently via IRMS (McIlvin and Casciotti 2010; Kelly et al. 2023), was $\delta(^{15}N^{\alpha}) = -1.5\pm0.2$ ‰, $\delta(^{15}N^{\beta}) = 0.2\pm0.4$ ‰, $\delta(^{15}N^{bulk}) = -0.65\pm0.08$ ‰, and $\delta(^{18}O) = 37.4\pm0.3$ ‰.

A total of 27 incubation samples were produced at each experimental depth, comprised of triplicate samples for each of three time points and three tracers. For each station and depth, nine samples were amended with $^{15}NH_4Cl$ (98.8 *atm*%, Sigma-

Aldrich) to a final concentration of 0.501 µM, plus Na$^{14}NO_2$ to a final concentration 1.01 µM. Nine samples were amended with Na$^{15}NO_2$ (98.8 *atm*%, Sigma-Aldrich) to a final concentration of 5.00 µM, plus $^{14}NH_4Cl$ to a final concentration of 0.510 µM. Finally, nine samples were amended with K$^{15}NO_3$ (98.8 *atm*%, Sigma-Aldrich) to a final concentration of 1.00 µM, plus 1.01 µM Na$^{14}NO_2$ and 0.510 µM $^{14}NH_4Cl$. Note that Na$^{15}NO_2$ tracer was added at a higher concentration than the other tracers or the Na$^{14}NO_2$ carrier; this discrepancy was due to a miscalculation that was caught midway through the cruise

but the high tracer addition was retained for the sake of consistency. The $NO_2^-$ and $NH_4^+$ tracer and carrier additions were confirmed via $[NO_2^-]$ and $[NH_4^+]$ measurements of sample aliquoted from each bottle immediately before samples were measured for N$_2$O isotopic content, using colorimetric and fluorometric techniques (Grasshoff et al. 1999; Holmes et al. 1999). The Na$^{14}NO_2$ and $^{14}NH_4Cl$ amendments served two purposes: 1) to have enough total $NO_2^-$ for isotopic analysis of $^{15}NO_2^-$ produced from $^{15}NH_4^+$, and 2) to minimize isotope dilution of the substrate pool, which can cause underestimation of

rates with low substrate additions. The fraction of $^{15}N$ in substrate pools was thus 56%–100% for $^{15}N$-$NH_4^+$, 65%–100% for $^{15}N$-$NO_2^-$, and 2%–92% for $^{15}N$-$NO_3^-$ experiments. Three samples for each tracer were terminated immediately after tracer addition with the addition of 100 µL saturated mercuric chloride (HgCl$_2$) solution. These also served as abiotic controls. The remaining samples were incubated at 12º C in the dark; three samples per tracer were terminated at 12 hours and at 24 hours with 100 µL saturated HgCl$_2$. All samples were incubated at 12º C, which was chosen as an intermediate temperature that

approximated subsurface conditions. After termination, samples were stored at room temperature (~20º C) in the dark until isotope analysis.

**2.3 Chemiluminescent optode oxygen measurements**

Eight 160 mL glass serum bottles were prepared with a chemiluminescent oxygen optode spot (PyroScience) affixed to the inner glass wall with silicone glue. These bottles were incubated alongside experimental bottles to monitor dissolved [O$_2$]

concentrations during incubations. At stations PS2 and PS3, two optode bottles per depth were filled, purged, amended with the N$_2$O carrier, and incubated without the addition of tracer or HgCl$_2$. At each timepoint, [O$_2$] was measured in each sensor bottle for at least 10 minutes using fiber optic cables paired to the optodes (PyroScience). The detection limit for each optode was calculated from the final asymptotic value reached when monitoring its bottle filled with seawater that had been purged with He for 90 minutes at a minimum flow rate of 100 mL/min. Those detection limits were specific to each optode and

varied from 146 – 880 nM [O$_2$]. The fiber optic cables were calibrated with a 2-point calibration before each measurement against a sodium sulfite solution (30 g/L in DI, or 0.24 M) and surface seawater saturated with atmospheric O$_2$ at 12º C (270 µM [O$_2$], based on a salinity = 35 psu and temperature = 12º C) (Garcia and Gordon 1992). Optical oxygen sensors are



susceptible to interference from NO, which results in higher [O$_2$] measurements (Kraft et al. 2022), so optode O$_2$ measurements may represent an overestimate in experiments with especially high rates of NO production. Given maximum

ammonia (NH$_3$) oxidation rates of 4.68±0.07 nM N/day, the release of equivalent amounts of NO would result in an [O$_2$] overestimate of 0.745 nM during a 24-hour incubation, based on the interference curve calculated by Kraft et al. (2022) ([O$_2$] overestimate = 0.159[NO]). The optode [O$_2$] measurements were adjusted for the detection limit specific to each sensor spot; optode [O$_2$] for each experiment was calculated as the mean measured [O$_2$] at each of the three timepoints. No optode measurements were made at station PS1, since this station lacked a secondary NO$_2^-$ maximum and thus incubations

performed at low-oxygen depths were not expected to occur under functional anoxia.

Optode [O$_2$] generally agreed with ambient [O$_2$] measured by the Sea-Bird oxygen sensor attached to the rosette (Fig. S1). Two important exceptions were the experiments at the base of ODZ and in the deep ODZ core at station PS2, which were not purged before tracer addition. As a result, the ambient [O$_2$] at these depths was below detection of the Sea-Bird sensor,

but the optode [O$_2$] in the incubation bottles from these depths were 17.7±0.1 µM and 19.2±0.8 µM, respectively (Fig. S1, Table S1). Additionally, two depths that were suboxic (and thus not sparged prior to tracer addition) had higher optode [O$_2$] than ambient [O$_2$]: in the deep oxycline at station PS2, ambient [O$_2$] was 6.8 µM and optode [O$_2$] was 14.8±0.2 µM; at the oxic-anoxic interface at station PS2, ambient [O$_2$] was 6.5 µM and optode [O$_2$] was 9.48±0.09 µM (Fig. S1, Table S1). Because of these few exceptions, we always report both optode and ambient [O$_2$] in the following figures and text.

**2.4 Nitrous oxide isotopocule measurements**

Immediately prior to isotope analysis, a 5 mL aliquot was removed from each sample by syringe and replaced with He gas. These aliquots were refrigerated until analysis for [NO$_2^-$] and [NH$_4^+$] to check tracer and carrier additions, as mentioned above. After this aliquot was removed, $^{14}$NH$_4$Cl, Na$^{14}$NO$_2$, or K$^{14}$NO$_3$ carrier was added to each sample to bring $^{15}$N tracer levels below 5000 ‰. Note that these carrier additions were *different* from the $^{14}$N carrier added to each incubation alongside

$^{15}$N tracer; the purpose of the later carrier additions was to prevent exposure of the IRMS system to highly $^{15}$N-enriched substrates.

Samples were measured for N$_2$O concentrations and $^{15}$N isotopocules on a custom-built purge and trap system coupled to a Thermo Finnigan DELTA V Plus IRMS, which was run in continuous flow mode and configured to measure *m/z* 30, 31, 44,

45, and 46 (McIlvin and Casciotti 2010). These measurements were made under normal operating conditions, using an ionization energy of 124 eV, emission current of 1.50 mA, and accelerating voltage of 3 kV. Samples were analyzed alongside reference materials (B6, S2, and atmosphere-equilibrated seawater) to calibrate the IRMS for scrambling in the ion source with the pyisotopomer software package in Python (Kelly et al. 2023). The number ratios of isotopomers $^{14}$N$^{15}$NO and $^{15}$N$^{14}$NO were calculated as in Kelly et al., 2023, with the following modifications to account for the production of

$^{15}$N$^{15}$NO.



In natural abundance samples, pyisotopomer solves the following four equations to obtain $^{15}R^{\alpha}$ and $^{15}R^{\beta}$:

$$^{45}R = {^{15}R^{\alpha}} + {^{15}R^{\beta}} + {^{17}R} \qquad (1)$$

$$^{46}R = \left({^{15}R^{\alpha}} + {^{15}R^{\beta}}\right){^{17}R} + {^{18}R} + {^{15}R^{\alpha}}\,{^{15}R^{\beta}} \qquad (2)$$

$$^{17}R / {^{17}R_{\text{VSMOW}}} = \left({^{18}R}/0.0020052\right)^{\beta}\left[\varDelta({^{17}\text{O}}) + 1\right] \qquad (3)$$

$$^{31}R = \frac{(1-\gamma)\,{^{15}R^{\alpha}} + \kappa\,{^{15}R^{\beta}} + {^{15}R^{\alpha}}\,{^{15}R^{\beta}} + {^{17}R}\left[1 + \gamma\,{^{15}R^{\alpha}} + (1-\kappa)\,{^{15}R^{\beta}}\right]}{1 + \gamma\,{^{15}R^{\alpha}} + (1-\kappa)\,{^{15}R^{\beta}}} \qquad (4)$$

In these equations, the term $({^{15}R^{\alpha}})({^{15}R^{\beta}})$ represents the statistically expected contribution of $^{15}\text{N}^{15}\text{N}^{16}\text{O}$ to the $^{46}R$ and $^{31}R$ ion number ratios, based on the probabilities of forming $^{15}\text{N}^{15}\text{N}^{16}\text{O}$. The probability of getting $^{15}\text{N}$ in $\text{N}^{\alpha}$ is assumed to be equal

to $^{15}R^{\alpha}$ and the probability of getting $^{15}\text{N}$ in $\text{N}^{\beta}$ is assumed to be equal to $^{15}R^{\beta}$; furthermore, the two probabilities are assumed to be independent, so the probability of getting $^{15}\text{N}$ in both positions would be $({^{15}R^{\alpha}})({^{15}R^{\beta}})$ (Kaiser et al. 2004). Predicting the concentration of $^{15}\text{N}^{15}\text{N}^{16}\text{O}$ from $^{15}R^{\alpha}$ and $^{15}R^{\beta}$ is a reasonable assumption for natural abundance samples, where the concentration of $^{15}\text{N}^{15}\text{N}^{16}\text{O}$ is extremely low (Magyar et al. 2016; Kantnerová et al. 2022).

For $^{15}\text{N}$-labeled samples, however, we cannot predict $^{15}\text{N}^{15}\text{N}^{16}\text{O}$ from $^{15}R^{\alpha}$ and $^{15}R^{\beta}$. This is because the relationship between the formation of $^{15}\text{N}^{15}\text{N}^{16}\text{O}$, $^{14}\text{N}^{15}\text{N}^{16}\text{O}$, and $^{15}\text{N}^{14}\text{N}^{16}\text{O}$ depends on production mechanism and the atom fraction of the substrate. For example, in $^{15}\text{N-NO}_2^{-}$ experiments when the denitrification rate is high, there may be far more $^{15}\text{N}^{15}\text{N}^{16}\text{O}$ production than the amount predicted from $^{15}R^{\alpha}$ and $^{15}R^{\beta}$. To account for this, we added a term to the equations for $^{46}R$ and $^{31}R$:

$$^{46}R = \left({^{15}R^{\alpha}} + {^{15}R^{\beta}}\right){^{17}R} + {^{18}R} + \left({^{15}R^{\alpha}}\,{^{15}R^{\beta}}\right)_{t0} + {^{15}\text{N}^{15}\text{N}^{16}\text{O}_{excess}} \qquad (5)$$

$$^{31}R = \frac{(1-\gamma)\,{^{15}R^{\alpha}} + \kappa\,{^{15}R^{\beta}} + \left({^{15}R^{\alpha}}\,{^{15}R^{\beta}}\right)_{t0} + {^{15}\text{N}^{15}\text{N}^{16}\text{O}_{excess}} + {^{17}R}\left[1 + \gamma\,{^{15}R^{\alpha}} + (1-\kappa)\,{^{15}R^{\beta}}\right]}{1 + \gamma\,{^{15}R^{\alpha}} + (1-\kappa)\,{^{15}R^{\beta}}} \qquad (6)$$

where $^{15}\text{N}^{15}\text{N}^{16}\text{O}_{excess}$ represents the amount of $^{15}\text{N}^{15}\text{N}^{16}\text{O}$ produced in the sample over the course of the experiment. To quantify $^{15}\text{N}^{15}\text{N}^{16}\text{O}_{excess}$ in tracer samples, we assumed that any increase in $^{46}R$ over the course of the experiment is due to added $^{15}\text{N}^{15}\text{N}^{16}\text{O}$, i.e., that $\delta(^{18}\text{O})$ remains constant. This should be a reasonable assumption — while denitrification and $\text{N}_2\text{O}$ consumption could cause natural abundance-level increases in $\delta(^{18}\text{O})$ and thus $^{46}R$ (10's of per mil), $\text{N}_2\text{O}$ production from $^{15}\text{N}$-labeled substrates are expected to cause much greater increases in $^{46}R$ (100's to 1,000's of per mil). We calculated

the term $^{15}\text{N}^{15}\text{N}^{16}\text{O}_{excess}$ by subtracting the mean $^{46}R$ at $t_0$ from the measured $^{46}R$ in later timepoints using the pyisotopomer



template designed for tracer experiments (Kelly 2023). Then, we used the "Tracers" function in pyisotopomer, which takes this $^{15}N^{15}N^{16}O_{excess}$ into account, to calculate $^{15}R^{\alpha}$ and $^{15}R^{\beta}$.

The concentration of $^{44}N_2O$ in each sample was calculated from $m/z$ 44 peak area and a linear conversion factor, divided by the sample volume (McIlvin and Casciotti 2010). The concentrations of $^{45}N_2O^{\alpha}$, $^{45}N_2O^{\beta}$, and $^{46}N_2O$ were finally calculated by multiplying $^{15}R^{\alpha}$, $^{15}R^{\beta}$, and $^{46}R$ by the average $[^{44}N_2O]$ across all timepoints for that tracer experiment. Average values of $[^{44}N_2O]$ were used to avoid aliasing random variability in $[^{44}N_2O]$ over increases in $^{15}R^{\alpha}$, $^{15}R^{\beta}$, and $^{46}R$. The analytical precisions for $N_2O$ isotopocule measurements, based on the pooled standard deviations of reference materials run alongside samples, were $\delta(^{15}N^{\alpha}) = 4.4$ ‰, $\delta(^{15}N^{\beta}) = 3.4$ ‰, $\delta(^{15}N^{bulk}) = 3.5$ ‰, and $\delta(^{18}O) = 2.1$ ‰. The analytical precision was lower

than that in a similar natural abundance dataset (Kelly et al. 2021) due to minor $^{15}N$ carry-over in some of the standards analyzed immediately following highly enriched samples.

**2.5 Nitrite and nitrate isotope measurements**

After $N_2O$ analysis, approximately 2 mL sample remained in each bottle, which was prepared for analysis of $\delta(^{15}N\text{-}NO_2^-$ $+NO_3^-)$, $\delta(^{15}N\text{-}NO_3^-)$, or $\delta(^{15}N\text{-}NO_2^-)$, to determine the rates of $NH_3$ oxidation, $NO_2^-$ oxidation, and $NO_3^-$ reduction,

depending on the tracer experiment. Samples incubated with $^{15}N\text{-}NH_4^+$ were prepared for $\delta(^{15}N\text{-}NO_2^-+NO_3^-)$ analysis using the denitrifier method (Sigman et al. 2001; Casciotti et al. 2002), with updates from McIlvin and Casciotti (2011), to determine rates of $NH_3$ oxidation. These samples were run on a Thermo-Finnigan DELTA$^{PLUS}$ XP IRMS alongside a process blank and reference materials USGS32, USGS34, and USGS35 (Böhlke et al. 2003) to obtain $\delta(^{15}N\text{-}NO_2^-+NO_3^-)$.

Samples incubated with $^{15}N\text{-}NO_2^-$ were first treated with 5% sulfamic acid (weight-by-volume, or 10 mM final concentration) to remove $^{15}N\text{-}NO_2^-$ (Granger and Sigman 2009), then prepared with the denitrifier method for $\delta(^{15}N\text{-}NO_3^-)$ analysis (Sigman et al. 2001; Casciotti et al. 2002; McIlvin and Casciotti 2011) to determine rates of $NO_2^-$ oxidation. For these analyses, reference materials USGS32, USGS34, and USGS35 (Böhlke et al. 2003) were also treated with 5% sulfamic acid and prepared with the denitrifier method alongside samples. Despite using a moderate level of sulfamic acid, high $t_0$

$\delta(^{15}N)$ values (>1000 ‰) for incubations with low ambient $[NO_3^-]$ indicated that the sulfamic treatment either did not fully remove all of the $^{15}N\text{-}NO_2^-$ tracer or that the sulfamic treatment chemically converted some $^{15}N\text{-}NO_2^-$ tracer to $^{15}N\text{-}NO_3^-$ (Fig. S2). Regardless, this would have shifted all three timepoints equally, and thus should not introduce a bias into the slope of $\delta(^{15}N\text{-}NO_3^-)$ with time and the rates calculated there from.

Finally, samples incubated with $^{15}N\text{-}NO_3^-$ were prepared for $\delta(^{15}N\text{-}NO_2^-)$ isotopic analysis with the azide method (McIlvin and Altabet 2005) to determine rates of $NO_3^-$ reduction to $NO_2^-$. The 2 mL of remaining sample was transferred into 20 mL vials, where it was prepared alongside reference materials RSIL-N23, -N7373 and -N10219 (Casciotti et al. 2007). Reference





materials were diluted from 200 mM working stocks into 3 mL $NO_2^-$-free seawater water in 5 and 10 nmol quantities of $NO_2^-$ to correct for the contribution of a consistent blank to a range of sample sizes. The analytical precisions for $\delta(^{15}N\text{-}NO_x^-$

), $\delta(^{15}N\text{-}NO_3^-)$, and $\delta(^{15}N\text{-}NO_2^-)$ were 0.9 ‰, 1.2 ‰, and 0.4 ‰, respectively. The $\delta(^{15}N)$ analytical precision for the denitrifier method is typically better but tracer measurements tend to have lower analytical precision than natural abundance measurements.

The rates of $NH_3$ and $NO_2^-$ oxidation were calculated using a weighted least squares linear regression through product $^{15}N$

vs. incubation time (Fig. S3). Each sample was weighted by its uncertainty, which was calculated based on the slope and intercept of the calibration curve, blank peak area, and sample peak area (Appendix A). Although using this uncertainty calculation is complex, it allows for the assessment of relative error, and for the inclusion of low-peak area samples that had high enough $\delta(^{15}N)$ enrichments such that the relative error remained below 10% (and in most cases, 1%). A weighted least squares regression was used in place of an ordinary least squares regression to prevent samples with high uncertainties from

biasing the slope estimate (e.g., two samples in Fig. S3b). Then, the rate was calculated by:

$$rate \text{ (nM N/day)} = \frac{slope(^{15}F_{product})[product]}{^{15}F_{substrate}} \qquad (7)$$

where $slope(^{15}F_{product})$ is the slope of the atom fraction of $^{15}N$ in the product vs. incubation time, $[product]$ is the mean product concentration (e.g., $NO_3^-$ in a $NO_2^-$ oxidation experiment), and $^{15}F_{substrate}$ is the atom fraction of $^{15}N$ in the substrate (e.g., $NO_2^-$ in a $NO_2^-$ oxidation experiment). Rates of $NO_3^-$ reduction were calculated using equation (7), but with an ordinary least squares regression since $\delta(^{15}N\text{-}NO_2^-)$ measurements were not accompanied by individual uncertainties.

**2.6 Modeling N2O production mechanisms**

A simple time-dependent model was constructed to infer the rates and mechanisms of $N_2O$ production from the measured isotopocule time courses in each incubation experiment. While it is possible to calculate rates of hybrid and bacterial $N_2O$ production with linear regressions of $^{45}N_2O$ and $^{46}N_2O$ with time (Trimmer et al. 2016), these calculations cannot take into account $^{15}N$ exchange between substrates, and more importantly produce separate rate estimates for separate tracer

experiments. They also do not leverage the additional information provided by $N_2O$ isotopomers. We sought to solve for a common set of $N_2O$ production rate constants across the three parallel tracer experiments at a given station and depth, wherein the only differences between each tracer experiment were the starting concentrations of $^{14}N$ and $^{15}N$ in $NH_4^+$, $NO_2^-$, and $NO_3^-$ (Fig. 2). The model encoded four different $N_2O$ producing pathways: 1) **production from solely $NH_4^+$**, which includes $N_2O$ from hydroxylamine oxidation (Wan et al., 2023 Pathway 1), hybrid production using cellular $NO_2^-$ (Wan et

al., 2023 Pathway 2), and nitrifier-denitrification using cellular $NO_2^-$; 2) **hybrid production** using extracellular $NO_2^-$ (Wan et al., 2023 Pathway 3); 3) **production from $NO_2^-$,** i.e. denitrification or nitrifier-denitrification using extracellular $NO_2^-$; and 4) **production from $NO_3^-$**, i.e. denitrification using cellular $NO_2^-$ (Fig. 2). Using this model, the relative importance of



each of these pathways was determined at each incubation depth based on the production of $^{15}$N-labeled N$_2$O isotopocules in parallel experiments supplied with different $^{15}$N substrates.

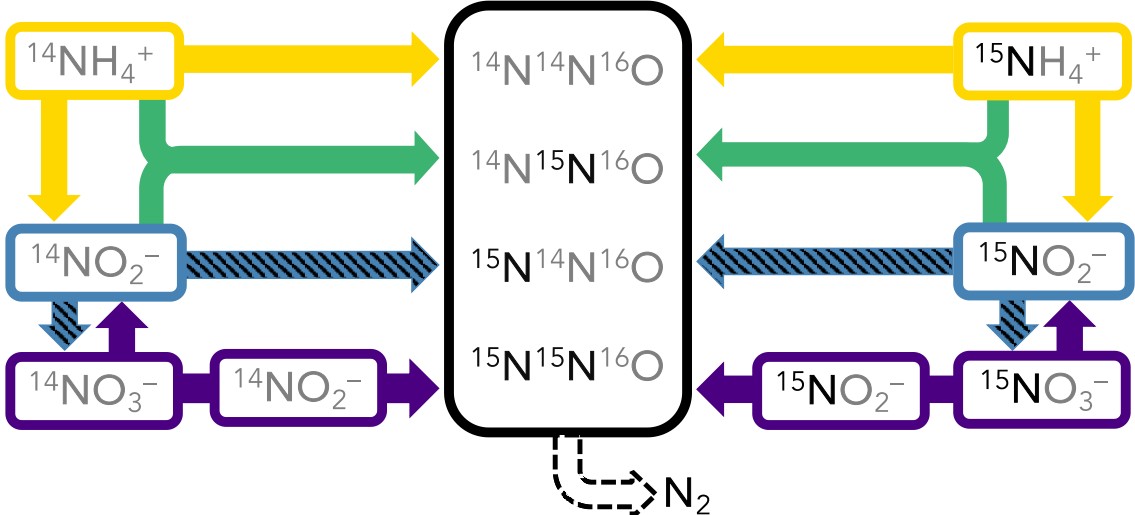

| Pathway name | Includes N$_2$O production from… | Corresponding process in Wan et al. (2023) |
|---|---|---|
| Production from solely NH$_4^+$ | Hydroxylamine oxidation | Pathway 1 |
| | Hybrid production using cellular NO$_2^-$ | Pathway 2 |
| | Nitrifier-denitrification using cellular NO$_2^-$ | N/A |
| Hybrid production | Hybrid production using extracellular NO$_2^-$ | Pathway 3 |
| Production from NO$_2^-$ | Denitrification or nitrifier-denitrification using extracellular NO$_2^-$ | N/A |
| Production from NO$_3^-$ | Denitrification using cellular NO$_2^-$ | N/A |


**Figure 2. Schematic of the forward-running model used to solve for rates of N$_2$O production. Horizontal arrows represent processes whose rates are solved for, while vertical arrows represent processes whose rates are prescribed based on our experimental results. The model solves for 2$^{nd}$-order rate constants for four N$_2$O-producing processes: 1) production from solely NH$_4^+$ (yellow horizontal arrows), which includes N$_2$O from hydroxylamine oxidation (Wan et al., 2023 Pathway 1), hybrid production using cellular NO$_2^-$ (Wan et al., 2023 Pathway 2), and nitrifier-denitrification using cellular NO$_2^-$; 2) hybrid production using NH$_4^+$ and extracellular NO$_2^-$ (green arrows, Wan et al., 2023 Pathway 3); 3) production from NO$_2^-$, i.e. denitrification using extracellular NO$_2^-$ (blue hatched horizontal arrows); and 4) production from NO$_3$, i.e. denitrification or nitrifier-denitrification using cellular NO$_2^-$ (indigo horizontal arrows). The model also solves for $f$, the proportion of N$^\alpha$ derived from NO$_2^-$ during hybrid N$_2$O production. NH$_3$ oxidation (yellow vertical arrows), NO$_2^-$ oxidation (blue hatched vertical arrows), and NO$_3^-$ reduction to NO$_2^-$ (indigo vertical arrows) are modeled as first-order rates to account for $^{15}$N exchange between substrate pools, as described in the main text. Finally, N$_2$O consumption (black dashed arrow) is modeled as first-order to N$_2$O. It is assumed that while the distribution of $^{15}$N in each tracer experiment at a given station and depth is different, the overall rates and mechanisms of N$_2$O production are the same regardless of which substrate is labeled. The model is optimized against the observed $^{46}$N$_2$O, $^{45}$N$_2$O$\alpha$, $^{45}$N$_2$O$\beta$, and $^{44}$N$_2$O at each timepoint in each tracer experiment (black box).**








The concentration of each nitrogen species was modeled as:

$$N_{t+1} = N_t + \Delta t \left( \sum_{n=1}^{i} J_i^{source} - \sum_{n=1}^{k} J_k^{sink} \right) \tag{8}$$

where $N_t$ is the concentration of a given N species (e.g., $NH_4^+$, $NO_2^-$, $NO_3^-$, or $N_2O$) at time $t$, $N_{t+1}$ is its concentration at time $t+1$, $\Delta t$ represents the model timestep (days), $\sum_{n=1}^{i} J_i^{source}$ is the sum of $i$ individual source processes of that species (nM/day), and $\sum_{n=1}^{k} J_k^{sink}$ is the sum of $k$ individual sink processes of that species (nM/day).

The pattern of $N_2O$ isotopocule production for a given process was set by the total rate $J$ of $N_2O$ production for that process, multiplied by the probability of forming each isotopocule from a given pair of substrates. The probabilities of forming each isotopocule were based on the atom fractions of the two substrates from which the nitrogen atoms in $N_2O$ are derived:

$$P\left( ^{46}N_2O \right) = \left( ^{15}F_1 \right)\left( ^{15}F_2 \right) \tag{9}$$

$$P\left( ^{45}N_2O^{\alpha} \right) = f\left( ^{15}F_1 \right)\left( 1 - {}^{15}F_2 \right) + (1-f)\left( 1 - {}^{15}F_1 \right)\left( ^{15}F_2 \right) \tag{10}$$

$$P\left( ^{45}N_2O^{\beta} \right) = (1-f)\left( ^{15}F_1 \right)\left( 1 - {}^{15}F_2 \right) + f\left( 1 - {}^{15}F_1 \right)\left( ^{15}F_2 \right) \tag{11}$$

$$P\left( ^{44}N_2O \right) = \left( 1 - {}^{15}F_1 \right)\left( 1 - {}^{15}F_2 \right) \tag{12}$$

where $P(^{46}N_2O)$, $P(^{45}N_2O^{\alpha})$, $P(^{45}N_2O^{\beta})$, and $P(^{44}N_2O)$ are the probabilities of forming each isotopocule, $^{15}F_1$ is the atom fraction of $^{15}N$ in substrate 1, $^{15}F_2$ is the atom fraction of $^{15}N$ in substrate 2, and $f$ is the proportion of $N^{\alpha}$ derived from substrate 1; $1 - f$ is the proportion of $N^{\alpha}$ derived from substrate 2. Assuming a 1:1 pairing of substrates 1 and 2, $f$ also represents the proportion of $N^{\beta}$ derived from substrate 2, and $1 - f$ represents the proportion of $N^{\beta}$ derived from substrate 1. Processes that derive both nitrogen atoms from the same substrate pool are a special case of eqns. (9-12), where $^{15}F_1 = {}^{15}F_2$. Measuring bulk $^{45}N_2O$ production instead of individual isotopomers (Trimmer et al. 2016) is also a special case of eqns. (9-12), where $P(^{45}N_2O) = P(^{45}N_2O^{\alpha}) + P(^{45}N_2O^{\beta})$ and $f$ cancels out.

To represent each $N_2O$-producing $J$ term in the model, the rates of $N_2O$ production were modeled as second-order:

$$J_i = k_i [substrate_1][substrate_2] \tag{13}$$

where $J_i$ is the rate of $N_2O$ production process $i$ in nM N/day, $k_i$ is a second-order rate constant for that process, $[substrate_1]$ is the concentration of substrate 1 for process $i$, and $[substrate_2]$ is the concentration of substrate 2 for process $i$. Each rate constant $k_i$ was optimized in the model for each station and depth. Again, $N_2O$ production processes that draw both nitrogen atoms from the same substrate are a special case, where $[substrate_1] = [substrate_2]$. $J$ was multiplied by ½ to convert the rate



from nM N/day to nM $N_2O$/day, which was then multiplied by eqns. (9-12) to obtain the rates of production of each isotopocule. For example, the rate of hybrid $^{46}N_2O$ production was represented as:

$$J_{hybrid}^{46N2O} = \frac{1}{2}\left(k_{hybrid}[NH_4^+][NO_2^-]\right)\left(^{15}F_{NH_4^+}\right)\left(^{15}F_{NO_2^-}\right) \tag{14}$$

where $J_{hybrid}^{46N2O}$ is the rate of $^{46}N_2O$ production via hybrid production in nM $N_2O$/day. $N_2O$ consumption was modeled as first-order to the concentration of each isotopocule, based on the $[O_2]$-corrected rates of $N_2O$ consumption measured on the same cruise (Sun et al. 2021b).

To relate the $J$ terms to the substrate pools ($NH_4^+$, $NO_2^-$, and $NO_3^-$), $J$ draws upon the $^{15}N$ and $^{14}N$ substrate pools according
to the atom fractions of $^{15}N$ in each substrate:

$$J_i^{15} = J_i \cdot {}^{15}F_{substrate} \ and \ J_i^{14} = J_i \cdot \left(1 - {}^{15}F_{substrate}\right) \tag{15}$$

where $J_i^{15}$ and $J_i^{14}$ are the rates of consumption of the $^{15}N$ and $^{14}N$ substrate pools by $N_2O$ producing process $i$, $J_i$ is the rate in nM N/day calculated in eqn. (13) for $N_2O$ production process $i$, and $^{15}F_{substrate}$ is the atom fraction of $^{15}N$ in the given substrate pool ($NH_4^+$, $NO_2^-$, and $NO_3^-$). Essentially, eqn. (15) relates how each rate $J_i$ draws from the $^{15}N$ and $^{14}N$ substrate pools, while eqns. (9-12) determine the $^{15}N$ and $^{14}N$ distribution in the product $N_2O$. For example, the rate of $^{15}NH_4^+$
consumption by hybrid $N_2O$ production was represented as:

$$J_{hybrid}^{15NH4+} = \left(k_{hybrid}[NH_4^+][NO_2^-]\right)\left(^{15}F_{NH_4^+}\right) \tag{16}$$

Rates of $^{15}N$ and $^{14}N$ exchange between substrate pools via $NH_3$ oxidation, $NO_2^-$ oxidation, and $NO_3^-$ reduction were also included in the model. Unlike the $N_2O$ production rates, which were optimized for, $NH_4^+$ oxidation, $NO_2^-$ oxidation, and $NO_3^-$ reduction were constrained by the rates calculated in Sect. 2.5, eqn. (7) (Table S2). These rates were represented in the model as first-order:

$$J^{15} = \frac{k}{\alpha}\left[^{15}N\right] \ and \ J^{14} = k\left[^{15}N\right] \tag{17}$$

Where $J^{15}$ and $J^{14}$ represent the rates of $^{15}N$ and $^{14}N$ transformation via $NH_4^+$ oxidation, $NO_2^-$ oxidation, or $NO_3^-$ reduction, $k$ is a first-order rate constant derived from measured rates, $\alpha$ is a fractionation factor, $[^{15}N]$ is the concentration of the $^{15}N$ species, and $[^{14}N]$ is the concentration of the $^{14}N$ species.

The model was optimized against isotopocule data at each timestep, in each tracer experiment (Fig. S4). The parameters
being optimized (inputs to the cost function) were the $2^{nd}$-order rate constants $k_i$ for $N_2O$ production from solely $NH_4^+$, $N_2O$ production from $NO_2^-$ via denitrification or nitrifier-denitrification, $N_2O$ production from $NO_3^-$ via denitrification, hybrid $N_2O$ production using extracellular $NO_2^-$, and $f$ (Fig. 2). In the model, these are all separate processes that operate





independently. The model was optimized using the Nelder-Mead Simplex algorithm (Nelder and Mead 1965), , implemented
in the Scipy optimization library (Virtanen et al. 2020), which has been used successfully for natural abundance $N_2O$

isotopocule models (Monreal et al. 2022). Model error was estimated by optimizing the model at each station and depth with
100 combinations of model parameters, randomly varying the initial concentrations of each $^{15}N$ and $^{14}N$ substrate and rate
constants for $NH_4^+$ oxidation, $NO_2^-$ oxidation, and $NO_3^-$ oxidation by up to 25%.

To ground truth the model, rates of $N_2O$ production obtained from the model were compared to the measured net rates of

$^{46}N_2O$ production (Fig. S5). For processes drawing both nitrogen atoms from the same substrate pool (i.e., not hybrid
production), the modeled rates of $N_2O$ production from each substrate should correspond roughly to the net rate of $^{46}N_2O$
production from the same $^{15}N$-labeled substrate. Higher modeled rates of $N_2O$ production from solely $NH_4^+$ corresponded
generally to higher net rates of $^{46}N_2O$ production from $^{15}N$-$NH_4^+$ (Fig. S5a). Since the model cannot produce negative rates,
negative net rates of $^{46}N_2O$ production from $^{15}N$-$NH_4$ corresponded to modeled $N_2O$ production rates equal to zero (Fig. S5a).

Modeled rates of $N_2O$ production from $NO_2^-$ and $NO_3^-$ via denitrification also corresponded to higher measured rates of
$^{46}N_2O$ production from $^{15}N$-$NO_2^-$ and $^{15}N$-$NO_3^-$, respectively (Fig. S5b, c).

## 3 Results

### 3.1 Depth distributions of oxygen, nitrite, and nitrous oxide

Station PS1, which was at the edge of the ODZ, represented a "background" station with no secondary $NO_2^-$ maximum and a

less pronounced minimum in $[N_2O]$ below the oxycline (Fig. S6; Kelly et al., 2021). At station PS1, the oxic-anoxic interface
— defined in this study as the depth just above the ODZ — occurred at the base of the mixed layer, at 100 m depth (Fig. S6).
Station PS2 was near the geographic center of the oxygen-deficient region and had a secondary $NO_2^-$ maximum of 2.2 µM,
indicating functional anoxia (Fig. S6). The oxic-anoxic interface at Station PS2 occurred at 92 m depth (Fig. S6). Below the
oxic-anoxic interface, $[N_2O]$ declined to 4.5±0.3 nM before increasing again at the base of the secondary $NO_2^-$ maximum and

reaching a local maximum around 800 m depth. Station PS3 was approximately 12 miles from the coast of Mexico and had a
shallow oxic-anoxic interface that moved up and down on timescales of days: on April 10[th], the oxic-anoxic interface
occurred at 40 m depth; two days later, the oxic-anoxic interface had deepened to 62 m depth. Experiments were performed
at the oxic-anoxic interface on both days and are designated with abbreviations "Interface" and "Interface2" in the
experimental metadata (Table S1). The chemical profiles from April 11[th] (Fig. S6), on which the near-surface $[N_2O]$

maximum occurred at 61 m (Kelly et al. 2021), are displayed along with the rate data in this study. Station PS3 had a
pronounced secondary $NO_2^-$ maximum of 2.8 µM at 161 m depth (Fig. S6) and an $NH_4^+$ maximum of 400 nM at 15 m depth
(not shown). On April 11[th], $[N_2O]$ reached a maximum of 195±13 nM at the oxic-anoxic interface and declined below this
depth. Below 600 m depth, $[N_2O]$ began to increase again to 44±3 nM. At every station, a deep, secondary chlorophyll *a*
maximum was observed near the oxic-anoxic interface, where photosynthetically active radiation was much reduced and



[$NO_3^-$] was abundant (Travis et al. 2023). This secondary chlorophyll *a* maximum tended to develop between the depths of the oxic-anoxic interface and secondary $NO_2^-$ maximum (Travis et al. 2023).

### 3.2 Nitrification and nitrate reduction rates

$NH_3$ oxidation to $NO_2^-$ occurred at small, but significant rates ranging from 0.19±0.0004 nM N/day to 4.68±0.07 nM N/day
(Table S2). At every station, rates of $NH_3$ oxidation peaked near the base of the mixed layer, at the same depth as the near-surface [$N_2O$] maximum (Fig. 3a, d, g). At station PS2, $NH_3$ oxidation showed a secondary peak at the same depth as the deep [$N_2O$] maximum (Fig. 3d). At station PS3, there was also a small, significant rate of $NH_3$ oxidation (0.303±0.005 nM N/day) just above the sediment-water interface (Fig. 3g). Rates of $NH_3$ oxidation were undetectable in oxygen deficient waters (Fig. 3a, d, g).


$NO_2^-$ oxidation occurred at higher rates than $NH_4^+$ oxidation, ranging from 13.05±0.08 nM N/day to 465±86 nM N/day (Table S2). The highest rates of $NO_2^-$ oxidation occurred within apparently oxygen deficient waters, at 81.0±0.2 nM N/day in the secondary chlorophyll *a* maximum at station PS2 and at 465±86 nM N/day in the secondary $NO_2^-$ maximum at station PS3 (Fig. 3e, h; Table S2). Note that these are potential rates, since the $^{15}N$ addition was generally much greater than the
ambient concentration (Lipschultz 2008). In some cases, $NO_2^-$ oxidation rates appeared negative due to a decrease in $^{15}N$-$NO_3^-$ vs. incubation time (Fig. 3b, h), but we chose not to left-censor the data.

$NO_3^-$ reduction to $NO_2^-$ occurred at rates ranging from 0.54±0.04 to 33.2±0.1 nM N/day (Table S2). The highest rates of $NO_3^-$ reduction to $NO_2^-$ occurred in the deep, anoxic waters at station PS2 (33.24±0.01 nM N/day; Fig. 3f) and in the
secondary chlorophyll maximum at station PS3 (19.2±0.1 nM N/day; Fig. 3i). There was also a small, significant rate of $NO_3^-$ reduction to $NO_2^-$ in apparently aerobic waters near the surface at station PS1 (Fig. 3c).





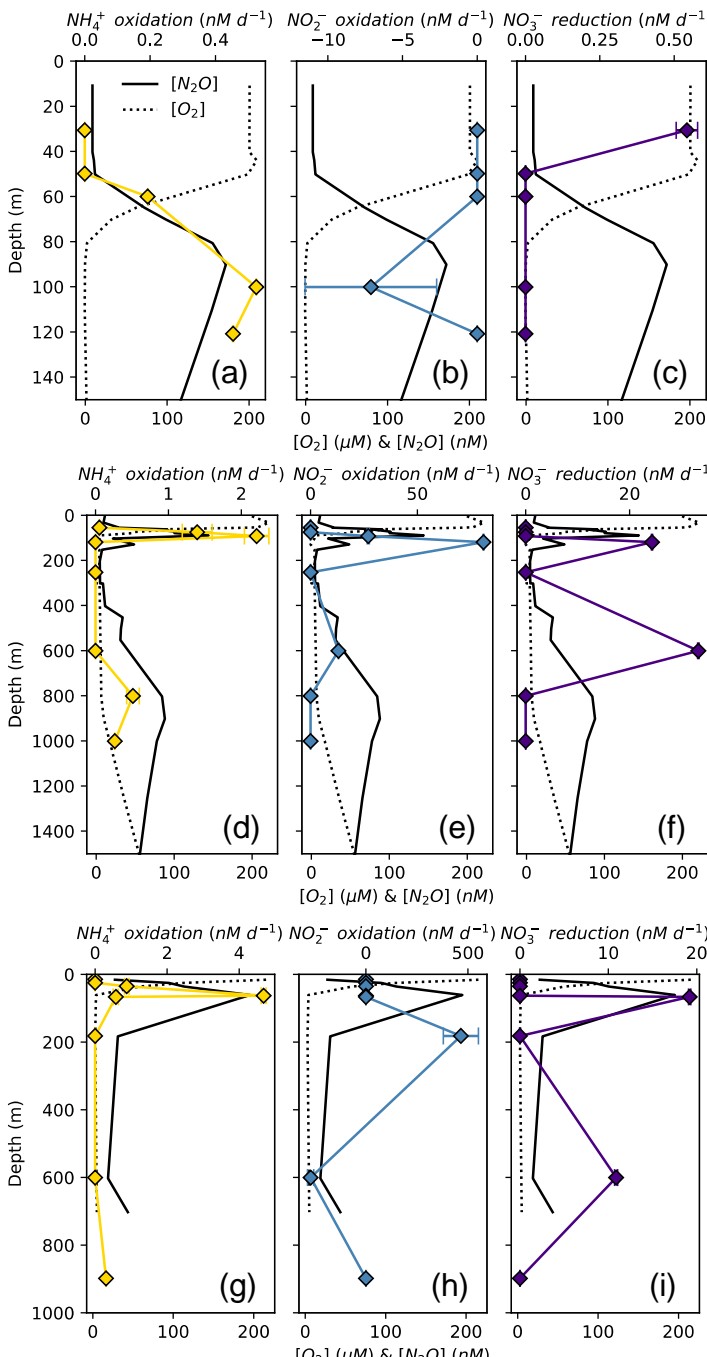

**Figure 3. Rates of NH$_4^+$ oxidation to NO$_2^-$ + NO$_3^-$ (a, d, g, yellow), NO$_2^-$ oxidation to NO$_3^-$ (b, e, h, blue), and NO$_3^-$ reduction to NO$_2^-$ (c, f, i, indigo) at stations PS1 (a-c), PS2 (d-f), and PS3 (g-i). Rates are plotted over depth profiles of dissolved [O$_2$] (dashed lines) and [N$_2$O] (solid lines, from Kelly et al., 2021). Error bars represent rate error, calculated from the error of the slope of product $^{15}$N vs. time. Note the different x-axis scales for rate measurements (top) and [O$_2$] and [N$_2$O] (bottom).**




### 3.3 Net production rates of $^{45}N_2O^\alpha$, $^{45}N_2O^\beta$, and $^{46}N_2O$ (measured net rates)

At each station, the observed rates of net $^{46}N_2O$ (Fig. S7), $^{45}N_2O^\alpha$ and $^{45}N_2O^\beta$ (Fig. S8) production from $^{15}N-NH_4^+$, $^{15}N-NO_2^-$, and $^{15}N-NO_3^-$ all peaked at or just below the oxic-anoxic interface, where the near surface [$N_2O$] maximum was found.

There were also high rates of net $^{46}N_2O$ production from $^{15}N-NO_2^-$ and $^{15}N-NO_3^-$ within the secondary $NO_2^-$ maximum (253 m) at station PS2 (Fig S6, S7e-f). High rates of net $^{45}N_2O^\alpha$ and $^{45}N_2O^\beta$ production also occurred in the secondary $NO_2^-$ maximum at stations PS2 (253 m; Fig. S6, S8e-f) and PS3 (182 m; Fig. S6, S8h-i). The net rates of $^{45}N_2O^\alpha$ and $^{45}N_2O^\beta$ production varied in concert at almost every station and depth, with a few exceptions (Fig. S8).

For example, in the secondary $NO_2^-$ maximum (182 m) at station PS3, in the $^{15}N-NO_2^-$ experiment, the production of $^{45}N_2O^\alpha$ was $0.06\pm0.03$ nM $N_2O$/day ($p = 0.09$) and there was no significant production of $^{45}N_2O^\beta$ (Fig. S8h). In the parallel $^{15}N-NH_4^+$ experiment, the production of $^{45}N_2O^\beta$ was $0.0007\pm0.003$ nM $N_2O$/day ($p = 0.06$) and there was no significant production of $^{45}N_2O^\alpha$. At this station and depth, $f$ was equal to $0.9\pm0.2$ (Table S3). The second experiment in which labeling was unequal occurred at the oxic-anoxic interface (92 m) at station PS2, where in the $^{15}N-NH_4^+$ experiment, the production of $^{45}N_2O^\alpha$ was

$0.005\pm0.002$ nM $N_2O$/day ($p = 0.02$) and there was no significant production of $^{45}N_2O^\beta$ (Fig. S8d). Here, $f$ was equal to $0.2\pm0.1$. Finally, at the mid-oxycline depth (25 m) at station PS3, in the $^{15}N-NH_4^+$ experiment, the production of $^{45}N_2O^\alpha$ was $0.00023\pm0.00008$ nM $N_2O$/day ($p = 0.02$) and there was no significant production of $^{45}N_2O^\beta$. Here, $f$ was statistically indistinguishable from 0.

At many stations and depths, the net production of $^{45}N_2O^\alpha$ and $^{45}N_2O^\beta$ exceeded the values expected for a process that draws both nitrogen atoms from the same substrate pool (Fig. S9). This expected value is calculated from the atom fraction of $^{15}N$ in the substrate and a binomial distribution of the isotopocules of $N_2O$ during $N_2O$ production (Trimmer et al. 2016):

$$p_{expected}^{45} = \frac{p^{46}}{(AF)^2} 2(AF)(1-AF) = \frac{p^{46}}{AF} 2(1-AF) \qquad (18)$$

where $p_{expected}^{45}$ is the expected production of $^{45}N_2O^\alpha$ and $^{45}N_2O^\beta$ from a process that draws both nitrogen atoms from the same substrate pool, $p^{46}$ is the net production rate of $^{46}N_2O$, and AF is the atom fraction of $^{15}N$ in the substrate pool (for

example, $NO_2^-$ in a $^{15}N-NO_2^-$ experiment). Then, excess production of $^{45}N_2O$ is any $^{45}N_2O$ production above and beyond this expected rate:

$$p_{excess}^{45} = p^{45} - p_{expected}^{45} = p^{45} - \frac{p^{46}}{AF} 2(1-AF) \qquad (19)$$

where $p_{excess}^{45}$ is excess production of $^{45}N_2O$ above and beyond that expected for a process drawing both nitrogen atoms from the same pool and $p^{45}$ is the measured net production of $^{45}N_2O$. The equations for $^{45}N_2O^\alpha$ and $^{45}N_2O^\beta$ are the same as eqn.



(19), except for the factor of 2. In many of the $^{15}$N-NH$_4^+$ experiments, there was significant excess $^{45}$N$_2$O$^\alpha$ and $^{45}$N$_2$O$^\beta$ production (Fig. S9a). Similarly, there was significant excess $^{45}$N$_2$O$^\alpha$ and $^{45}$N$_2$O$^\beta$ production in many of the $^{15}$N-NO$_2^-$ experiments, although this was harder to discern due to the wider range of atom fractions in these experiments (Fig. S9b). In a few experiments, excess $^{45}$N$_2$O$^\alpha$ and $^{45}$N$_2$O$^\beta$ production diverged.

### 3.4 N$_2$O production rates and yields (model results)

Based on model results, the rates of N$_2$O production from NO$_3^-$ (denitrification using cellular NO$_2^-$, Fig. 2) were the highest among the N$_2$O production processes measured in this study. In suboxic to anoxic depths, the rates of N$_2$O production from NO$_3^-$ were orders of magnitude higher than all the other N$_2$O production rates (Fig. 4). N$_2$O production from NO$_3^-$ ranged from $6\pm3$ x10$^{-4}$ nM N$_2$O/day to $1.6\pm0.4$ nM N$_2$O/day (Table S3) and peaked at the same depth as the near surface [N$_2$O] maximum at every station (Fig. 4c, g, k), where there were also high rates of NO$_3^-$ reduction to NO$_2^-$ at stations PS2 and PS3 (Fig. 4g, k). N$_2$O production from NO$_2^-$ (denitrification using extracellular NO$_2^-$, Fig. 2) exhibited lower rates, ranging from $5.2\pm0.3$ x10$^{-5}$ nM N$_2$O/day to $0.51\pm0.04$ nM N$_2$O/day (Table S3). At stations PS1 and PS3, N$_2$O production from NO$_2^-$ peaked at the same depth as the near surface [N$_2$O] maximum (Fig. 4b, j); at station PS2, N$_2$O production from NO$_2^-$ exhibited high rates in the near surface [N$_2$O] maximum but peaked in the secondary NO$_2^-$ maximum (253 m, Fig. 4f). N$_2$O production from solely NH$_4^+$ occurred at the smallest rates overall, ranging from $1.18\pm0.25$ x10$^{-5}$ nM N$_2$O/day to $8.1\pm2.1$ x10$^{-3}$ nM N$_2$O/day (Table S3). N$_2$O production from solely NH$_4^+$ peaked around the near-surface [N$_2$O] maximum at each station (Fig. 4a, e, i), as well as in the secondary NO$_2^-$ maximum at station PS2 (Fig. 4e).

Hybrid N$_2$O production occurred at a similar rate as N$_2$O production from NO$_2^-$, ranging from $1.9\pm0.3$ x10$^{-4}$ nM N$_2$O/day to $0.23\pm0.08$ nM N$_2$O/day. Hybrid N$_2$O production peaked within the near surface [N$_2$O] maximum at all stations (Fig. 4d, h, l). At station PS2, hybrid N$_2$O production exhibited the highest rates at the same depths as NH$_4^+$ oxidation, with a secondary peak in the deep [N$_2$O] maximum (Fig. 4h). At station PS3, hybrid N$_2$O production, like NH$_4^+$ oxidation, exhibited a small, significant rate just above the sediment-water interface (Fig. 4l).





**Figure 4.** N₂O production from solely NH₄⁺ (a, e, i, yellow diamonds), N₂O production from NO₂⁻ (b, f, j, blue diamonds), N₂O production from NO₃⁻ (c, g, k, indigo diamonds), and hybrid N₂O production (d, h, l, green diamonds) at stations PS1 (a-d), PS2 (e-h), and PS3 (i-l). Panels a, e, and i show depth profiles of dissolved [O₂] (dashed lines) and [N₂O] (solid lines, from Kelly et al., 2021). Panels c, g, and k show rates of NO₃⁻ reduction to NO₂⁻ (gray circles). Panels d, h, and l show rates of NH₃ oxidation (gray circles). N₂O production rate error bars are calculated from 100 model optimizations, varying key parameters by up to 25%. Note the different x-axis scales for N₂O production (top), [O₂] and [N₂O] (a, e, i, bottom), NO₃⁻ reduction to NO₂⁻ (c, g, k, bottom), and NH₃ oxidation (d, h, l, bottom).





The percentage of $N_2O$ production from $NH_4^+$ comprised by hybrid $N_2O$ was calculated as:

$$\% \, hybrid = \frac{hybrid \, N_2O \left( \frac{nM \, N_2O}{day} \right)}{N_2O \, from \, hydroxylamine \left( \frac{nM \, N_2O}{day} \right) + hybrid \, N_2O \left( \frac{nM \, N_2O}{day} \right)} \quad (20)$$

On average, hybrid $N_2O$ production was $86\pm28\%$ of $N_2O$ production from $NH_4^+$. Hybrid $N_2O$ production was $> 75\%$ of the total $N_2O$ production from $NH_4^+$ at all stations and depths except for the top of the oxycline at station PS1 (Fig. 5a), the middle of the oxycline at station PS2 (Fig. 5b), and the top of the oxycline at station PS3 (Fig. 5c), where it comprised 0%, 68%, and 19% of $N_2O$ production from $NH_4^+$, respectively. Hybrid production as a percentage of total $N_2O$ production from

$NH_4^+$ declined with increasing dissolved oxygen (Fig. S10), although more measurements are needed to fully evaluate this trend.

The percentage of hybrid $N_2O$ production as a proportion of total $N_2O$ production was more variable and tended to decline with decreasing dissolved oxygen as production from $NO_3^-$ increased (Fig. 5). Hybrid $N_2O$ production was greater than 75%

of total $N_2O$ production only at the surface at station PS1 (Fig. 5a), the top of the oxycline and deep $[N_2O]$ maximum at station PS2 (Fig. 5b), and the deep $[N_2O]$ maximum at station PS3 (Fig. 5c).

$N_2O$ production from $NO_3^-$ comprised a much greater proportion of total $N_2O$ production overall (Fig. 5). In the near-surface $[N_2O]$ maximum at station PS1, $N_2O$ production was predominantly (95.4%) from $NO_3^-$, with smaller contributions from

hybrid production (4.0%) and denitrification from $NO_2^-$ (0.6%; Fig. 5a). In the near-surface $[N_2O]$ maximum at station PS2, $N_2O$ was produced 60.2% from $NO_3^-$, 32.1% from hybrid production, 7.3% from $NO_2^-$, and 0.4% from solely $NH_4^+$ (Fig. 5b). In the near-surface $[N_2O]$ maximum at station PS3, $N_2O$ production was 87.0% from $NO_3^-$, 12.4% from hybrid production, 0.5% from $NO_2^-$, and 0.1% from solely $NH_4^+$ (Fig. 5c).





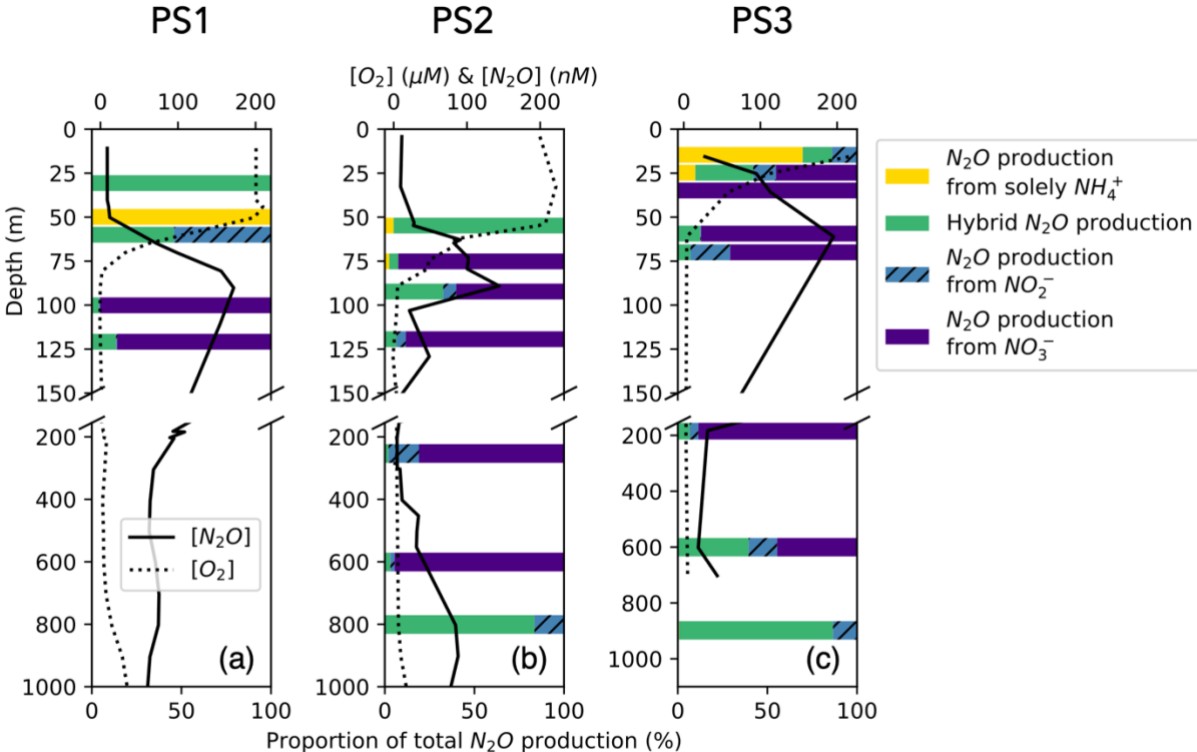

**Figure 5. N₂O production from solely NH₄⁺ (yellow bars), hybrid N₂O production (green bars), N₂O production from NO₂⁻ (blue hatched bars), and N₂O production from NO₃⁻ (indigo bars) as proportions of total N₂O production station PS1 (a), PS2 (b), and PS3 (c). Data are plotted over depth profiles of dissolved [O₂] (dashed lines) and [N₂O] (solid lines, from Kelly et al., 2021). Note broken y-axes and different x-axis scales for [O₂] and [N₂O] (top) and proportions (bottom).**

## 3.5 Oxygen dependence of N₂O production

The oxygen dependencies of N₂O production were determined by fitting model derived N₂O production vs. [O₂] using the following rate law:

$$rate = ae^{-b[O_2]} \tag{21}$$

In this analysis, both ambient [O₂] measured by the Sea-Bird sensor mounted on the rosette ("ambient [O₂]") and [O₂] measured by chemiluminescent optodes mounted inside incubation bottles ("incubation [O₂]") were examined (Fig. 6). The rate dependencies on ambient and incubation [O₂] reflect both preconditioning (i.e., the ambient [O₂] in which the microbial community was living before the incubation experiment), and response to perturbation (i.e., the experimental conditions inside the incubation bottles, if different from the environment). In particular, those incubations that had higher incubation [O₂] than the ambient [O₂], had received small oxygen perturbations.



Hybrid N$_2$O production rates decreased exponentially with increasing dissolved [O$_2$] (Fig. 6). Fitting hybrid rates vs. ambient

[O$_2$] produced a rate equation (19) with $a = 0.066$ and $b = 0.17$ (Fig. 6a); hybrid rates vs. incubation [O$_2$] produced fits with

$a = 0.076$ and $b = 0.067$ (Fig. 6b).

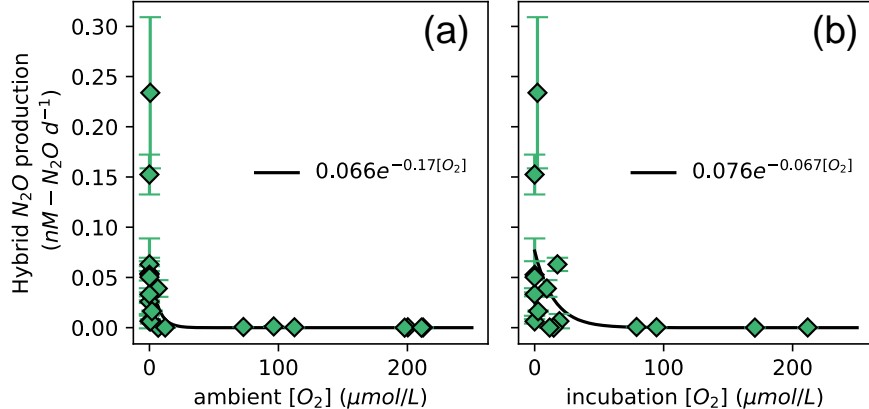

**Figure 6. Hybrid N$_2$O production rates (green diamonds), along a range of ambient [O$_2$] measured by a Seabird sensor for the Niskin bottles from which samples were taken (a) and [O$_2$] measured by chemiluminescent optodes mounted inside sample bottles**
**(b). Error bars are calculated from 100 model optimizations, varying key parameters by up to 25%. Curves of form rate =**
$\boldsymbol{ae^{-b[O_2]}}$ **are fit through the data (black lines); values of $a$ and $b$ are shown in white boxes in each plot.**

The rate of N$_2$O production from solely NH$_4^+$ also decreased exponentially with increasing dissolved [O$_2$]. The highest rates

of N$_2$O production from solely NH$_4^+$ occurred in the secondary chlorophyll maximum at station PS3 (Table S3), where

dissolved oxygen was below detection. N$_2$O yield during production from solely NH$_4^+$ also exhibited exponentially

decreasing relationships with dissolved [O$_2$] (Fig. 7). To ensure mass balance in terms of NH$_4^+$ consumption (Fig. S11), N$_2$O

yield (%) during production from solely NH$_4^+$ was calculated as:

$$yield\ (\%) = \frac{2\left[N_2O\ from\ solely\ NH_4^+\ \left(^{nM\ N_2O}/_{day}\right)\right]}{2\left[N_2O\ from\ solely\ NH_4^+\ \left(^{nM\ N_2O}/_{day}\right)\right] + hybrid\ N_2O\ \left(^{nM\ N_2O}/_{day}\right) + NH_3\ oxidation\ \left(^{nM\ N}/_{day}\right)} \quad (22)$$

where N$_2$O production from solely NH$_4^+$ is in units of nM N$_2$O/day, hybrid N$_2$O production is in units of nM N$_2$O/day, and

NH$_3$ oxidation to NO$_2^-$ is in units of nM N/day. This assumes that the formation of N$_2$O from solely NH$_4^+$ draws two

nitrogen atoms from the NH$_4^+$ pool, while hybrid N$_2$O production and the oxidation of NH$_4^+$ to NO$_2^-$ each draw one atom

from the NH$_4^+$ pool (Fig. S11). Following the same convention, N$_2$O yield (%) during hybrid production was calculated as:

$$yield\ (\%) = \frac{hybrid\ N_2O\ \left(^{nM\ N_2O}/_{day}\right)}{2\left[N_2O\ from\ solely\ NH_4^+\ \left(^{nM\ N_2O}/_{day}\right)\right] + hybrid\ N_2O\ \left(^{nM\ N_2O}/_{day}\right) + NH_3\ oxidation\ \left(^{nM\ N}/_{day}\right)} \quad (23)$$

The maximum N$_2$O yield during production from solely NH$_4^+$ was 2.2±0.7% (Fig. 7a, b), while the maximum N$_2$O yield

from hybrid production was 21±7% (Fig. 7c, d). N$_2$O yield during production from solely NH$_4^+$ declined more sharply with



increased $O_2$ than $N_2O$ yield during hybrid production (Fig. 7). $N_2O$ production via denitrification also exhibited an
exponential relationship with dissolved $O_2$, where $N_2O$ production from $NO_2^-$ was more inhibited by dissolved $O_2$ than $N_2O$
production from $NO_3^-$ (Fig. 7e-h).

When looking at the oxygen dependence of denitrification, we found several non-zero rates of $N_2O$ production from $NO_3^-$
via denitrification with dissolved $[O_2]$ greater than 3 µM. For example, at the oxic-anoxic interface at station PS2, where
ambient $[O_2]$ was 6.49 µM and incubation $[O_2]$ was 6.29±0.07 µM (Table S1), $N_2O$ production from $NO_3^-$ was 0.073±0.011
nM $N_2O$/day (Fig. 4g, Table S3). $N_2O$ production from $NO_2^-$ at the same station and depth was 0.0089±0.0002 nM $N_2O$/day
(Fig. 4f, Table S3). Similarly, at the oxic-anoxic interface of station PS3, where ambient $[O_2]$ was 12.48 µM and incubation
$[O_2]$ was 6.64±0.03 µM (Table S1), $N_2O$ production from $NO_3^-$ was 0.12±0.02 nM $N_2O$/day (Fig. 4k, Table S3). There were
also two anoxic depths at station PS2 that were not sparged with He before tracer addition ("base of ODZ" and "deep ODZ
core"), where ambient $[O_2]$ was below detection but incubation $[O_2]$ was significantly elevated (17.7±0.1 µM and 19.2±0.8
µM, respectively; Table S1). At these depths, $N_2O$ production from $NO_2^-$ was 0.012±0.001 nM $N_2O$/day and 0.0052±0.0003
nM $N_2O$/day, respectively (Fig. 4f, Table S3). $N_2O$ production from $NO_3^-$ at the "deep ODZ core" depth was 0.21±0.04 nM
$N_2O$/day (Fig. 4g, Table S3).




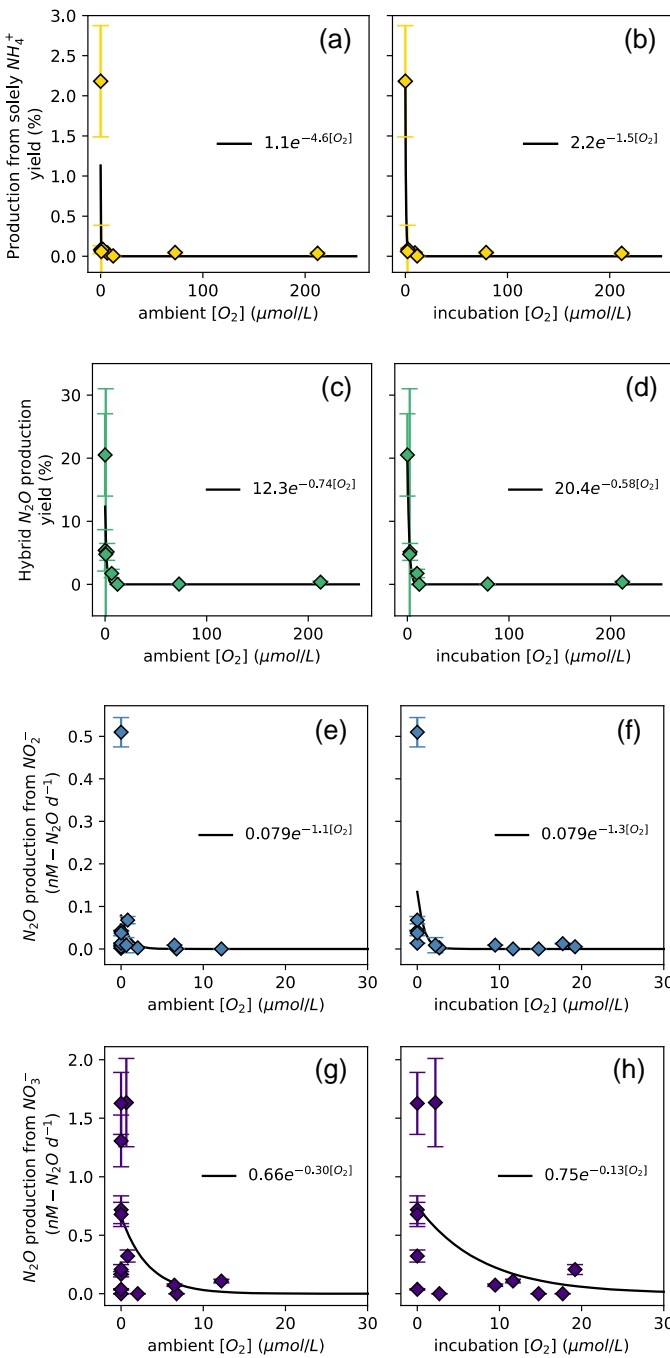

**Figure 7. N₂O yield (%) during hydroxylamine oxidation (a, b), N₂O yield (%) during hybrid production (c, d), N₂O production from NO₂⁻ via denitrification (e, f), and N₂O production from NO₃⁻ via denitrification (g, h), measured at a range of [O₂]. Yields are plotted along a range of ambient [O₂] measured by a Seabird sensor (a, c, e, g) and [O₂] measured by chemiluminescent optodes mounted inside sample bottles (b, d, f, h). Yields are only calculated at stations and depths where rates of NH₃ oxidation are greater than 0. Curves of form yield $= ae^{-O_2 b}$ are fit through the data (black lines); values of $a$ and $b$ are shown in white boxes in each plot.**



## 4 Discussion

### 4.1 Rates of nitrification and $N_2O$ production from solely $NH_4^+$

The rates of $N_2O$ production from $NH_4^+$ in this study — including both hybrid $N_2O$ production and $N_2O$ production from solely $NH_4^+$ — ranged from $1.2 \times 10^{-5}$ to 0.23 nM $N_2O$/day. These were similar to those measured on the same cruise by Frey et al. (2023), who measured rates of $N_2O$ production from $NH_4^+$ in the oxycline of 0.028 – 0.149 nM $N_2O$/day (Frey et al. 2023). The low rates of $NH_3$ oxidation to $NO_2^-$ in this study (0.05 – 4.68 nM N/day) were also similar to those measured by Frey et al. (2023), who measured $NH_3$ oxidation of 1.0 – 11.7 nM/day in the oxycline. $NH_3$ oxidation rates in this study were

smaller than those measured on the same cruise by Travis et al. (2023), who measured $NH_3$ oxidation rates as high as 48.7 nM/day at station PS3. The highest rates of $NO_2^-$ oxidation occurred in anoxic depths at stations PS2 and PS3 (Fig. 3e, h), which agrees with mounting evidence suggesting the importance of $NO_2^-$ oxidation in apparently anoxic regions (Sun et al. 2017, 2021a).

When [$O_2$] was less than 10 µM, the rates of hybrid $N_2O$ production (0.006 – 0.23 nM $N_2O$/day) were orders of magnitude greater than the rates of $N_2O$ production from solely $NH_4^+$ at the same depths (0 – 0.016 nM $N_2O$/day) (Fig. 4). Indeed, at the upper oxic-anoxic interface, the rates of hybrid $N_2O$ production were on a similar order of magnitude to $N_2O$ production from $NO_2^-$ via denitrification (0.008 – 0.51 nM $N_2O$/day). These results agree with previous work showing that hybrid $N_2O$ formation represents a high percentage of total $N_2O$ production from $NH_4^+$ in the ETNP and eastern tropical South Pacific

(ETSP) (Frey et al. 2020, 2023). The results in this study also agree with recent culture work: the $^{15}N$-$NH_4^+$ experiments in this study fell along a range of [$^{15}N$-$NH_4^+$]/[$NO_2^-$] of 0.14-0.5, in which Wan et al. (2023) found that hybrid $N_2O$ production occurred at a rate two to four times greater than $N_2O$ production via hydroxylamine oxidation in cultures of *Nitrosopumilus maritimus*.

We found three depths near the surface where hybrid production comprised a smaller percentage (0, 68, and 19%) of total $N_2O$ production from $NH_4^+$ (Fig. 5a-c). While previous work in the ETNP found that hybrid $N_2O$ production always comprised > 90% of $N_2O$ production from $NH_4^+$ (Frey et al. 2023), it is possible that we sampled different conditions than those reported previously, which may account for this discrepancy. We also found that hybrid $N_2O$ formation generally comprised a small proportion of total $N_2O$ production, which was dominated by $N_2O$ production from $NO_3^-$, especially at

suboxic depths (Fig. 5d-h). This is similar to previous findings from the ETSP, which showed that hybrid formation comprised 0 – 95% of total $N_2O$ production from $NO_2^-$ along the natural [$O_2$] gradient (Frey et al. 2020). This large range is due to the large range of rates of $N_2O$ production from $NO_2^-$, which can occur at orders of magnitude higher or lower than hybrid $N_2O$ production.



## 4.2 Pathways of hybrid $N_2O$ production and implications for hybrid $\delta(^{15}N^{sp})$

Hybrid $N_2O$ production peaked in the same depths as $NH_3$ oxidation (Fig. 4d, h, l), which were also the depths at which ammonia-oxidizing archaea were most abundant (Frey et al. 2023), consistent with $N_2O$ production associated with ammonia-oxidizing archaea. At most stations and depths, the production of $^{45}N_2O^\alpha$ and $^{45}N_2O^\beta$ in both the $^{15}N$-$NO_2^-$ and $^{15}N$-$NH_4^+$ experiments were roughly equal. From this we conclude that during hybrid formation, $N^\alpha$ and $N^\beta$ each retained nitrogen atoms derived from both $NH_4^+$ and $NO_2^-$.


Although our data do not allow us to comment directly on the enzymatic machinery of hybrid $N_2O$ formation, our data can be used to distinguish between hypothetical pathways for hybrid $N_2O$ production. Firstly, we see much higher rates of hybrid production using ambient $NO_2^-$ (Pathway 3 in Wan et al., 2023) than hybrid production using cellular $NO_2^-$ (Pathway 2 in Wan et al., 2023). Again, this agrees with the results of Wan et al. (2023), who see higher rates of hybrid formation from

extracellular $NO_2^-$ within the range of [$^{15}N$-$NH_4^+$]/[$NO_2^-$] covered by our experiments. Additionally, if $NO_2^-$ and $NH_4^+$ were both converted to a common intermediate (such as NO) within the cell, which reacts to form $N_2O$ either via an unknown nitric oxide reductase or a chemo-denitrification-like abiotic pathway, the resulting $N_2O$ would likely contain a randomized ratio of $NH_4^+$: $NO_2^-$. If instead, $NH_4^+$ and $NO_2^-$ were combined to form an intermediate such as hyponitrite (HONNOH or $^-$ONNO$^-$ in its deprotonated form), which reacts to form $N_2O$ via breakage of one of the N–O bonds, the resulting $N_2O$ would

likely contain a 1:1 ratio of $NH_4^+$: $NO_2^-$. Our data indicate that hybrid $N_2O$ predominantly involves nitrogen derived from a 1:1 ratio of $NH_4^+$ and $NO_2^-$, and thus supports the second of these scenarios. This result agrees with Stieglmeier et al. (2014), who observe a nearly 1:1 contribution of $NH_4^+$ and $NO_2^-$ to $N_2O$ production in cultures of *Nitrososphaera viennensis*. With a precursor such as hyponitrite, equal formation of $^{45}N_2O^\alpha$ and $^{45}N_2O^\beta$ could be achieved with non-selective N–O bond breakage; unequal formation of $^{45}N_2O^\alpha$ and $^{45}N_2O^\beta$ could be achieved when the N–O bond containing the nitrogen derived

from $NO_2^-$ breaks at a different rate than that containing the nitrogen derived from $NH_4^+$.

Still, there were some depths that did show unequal formation of $^{45}N_2O^\alpha$ and $^{45}N_2O^\beta$: the oxic-anoxic interface at station PS2, the secondary $NO_2^-$ maximum at station PS3, and the mid-oxycline depth at PS3. At these stations and depths, the production $^{45}N_2O^\alpha$ and $^{45}N_2O^\beta$ diverged, and the value of $f$ was significantly different from 0.5. This indicates that $N^\alpha$

retained a different proportion of nitrogen derived from $NO_2^-$ and $NH_4^+$ than $N^\beta$.

Values of $f$, the proportion of $N^\alpha$ derived from $NO_2^-$ during hybrid $N_2O$ production, centered on 0.5 but varied with depth at each station (Table S3). The mean value of $f$ across all stations and depths was 0.5±0.2. A significant relationship ($R^2 = 0.84$, $p = 1.6 \times 10^{-8}$) emerged between $f$ and ambient [$O_2$] (Fig. S12a). A significant relationship ($R^2 = 0.72$, $p = 2.4 \times 10^{-6}$) also

emerged between $f$ and potential density anomaly ($\sigma_\theta$), where $\sigma_\theta$ is the density of a water parcel calculated from *in situ* salinity, potential temperature, and pressure, minus 1000 kg/m$^3$ (Fig. S12b). In these cases, $f$ decreased with decreasing [$O_2$]




and increasing $\sigma_\theta$ — i.e., the proportion of $N^\alpha$ derived from $NO_2^-$ during hybrid $N_2O$ production decreased with decreasing $[O_2]$. Both relationships, however, exhibited a large amount of scatter (Fig. S12). Values of $f$ less than or greater than 0.5 resulted from the unequal production of $^{45}N_2O^\alpha$ and $^{45}N_2O^\beta$ at certain depths.


These findings of unequal $^{45}N_2O$ production have important implications for the natural abundance $\delta(^{15}N^{sp})$ of $N_2O$ produced by the hybrid $N_2O$ process. Assuming that hybrid $N_2O$ production proceeds through a symmetrical intermediate in which $NH_4^+$ and $NO_2^-$ are paired in a 1:1 ratio, we can model $\delta(^{15}N^{sp})$ as:

$$\delta\left(^{15}N^{sp}\right) = \delta\left(^{15}N^\alpha\right) - \delta\left(^{15}N^\beta\right)$$

$$\begin{aligned}
\delta\left(^{15}N^{sp}\right) = &\left[f\delta\left(^{15}N - NO_2^-\right) + (1-f)\delta\left(^{15}N - NH_4^+\right)\right] \\
&- \left[(1-f)\delta\left(^{15}N - NO_2^-\right) + f\delta\left(^{15}N - NH_4^+\right) - \varepsilon\right]
\end{aligned} \tag{24}$$

where $f$ is the proportion of the $\alpha$ nitrogen derived from $NO_2^-$ and the proportion of the $\beta$ nitrogen derived from $NH_4^+$, and $\varepsilon$ is the fractionation factor associated with $N^\beta$–O bond breakage. It is apparent from eqn. (24) that if $f = \frac{1}{2}$, as was the case for *most* experimental depths in this study, hybrid $\delta(^{15}N^{sp})$ should depend only on $\varepsilon$ and not the isotopic composition of each substrate. If, however, $f \neq \frac{1}{2}$, hybrid $\delta(^{15}N^{sp})$ retains a dependence on the $\delta(^{15}N)$ of the substrates — or more accurately, the difference in $\delta(^{15}N)$ of the two substrates; if the $\delta(^{15}N)$ of the substrates is equal, it will cancel out regardless of $f$. If $\delta(^{15}N-$
$NH_4^+) > \delta(^{15}N-NO_2^-)$, as is generally the case in the secondary nitrite maximum (Buchwald et al. 2015; Casciotti 2016), then low values of $f$ should produce high hybrid $\delta(^{15}N^{sp})$, and high values of $f$ should produce low hybrid $\delta(^{15}N^{sp})$ (Fig. 8).

Previous work has identified a $\delta(^{15}N^{sp})$ minimum of 10‰ near the surface at Station ALOHA in the subtropical North Pacific gyre, which cannot be explained with a combination of nitrification and denitrification because the water column is fully
aerobic (Popp et al. 2002). If this minimum is produced by ammonia-oxidizing archaea, however, it appears to conflict with culture experiments indicating that ammonia-oxidizing archaea produce $N_2O$ with $\delta(^{15}N^{sp}) = 19.4‰ - 31.5‰$ (Santoro et al. 2011; Jung et al. 2014). The potential for a variable hybrid $\delta(^{15}N^{sp})$ could explain this apparent conflict between field and culture data. More studies are needed to determine the driving factors behind the proportions of each nitrogen atom in hybrid $N_2O$ derived from each substrate, and thus to determine the conditions under which hybrid $N_2O$ could have a variable
$\delta(^{15}N^{sp})$.



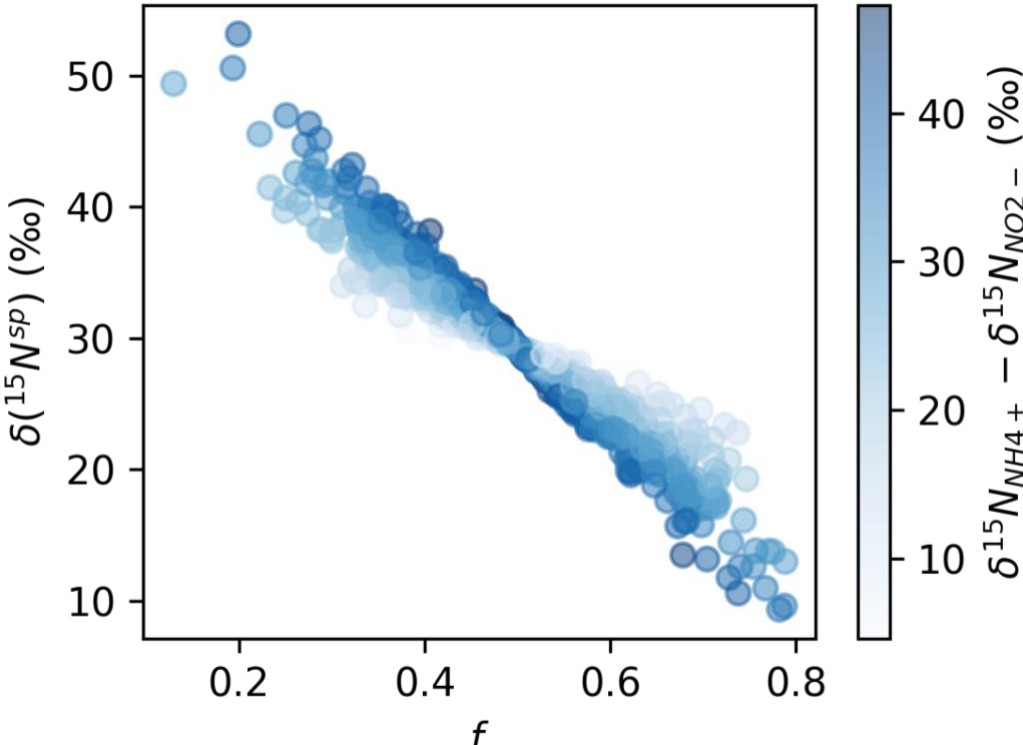

**Figure 8. Simulated values of $\delta(^{15}N^{sp})$ calculated with a range of $f$ (the proportion of $N^{\alpha}$ derived from $NO_2^-$ during hybrid $N_2O$ production) and $\delta(^{15}N\text{-}NH_4^+) - \delta(^{15}N\text{-}NO_2^-)$, assuming ε = 30.3‰ (Santoro et al., 2011). Results are shaded by $\delta(^{15}N\text{-}NH_4^+) - \delta(^{15}N\text{-}NO_2^-)$. When $f$ is less than or greater than ½, there is the potential for site preference to depend on the isotopic compositions of**
**each substrate.**

## 4.3 Rates of $N_2O$ production via denitrification

The high percentage of $N_2O$ production from $NO_3^-$ in the near surface [$N_2O$] maximum supports the conclusion from the natural abundance isotopomer literature that the near surface [$N_2O$] maximum results from a combination of hybrid $N_2O$ production and production from $NO_3^-$, which produces a local minimum in $\delta(^{15}N^{sp})$ (Kelly et al. 2021; Monreal et al. 2022).

Production from $NO_3^-$ also dominated $N_2O$ production within the ODZ. Natural abundance isotopomer work has shown that $N_2O$ production from $NO_3^-$ could be an important source of $N_2O$ in the anoxic core of ODZs if it is allowed to produce $N_2O$ with a positive $\delta(^{15}N^{sp})$ (Casciotti et al. 2018; Kelly et al. 2021; Monreal et al. 2022). Here, the high rates of $N_2O$ production from $NO_3^-$ measured within the ODZ core support this hypothesis. While most denitrifying strains produce $N_2O$ with $\delta(^{15}N^{sp}) \approx 0$‰ (Sutka et al. 2006), at least one strain of denitrifying bacteria can produce $N_2O$ with $\delta(^{15}N^{sp}) = 22$‰ (Toyoda

et al. 2005) and denitrifying fungi produce $N_2O$ with $\delta(^{15}N^{sp}) = 35$–37‰ (Sutka et al. 2008; Rohe et al. 2014; Yang et al. 2014; Lazo-Murphy et al. 2022). High rates of $N_2O$ production from $^{15}N\text{-}NO_3^-$, combined with natural abundance isotopomer studies, suggest that strains of denitrifying bacteria and fungi with characteristics similar to these cultured examples may be important contributors to $N_2O$ in the core of ODZs. The importance of $N_2O$ production from $NO_3^-$ also



presents an important exception to the modular view of the microbial nitrogen cycle network, which holds that intermediates
are passed externally from one cell to the next, rather than being held internally (Kuypers et al. 2018). This process is
currently left out of most biogeochemical models of nitrogen cycling in and around oxygen-deficient zones (Bianchi et al.
2023), and modeling work is needed that includes this as a source of $N_2O$.

### 4.4 Oxygen dependence of $N_2O$ production rates and yields

While this study and others have found that hybrid $N_2O$ production represents a consistent *percentage* of $N_2O$ production
from $NH_4^+$ along a range of ambient $[O_2]$ (Frey et al. 2020, 2023), the *rate* of hybrid $N_2O$ production followed a clear
exponential dependence on dissolved oxygen (Fig. 6). The differences in ambient and incubation $[O_2]$ resulted in slight
differences in the coefficients for each yield curve; nevertheless, hybrid rates plotted along both ambient and incubation $[O_2]$
gradients exhibited remarkably similar $[O_2]$ inhibition curves, with the highest rates at $[O_2] < 7$ µM. These results are similar
to those of Frey et al. (2023), who showed a decrease in $N_2O$ production from $NH_4^+$ with increasing $[O_2]$.


$N_2O$ yields during production from solely $NH_4^+$ also increased with decreasing $[O_2]$ (Fig. 7a,b), as previously reported
(Goreau et al. 1980; Nevison et al. 2003; Ji et al. 2018; Frey et al. 2020). $N_2O$ yields during production from solely $NH_4^+$
increased sharply with decreasing $[O_2]$ along both ambient and incubation $[O_2]$ gradients but were much smaller than the
yields from hybrid $N_2O$ production (Fig. 7c-d). The maximum yields during production from solely $NH_4^+$ were similar to the
maximum yields found by another study in the ETNP, which were around 3% (Frey et al. 2023), and much higher than
yields from ammonia-oxidizing archaea in soils and culture (up to 0.03%) (Hink et al. 2017a; b).

The maximum $N_2O$ yield for hybrid production (21%; Fig. 7c,d) was an order of magnitude higher than $N_2O$ yields during
$NH_4^+$ oxidation that were previously measured in the ETSP and ETNP (Ji et al. 2018). These high yields occurred at the
oxic-anoxic interface at Station PS1 and just below the oxic-anoxic interface at Station PS3, where ambient $[O_2]$ was below
detection but $NH_3$ oxidation still occurred (Fig. 3). This indicates the potential for extremely high yields of $N_2O$ from hybrid
production where $NH_3$ oxidation is active in suboxic to anoxic environments.

$N_2O$ production from $NO_2^-$ and $NO_3^-$ also exhibited exponential dependence on dissolved oxygen, albeit with smaller
maximum rates than those found in the ETSP (Ji et al. 2015; Frey et al. 2020). Most surprising were the significant rates of
$N_2O$ production via denitrification at $[O_2] > 3$ µM (Fig. 7g-h), which has previously been suggested as the threshold above
which denitrification ceases (Dalsgaard et al. 2014). These observations are particularly evident in the plots of $N_2O$
production from $NO_3^-$ vs. incubation $[O_2]$ (Fig. 7h), where positive, significant rates of $N_2O$ production from $NO_3^-$ were
evident at incubation $[O_2]$ as high as $19.2 \pm 0.8$ µM (PS2 Deep ODZ Core experiment). These results showed that $N_2O$
production from $NO_3^-$ can occur in $[O_2]$ as high as $19.2 \pm 0.8$ µM, which is similar to results from the ETSP which showed
$N_2O$ production from $NO_3^-$ in manipulated $[O_2]$ as high as 30 µM (Frey et al. 2020). These results suggest that denitrifying





microbial communities acclimatized to lower ambient [$O_2$] can continue to produce $N_2O$ when [$O_2$] is suddenly increased. The volume of suboxic water in the ocean has been increasing over the last 50 years and will likely continue to expand over the 21st century (Stramma et al. 2008; Schmidtko et al. 2017; Oschlies et al. 2018), although the extent of this deoxygenation

remains uncertain (Cabré et al. 2015; Bianchi et al. 2018; Busecke et al. 2022). Constraining the window of oxygen concentrations under which denitrification leads to $N_2O$ production will be key to understanding how marine deoxygenation and $N_2O$ cycling will interact.

## 4.5 Experimental artifacts

Care was taken to minimize the effects of experimental set-up on the microbial communities in each sample. In addition to

the steps taken to prevent oxygen contamination (described in Sect. 2 Methods), a relatively short 24-hour incubation period was selected to minimize bottle effects and shifts in the microbial community composition over the course of each incubation. Nonetheless, sample collection, purging, and tracer addition likely affected the microbial communities in several ways. First, samples were frequently collected from depths where the water temperature was cooler than that of the lab, and while samples were returned to a cool temperature during incubation (12° C), they were exposed to warmer temperatures

(>20° C) during the two hours in which they underwent collection and manipulation prior to incubation. Likewise, during this interval, samples were exposed to higher light levels before being returned to the dark for incubation. While oxygen contamination was minimized during sample collection, it was not eliminated entirely, and a temporary oxygen intrusion before sparging may have poisoned certain anaerobic processes. The 90-minute sparge also likely removed carbon dioxide in addition to oxygen and $N_2O$, increasing the pH of each sample. Finally, the $NH_4^+$ and $NO_2^-$ tracer and carrier additions

exceeded the ambient concentrations of these substrates, potentially stimulating the rates of processes that rely on these substrates. All of these perturbations, while common among incubation studies, may have affected the microbial community differentially in each sample. Thus, the results presented here represent processes able to withstand these perturbations to ambient environmental conditions. Any abiotic reactions between the $HgCl_2$ preservative and $NO_2^-$ tracer and carrier would have shifted all three timepoints equally, and thus should not introduce a bias into the slopes of $^{15}N$-labeled $N_2O$ with time

and the rates calculated there from.

## 4.6 Alternate sources of $N_2O$

Other processes may have contributed to $N_2O$ production in our samples. A complementary set of experiments found that fungal denitrification comprised 50% of total $N_2O$ production via denitrification in the secondary chlorophyll *a* maximum depths discussed here (Peng and Valentine 2021). It is also possible that algal $N_2O$ production was occurring in our samples,

especially in the secondary chlorophyll *a* maximum, where algae may have been struggling due to the low light levels and thus producing more $N_2O$ (Burlacot et al. 2020). If algal $N_2O$ production were higher near the coast, it may help explain why we observed the highest rates of $N_2O$ production at station PS3, despite no significant increase in ammonia monooxygenase



(*amoA*) gene copies there (Frey et al. 2023). We cannot rule out fungal or algal $N_2O$ production in our samples, so we consider these processes as potential contributors to the bulk denitrifying flux discussed here.

**5. Conclusions**

We applied $N_2O$ isotopocule measurements to $^{15}N$ tracer incubations to measure $N_2O$ production rates and mechanisms in the ETNP. We found that $N_2O$ production rates peaked at the oxic-anoxic interface above the ODZ, with the highest rates of $N_2O$ production from $NO_3^-$. Hybrid $N_2O$ production peaked in both the shallow and deep oxyclines, where $NH_3$ oxidation was also active, and exhibited yields as high as 21%.


Based on variable production of $^{45}N_2O^\alpha$ and $^{45}N_2O^\beta$, as well as the 1:1 ratio in which $NH_4^+$ and $NO_2^-$ most frequently combine to form $N_2O$, we posit a two-step process for this hybrid mechanism, involving an initial bond-forming step that draws nitrogen atoms from each substrate to form a symmetric intermediate, and a second bond-breaking step that breaks the bonds in the symmetric intermediate to form $N_2O$. From this posited process, we infer that hybrid $N_2O$ production by
ammonia-oxidizing archaea may have a variable site preference that depends on the $^{15}N$ content of each substrate. This variable site preference may reconcile field experiments that identify a near-surface source with a site preference of 10‰ (Popp et al. 2002), and culture experiments showing that ammonia-oxidizing archaea produce $N_2O$ with a site preference of 19.4‰ – 31.5‰ (Santoro et al. 2011; Jung et al. 2014).

$N_2O$ production rates and yields of every process examined here were inhibited by dissolved oxygen. $N_2O$ yield via hydroxylamine oxidation was most sensitive to $O_2$, followed by the rates of $N_2O$ production from $NO_2^-$ via denitrification, hybrid $N_2O$ production, and $N_2O$ production from $NO_3^-$ via denitrification. Indeed, we measured positive, significant rates of $N_2O$ production from $NO_3^-$ at ambient $[O_2]$ as high as 12.5 µM, and at manipulated $[O_2]$ as high as 19.2 µM. These results suggest that a broad window of $[O_2]$ could support net $N_2O$ accumulation and additional studies are needed to further
constrain this window and the resulting feedbacks between denitrification and marine deoxygenation.

**6. Appendix A: Estimating uncertainties for denitrifier samples**

Since only 2 mL of sample was available for preparation and analysis with the denitrifier method, it was sometimes impossible to achieve consistent peak areas. Instead of discarding low peak area samples, however, we wanted to establish a method to estimate the uncertainties associated with individual samples, based on their peak area. What follows is a method
for estimating this uncertainty, using the slope and intercept of the calibration curve and blank peak area.





The first step of this method is to calculate the peak area and $\delta(^{15}N)$ of the blank using the slope and intercept of the calibration curve. Then, we create a range of theoretical measured $\delta(^{15}N)$ for a set of theoretical samples based on a range of "actual" $\delta(^{15}N)$, a range of theoretical peak areas, and the peak area and $\delta(^{15}N)$ of the blank. Then, we correct these theoretical measured $\delta(^{15}N)$ values with the calibration curve, as one would do normally, to obtain $\delta(^{15}N_{corrected})$ for each theoretical sample. We estimate the error for each theoretical sample by comparing the $\delta(^{15}N_{corrected})$ we have calculated to the $\delta(^{15}N_s)$ we have assigned to it. Then, for each run, we can fit a function through these errors, their corresponding peak areas, and corresponding $\delta(^{15}N_s)$. We can then feed this function the peak area and measured $\delta(^{15}N)$ of actual samples in that run to estimate their uncertainties.

We start with a simple mass balance that states that the measured $\delta(^{15}N)$ is a function of the sample $\delta(^{15}N)$, sample peak area, blank $\delta(^{15}N)$, and blank peak area:

$$\delta\left(^{15}N_m\right)(m) = \delta\left(^{15}N_s\right)(s) + \delta\left(^{15}N_b\right)(b) \tag{A1}$$

where $\delta(^{15}N_m)$ is the measured $\delta(^{15}N)$, $m$ is the measured peak area, $\delta(^{15}N_s)$ is the sample $\delta(^{15}N)$, $s$ is the sample peak area, $\delta(^{15}N_b)$ is the $\delta(^{15}N)$ of the blank, and $b$ is the blank peak area. Dividing through by $m$:

$$\delta\left(^{15}N_m\right) = \delta\left(^{15}N_s\right)\left(\frac{s}{m}\right) + \delta\left(^{15}N_b\right)\left(\frac{b}{m}\right) \tag{A2}$$

Eqn. (A2) can be expressed as a linear equation $y = mx + b$, or the slope of $\delta\left(^{15}N_m\right)$ vs. $\delta\left(^{15}N_s\right)$. Note that this is the opposite of the calibration curve used to correct sample data, which is calculated from $\delta\left(^{15}N_s\right)$ vs. $\delta\left(^{15}N_m\right)$. Thus:

$$slope_1 = \left(\frac{s}{m}\right) \tag{A3}$$

$$intercept_1 = \delta\left(^{15}N_b\right)\left(\frac{b}{m}\right) \tag{A4}$$

Here, $slope_1$ and $intercept_1$ indicate the slope and intercept of the linear regression of $\delta\left(^{15}N_m\right)$ vs. $\delta\left(^{15}N_s\right)$. We can obtain the mean blank peak area $b$ from this slope and the mean peak area of the reference materials:

$$\left(\frac{b}{m}\right) = 1 - \left(\frac{s}{m}\right) = 1 - (slope_1) \tag{A5}$$

$$b = [1 - (slope_1)](m) \tag{A6}$$

where $m$ is the mean peak area of the reference materials. Finally, we obtain $\delta(^{15}N_b)$ from:



$$\delta\left(^{15}N_b\right) = intercept_1 \bigg/ \left(\frac{b}{m}\right) = \frac{intercept_1}{1 - (slope_1)} \tag{A7}$$

We assign the theoretical samples a range of theoretical measured peak areas $m$. The ratio of the blank peak area to the measured peak areas for a given sample is given by dividing $b$ (calculated from eqn. A6) by this theoretical peak area to obtain $\left(\frac{b}{m_i}\right)$, where $m_i$ is the theoretical peak area for that sample. Then, the ratio of sample peak area to measured peak area for a given theoretical sample is given by:

$$\left(\frac{s}{m_i}\right) = 1 - \left(\frac{b}{m_i}\right) \tag{A8}$$

As a 2D example, we assign all of the theoretical samples the same sample $\delta(^{15}N_s)$ of 180 ‰. Then, to obtain a range of
theoretical measured $\delta(^{15}N_m)$, we plug the $\delta(^{15}N_b)$ calculated from eqn. (A7), the range of $\left(\frac{b}{m_i}\right)$, the range of $\left(\frac{s}{m_i}\right)$ calculated from eqn. (A8), and 180‰ into eqn. (A2):

$$\delta\left(^{15}N_{m_i}\right) = 180‰ \cdot \left(\frac{s}{m_i}\right) + \delta\left(^{15}N_b\right)\left(\frac{b}{m_i}\right) \tag{A9}$$

We correct the range of $\delta\left(^{15}N_{m_i}\right)$ calculated from eqn. (A9) with the slope and intercept of the calibration curve $\delta\left(^{15}N_s\right)$ vs. $\delta\left(^{15}N_m\right)$:

$$\delta\left(^{15}N_{corrected_i}\right) = slope\left(\frac{s}{m_i}\right) + \delta\left(^{15}N_b\right)\left(\frac{b}{m_i}\right) \tag{A10}$$

Here, $slope_2$ and $intercept_2$ indicate the slope and intercept of the linear regression of $\delta\left(^{15}N_s\right)$ vs. $\delta\left(^{15}N_m\right)$. Then we
calculate the error associated with each theoretical sample with:

$$\delta\left(^{15}N_{error}\right) = \left|\delta\left(^{15}N_{corrected_i}\right) - 180‰\right| \tag{A11}$$

Following this exercise with a range of theoretical peak areas from 0.5 Vs to 10 Vs produces the following curve (Figure A1). It shows that these theoretical errors increase as peak area decreases.





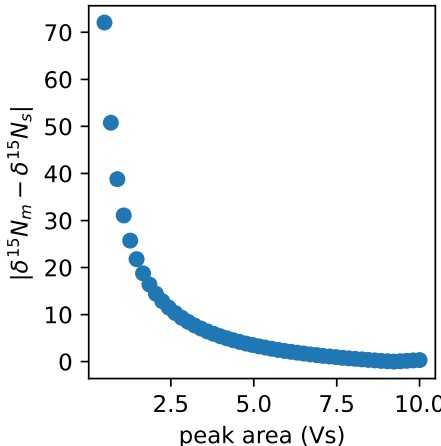

**Figure A1.** $\delta(^{15}N_{error})$ **vs. peak area for a range of theoretical samples with peak areas from 0.5 Vs to 10 Vs, based on the blank**
**peak area and** $\delta(^{15}N_b)$ **for a denitrifier run analysed on 6/24/2020.**

Repeating this exercise with a range of $\delta(^{15}N_s)$ from –20‰ to 180‰, instead of a single value, produces a 3D version of this curve (Figure S14). This shows that the estimated uncertainty is highest for samples with high $\delta(^{15}N_s)$ and low peak areas.

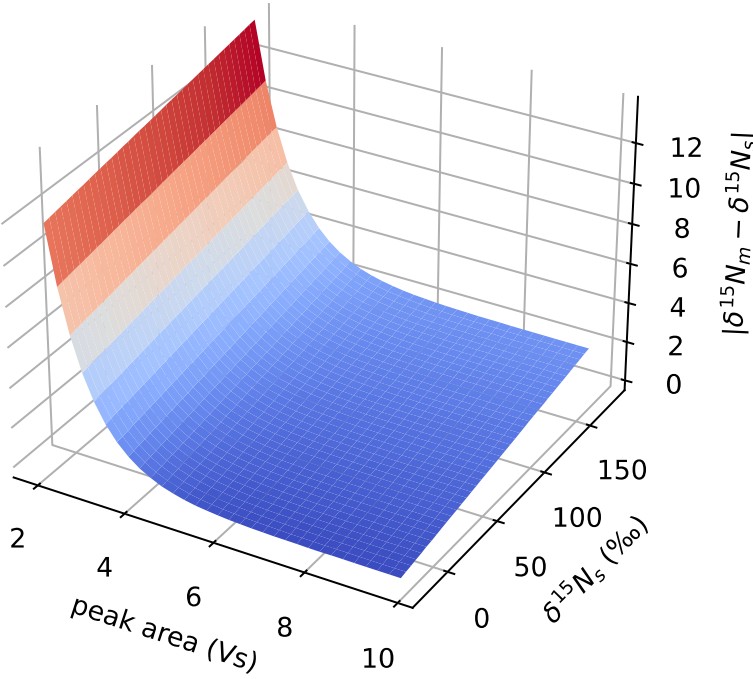

**Figure A2.** $\delta(^{15}N_{error})$ **vs. peak area and** $\delta(^{15}N_s)$ **for a range of theoretical samples with peak areas from 0.5 Vs to 10 Vs and** $\delta(^{15}N_s)$
**from –20‰ to 180‰, based on the blank peak area and** $\delta(^{15}N_b)$ **for a denitrifier run analysed on 6/24/2020.**



Finally, we fit a function of the following form through this theoretical data:

$$\delta\left(^{15}N_{error}\right) = a \cdot e^{b \cdot m} + c \cdot \delta\left(^{15}N_s\right) \qquad (A12)$$

where $a$, $b$, and $c$ are constants, $m$ is the assigned peak areas of the theoretical samples, and $\delta(^{15}N_s)$ is the assigned $\delta(^{15}N_s)$ for
the theoretical samples.

This procedure is repeated for each run of denitrifiers to produce coefficients $a$, $b$, and $c$ specific to that run. Then, to estimate the uncertainty associated with each measurement, we approximate $\delta(^{15}N_s)$ with the corrected $\delta(^{15}N)$ for each sample. We calculate uncertainty for each sample from $a$, $b$, and $c$ specific to each run, the corrected $\delta(^{15}N)$ for that sample,
and the peak area for that sample in eqn. (A12).

**Code/Data availability**

The data reported in this study can be found in the Stanford Digital Repository (https://doi.org/10.25740/ss974md4840).
Forward-running model code is available via Zenodo (https://doi.org/10.5281/zenodo.7810026). pyisotopomer, which was used for $N_2O$ isotopocule data corrections, is available for installation via the Python Package index
(https://pypi.org/project/pyisotopomer/) and Zenodo (http://doi.org/10.5281/zenodo.7552724).

**Author contributions**

CLK and KLC conceptualized the study, with input from CF and BBW. CLK and NMT carried out the experiments at sea, with assistance and supervision from CF and BBW. CLK and PAB analyzed the incubation samples in the laboratory. CLK performed the formal analysis of the data, developed the model code, and performed the model optimizations. KLC acquired
funding for the study. CLK prepared the manuscript with contributions from all co-authors.

**Competing interests**

The authors declare that they have no conflict of interest.

**Acknowledgements**

We would like to thank Stanford University and the Stanford Research Computing Center for providing computational
resources and support that contributed to this research. This research was supported by U.S. NSF grant OCE-1657868 to K.





L. Casciotti. C. L. Kelly was supported by an NSF Graduate Research Fellowship. The authors declare no competing financial interests.



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
