# Peer review of "Isotopomer labeling and oxygen dependence of hybrid nitrous oxide production"

_EGUsphere, 2023_

## Community Comment (CC1)

The authors present a supremely well executed study of N cycling rates in an oxygen deficient zones from well-controlled tracer incubations, from which they derive the relative contribution of respective processes to N2O production, and from which they document the sensitivity of said production pathways to dissolved oxygen concentrations. Their tracer incubations rely in part on site-preference measurements of isotopocules in order to determine pathways of production. Their data corroborate a dominance of denitrification in N2O production within the anaerobic regions of the water column, whereas multiple pathways operate concurrently in oxyclines. N2O production from ammonium, presumed to be catalyzed by nitrifiers, occurred dominantly through a hybrid pathway reliant on both ammonium and nitrite as substrates, whereas the hydroxylamine oxidation pathway (both N's in N2O from ammonium) was relegated to the well-oxygenated upper water column. The results and interpretation are highly informative, providing important constraints on pathways of N2O production and their respective sensitivity to oxygen.

I found the manuscript generally well written but, perhaps necessarily, a challenging read. I read it multiple times. The "cognitive challenge" arises from the inherent complexity of the topic and study design. It is also exacerbated by some structural elements of the manuscript that would benefit from revision: (a) The motivations for the study are not made clear in the introduction; (b) the general "order of operation" keeps jumping around in the results and discussion (I explain what I mean below), (c) there is a heavy reliance on supplementary materials, requiring a lot of back and forth.

I suggest a number of modifications that I think could improve ease of understanding by readers peripheral to the field of N2O isotopes who want to understand the findings and who also want to have a sense of the limitations of the findings.

The introduction does not effectively motivate the study. This study appears to be a companion to a published study where net rates of N cycling were determined from bulk tracer additions. I suppose that is why the bulk rate estimates figures were relegated to the supplements even though they are highly informative in the current context. Regardless, questions evidently emerged from the previous study that are presumably addressed herein, but these questions are not articulated in the introduction. I suggest the following paragraph sequence, which would make the intro more seamless:

The first paragraph alerts us that the study deals with nitrous oxide in oxygen deficient zones, with a justification of why N2O matters. In the second paragraph, the reader expects to learn where N2O is believed to come from in ODZ's. Instead, the paragraph otherwise begins with what seems a separate (but related) topic, N2O production by archaea, ocean-wide, not necessarily in ODZ's. In lieu, I suggest moving up the third paragraph to the second, to explain the current understanding that most N2O in ODZ's appears produced by denitrification. This would lead into a third paragraph that explains that nonetheless, a significant fraction appears to be produced by archaeal nitrification. I would present the current evidence that supports this hypothesis, in order to motivate "looking" for hybrid production, which is where this paper ultimately brings us.

The fourth paragraph should be explicit in whether it is referring to naturally occurring isotopes or tracer isotopes, since the subsequent paragraph jumps into tracers. To better motivate the study, perhaps this section can explain what naturally occurring isotopocules have divulged about N2O production in ODZ's specifically, and which questions remain unanswered – in order to link to the last paragraph of the intro.

IN the last paragraph, the motivation for measuring site preference on tracer experiments needs clearer articulation.  What additional insights can it provide that natural abundance or bulk tracer experiments did not? And your results, as I see them, inform on more than a dependence of oxygen on hybrid production, correct? They (a) corroborate previous findings on relative pathways of N2O production (b) uncover that the hybrid pathway dominates production by nitrification and (c) production from hydroxylamine is not a thing except at the surface. Importantly, do the results confirm inferences from natural abundance tracers in the same system? These can be posed as questions to which the authors can return in the discussion.

Methods:

Line 200: I would rephrase to "…. contribution of 15N15NO to masses 46 and 31, which, while negligible at natural abundance, becomes important in tracer experiments."

Equations 1-4: I think it would be wise to define ALL the terms in equations 1-4, for readers peripheral to this field who may still strive to understand the equations.

Line 245: Nitrate IS produced from nitrite when sulfamic acid (or any acid) is added to nitrite, due to the acid decomposition of nitrous acid. See Granger and Sigman 2009, Equations 6 and Figure 2. And 15N nitrate is a probable contaminant of the 15N nitrite solutions.

Line 274: what is N exchange between substrates?

Line 280: These "pathways" were not discovered by Wan et al. 2023. The citations are unclear to me.

Results:

I realize some of the data are published elsewhere but they are fundamental to navigating the paper. I suggest moving some of these back to the main text. In particular, the N2O production plots (mass 45 for each 15N substrate).

I suggest presenting the results in order of dominance of rates, and sticking to this pattern in all subsequent text and **figures**. Denitrification is fastest; detailing it first helps contextualize nitrite oxidation rates, which are also very high, and ammonium oxidation rates, which are puny.

Stick with one, NH3 or NH4 oxidation. It varies in the text.

Section 3.3 is very difficult to navigate. I read it multiple times. The term "high rates" is meaningless without context. Rates peak or not, but it can't be argued that rates of 45N2O-alpha are high even in this context, at picomolar per day. In this regard, I suggest using picomolar in lieu of multiple decimals in the text and figures, which are tiresome. And the Figure S8 is nearly impossible to navigate as every panel has a different x axis range.  Perhaps homogenize ranges for given isotopocule production? And I'm not sure why these figures are relegated to the supplements. I spent a long time looking at them. A long time…

The line at 215 belongs with the previous paragraph. And it's not clear whether this will be an example of rates varying in concert or not. Wordsmith accordingly.

Equation 13: In the case of nitrite where a higher concentration was added then intended, I would think that the flux derived therefrom, J, is no longer proportional to nitrite (zero order) at these concentrations. Does this matter?

Line 337: Wording of sentence is awkward.

Line 395: How can nitrite oxidation rates possibly be negative?

Line 420: Remind the reader what "f" designates.

Equation 19: "AP" was designated as "15F" in equations above…

Could p45excess result from misestimation of the actual atom percent of substrates the incubations? The rates are very small such having a small error on AP could potentially account for this? Or wrong proportion of carrier? I think Figure S9 may allude to this but the associated uncertainty needs to be better explained in the main text, whether or not the data evince unequal values of "f" beyond a reasonable "doubt"

Figure 4: Present in order brought up in text, which is N2O production from nitrate first.

Is production from NH4+ only necessarily hydroxylamine oxidation? It is called that in some figure captions. If so, it would be much easier for readers if it were called hydroxylamine oxidation throughout.

Section 3.5: I would start with describing N2O production "as a whole", followed by nitrate reduction (highest flux), etc… Same order of operation as suggested above.

Figure 4 d: the trace for ammonium oxidation differs from the corresponding trace in Figure 3 a.

Discussion:

Because the study is very complex, it would be beneficial for the discussion to begin with a paragraph that summarizes the dominant findings, rather than jumping into the deep end form the get go. In this regard, I would also get N2O production from denitrification out of the way first because it was the dominant flux, then discuss hybrid production. I find it interesting as well that production from hydroxylamine was virtually absent except at the surface – I think this merits more emphasis.

Section 4.3: I get that MOST N2O is produced by denitrification and 1/5 from hybrid production. Is that what is also inferred from natural abundance measurements, in these proportions? Curious minds want to know

Line 6452: What do you mean by "allowed?" Need better wording.

Line 650: qualify "this" , you mean the notion that internal pool are processed, not external…?

Line 600: Reader is left hanging: What are the implications for mechanisms of production? Need a concluding sentence for the paragraph to bridge it to the next, or simply amalgamate with the following paragraph.

Paragraph at line 605: Reads like something that should be in results section.

Line 610: Articulate fully for readers to catch up again "findings of unequal alpha vs. beta production during hybrid pathway have implications for interpretation of the natural abundance isotopes of N2O produced by hybrid process."

Paragraph at line 670: I don't understand why the results here should be different than cited study.

I remain perplexed by the following: In Figure S8, there is NO production of 45N2O from addition of 15NH4 at 100 m at station 1, yet there is reportedly 50 nM/day N2O production from the hybrid pathway at this depth… Am I fundamentally misunderstanding something about the experimental design? The hybrid pathway requires some input from 15NH4+ which should be detected as 45N2O?

---

## Author Response (AR1)

**Response to Reviewers 4/1/2024**
**Submission: "Isotopomer labeling and oxygen dependence of hybrid nitrous oxide production" [Manuscript ID egusphere-2023-2642]**

We thank the Editor and Reviewers for their time spent reviewing this manuscript and helping to improve it. We hope that the depth of this response to reviewers file reflects the thorough consideration the reviewers' suggestions. We provide a table of contents to aid in navigation.

**Table of Contents**

This Response to Reviewers file provides complete documentation of the changes made in response to each individual Reviewer comment. The document is designed so that these changes can be immediately read and understood, independent of the other comments and responses. While this comprehensive comment-by-comment explanation requires some duplication of material throughout the document, our intention is to help evaluate easily and effectively how each individual comment has been addressed.

Reviewer comments are shown in plain text**.** Author responses are shown in **bold**. Quotations from the revised manuscript are shown in *italics*.

**Reviewer 1**

General comments

This paper reports potential rates of $N_2O$ production by several important pathways in the oxygen deficient zone in the Eastern Tropical North Pacific. The studied oceanic region is one of the major $N_2O$ sources to the atmosphere, and therefore this work is crucial in understanding the origin of excess $N_2O$ and predicting the future emission of this global-warming and ozone-depleting gas under changing oceanic environment.

The strong point of this paper is that the $N_2O$ production rate and its dependence on dissolved oxygen concentration are determined for each of the possible $N_2O$ formation processes using [15]N-labeled substrates and isotopocule measurements. The authors found the significant contribution from hybrid $N_2O$ production during ammonia oxidation in the near-surface and deep $N_2O$ concentration maxima. They also found that the hybrid $N_2O$ formation is enhanced in low-oxygen water and that $N_2O$ can be produced by denitrification from nitrate even with oxygen concentrations higher than those considered to inhibit denitrification. I believe these findings will help us develop a clear picture of $N_2O$ cycling in and around the ODZ.

**Thank you for this positive and thorough assessment of our work.**

However, I think this work has a couple of drawbacks. First, the authors use conventional second-order kinetics to analyze $N_2O$ production processes in order to calculate the rate of each pathway from the $N_2O$ isotopocule ratios obtained by the $^{15}N$-incubation experiments. Considering that the amount of tracers added is sufficiently higher than those in initial seawater, I don't think it is always appropriate to assume that the rate is proportional to the concentration of substrates. I would like to see some justification or evidence on this assumption.

**If we understand the reviewer's comment correctly, the concern is that the rates of $N_2O$ production could have plateaued and not continued to increase (or decrease) with increasing (decreasing) substrate concentrations. In other words, the question is, what happens when you scale the $N_2O$ production rate to the substrate concentration instead of assuming that the rate has hit its maximum value?**

**There are two important things to clarify:**
1) **The model solves for the $2^{nd}$-order rate constant that best fits the data, given a certain concentration of substrate. These $2^{nd}$ order rate constants are not necessarily applicable to ambient substrate concentrations; thus, we report the rates, not the rate constants.**
2) **The substrate concentrations in eqns. (13) and (14) are the total concentration of substrate including the tracer and carrier additions, not just the ambient concentrations of each substrate. Because these substrate concentrations do not vary much in the incubations, eqns. (13) and (14) effectively amount to the same thing as assuming $N_2O$ production has plateaued and hit a maximum rate. Nonetheless there are some cases where the substrate concentrations change over the course of an incubation, and we assess below how this would influence our results.**

**The experiment with the highest rates of ammonia oxidation was at station PS3, feature "interface2" (63m, Table S1). Here, the rates of ammonia oxidation were 4.68 nM/day (Table S2). In the $^{15}N$-$NH_4^+$ experiment, the starting ammonium concentration was 0.52 μM, and the starting nitrite concentration was 1.61 μM. This includes the $^{15}N$-$NH_4^+$ tracer addition and $^{14}N$-$NO_2^-$ carrier addition. Then, the modeled hybrid $N_2O$ production rate declines by 1% over the course of the experiment:**

$$\frac{(0.52 - 0.00468)(1.61 + 0.00468)}{(0.52)(1.61)} \cdot 100 = 99\%$$

**Likewise, the modeled $N_2O$ production from solely ammonium declines by 2%:**

$$\frac{(0.52 - 0.00468)(0.52 - 0.00468)}{0.52^2} \cdot 100 = 98\%$$

**And the modeled rate of $N_2O$ production from $NO_2^-$ increases by 0.5%:**

$$\frac{(1.61 + 0.00468)(1.61 + 0.00468)}{1.61^2} \cdot 100 = 99\%$$

**Even in the experiment with the highest nitrite oxidation rate, from the secondary nitrite maximum (182 m) at station PS3, the modeled rate of N$_2$O production from NO$_2^-$ only declines by 12% over the course of the experiment, and the modeled rate of N$_2$O production from NO$_3^-$ only increases by 4% over the course of the experiment.**

**What if we compare the $^{15}$N-labeled ammonium treatment to the $^{15}$N-labeled nitrite treatment at the same experimental depth, since the tracer additions were unequal (5.00 µM $^{15}$N-NO$_2^-$ vs. 0.501 µM $^{15}$N-NH$_4^+$)? The $^{45}$N$_2$O and $^{46}$N$_2$O production rates in the $^{15}$N-labeled nitrite treatment were far higher than those in the $^{15}$N-labeled ammonium treatment, even when normalized by atom fraction. This is visualized below. In fact, the rates of production of $^{45}$N$_2$O and $^{46}$N$_2$O in the $^{15}$N-labeled ammonium treatments were so small, comparatively, that they are visually indistinguishable from zero when plotted on the same scale as the rates of production of $^{45}$N$_2$O and $^{46}$N$_2$O in the $^{15}$N-labeled nitrite treatments.**

[Figure]

Production of $^{45}$N$_2$O, divided by atom fraction, in the $^{15}$N-NO$_2^-$ treatment vs. $^{15}$N-NH$_4^+$ treatment at the same experimental depths. Red diamonds indicate p$^{45}$N$_2$O$^\alpha$/$^{15}$F and black diamonds indicate p$^{45}$N$_2$O$^\beta$/$^{15}$F. b) Production of $^{46}$N$_2$O, divided by atom fraction squared, in the $^{15}$N-NO$_2^-$ treatment vs. $^{15}$N-NH$_4^+$ treatment at the same experimental depths. In both plots, the dashed line is the 1:1 line.

**Since the tracer concentration was much higher in the $^{15}$N-labeled nitrite treatment (5.00 µM) than in the $^{15}$N-labeled ammonium treatment (0.501 µM), this imbalance of $^{45}$N$_2$O production supports the idea that there is some dependence of N$_2$O production rate on substrate concentration. The 2$^{nd}$ order kinetics in our model allow us to capture that dependence.**

Second, the contribution of suspended particulate matter to N$_2$O formation is not adequately taken into account in the interpretation of the results. Although the authors discuss the algal N$_2$O

production as an alternate source of N$_2$O, it seems that they do not pay more attention to other particulate matter. Why don't they consider potential N$_2$O production/consumption at anoxic microsites inside the particles? Although I don't know any reports on experimental evidence of such N$_2$O production, at least one paper suggested that active microbial CH$_4$ oxidation occurs within the oxic/anoxic boundary of sinking particles (Sasakawa, M.et al., 2008. JGR: Oceans, 113(C3). https://doi.org/10.1029/2007jc004217).

**We agree with the reviewer that particle-associated denitrification is a potential alternative N$_2$O source, especially at the highly productive coastal station. We have added particle associated N$_2$O production and consumption to the discussion of potential alternative sources of N$_2$O.**

*Additionally, since our samples were unfiltered, particle associated N$_2$O production and consumption may have occurred in some of our experiments, especially in experiments at the highly productive coastal station. We cannot rule out any of these alternative sources of N$_2$O in our samples, so we consider these processes as potential contributors to the bulk denitrifying flux discussed here.*

In summary, I recommend the publication of this paper after addressing the issue above and specific points below.

Specific comments
L64–66. Do the authors also mean NO does not undergo exchange with outside NO? In addition, are all the references listed here appropriate to cite? I cannot find the "evidence of nitrate reduction to N2O without exchange with an extracellular nitrite pool" in Monreal et al. (2022) and Toyoda et al. (2023).

**Yes, the process that we refer to here is N$_2$O production from externally sourced nitrate without exchange of intermediates outside the cell, including NO. This is the most likely mechanism explaining the large contribution of nitrate to N2O production, but as the reviewer pointed out, it has been implied but not tested experimentally. This is implicated in both of the cited papers as a major source of N$_2$O in the eastern tropical North Pacific and Bay of Bengal, respectively (Monreal et al., 2022; Toyoda et al., 2023). We have clarified this in the text.**

*Both direct rate measurements (Ji et al., 2015, 2018; Frey et al., 2020) and natural abundance isotope measurements (Kelly et al., 2021; Casciotti et al., 2018; Monreal et al., 2022; Toyoda et al., 2023) indicate that N2O production directly from nitrate (NO$_3^-$), i.e., without exchange with extracellular nitrite (NO$_2^-$) or nitric oxide (NO) pools, is the primary source of N2O in ODZs.*

L108–110. Is the STOX sensor identical with "Optode" in Table S1? It is confusing because "chemiluminescent optode" appears later in section 2.3.

**Apologies for the confusion here. The measurements from STOX sensor mounted on the rosette are different from the optode measurements reported in Table S1. We have removed the mention of the STOX sensor since we do not report any of its measurements.**

L131–133. I appreciate the authors' effort to avoid oxygen contamination, but isn't there any possibility that this procedure might reduce the oxygen concentration to the level lower than in situ seawater?

**This is indeed a concern, which is why only anoxic depths (where the ambient dissolved oxygen was below detection) were purged with He gas. Depths with low but non-zero ambient oxygen were not purged. The creation of a He headspace should also result in a small reduction in the dissolved oxygen in the sample after equilibration. In this case, however, the He headspace was so small (2 mL) that it did not outweigh or even compensate for the oxygen contamination introduced during sampling. This is shown in Figure S1.**

[Figure]

*Figure S1. [$O_2$] measured by chemiluminescent optodes mounted inside sample bottles vs. ambient [$O_2$] measured by a Seabird sensor for the bottles from which samples were taken. Data (circles) are plotted along the full range of [$O_2$] (a) and zoomed in to 0-20 $\mu M$ [$O_2$] (b). The dashed line in each plot is the 1:1 line. High values of optode [$O_2$] at 0 ambient [$O_2$] correspond to the two experiments at anoxic depths at station PS2 that were not purged before tracer addition.*

L161–162. How were the fiber optic cables pulled out of the bottle without air contamination?

**We apologize for the confusion. The FireSting fiber optic cables never enter the bottles, themselves. Instead, the fiber optic cables measure the signal from the oxygen sensor spot placed inside the bottles through the glass wall of the bottle. This has been clarified in the text.**

*At each timepoint, [$O_2$] was measured in each sensor bottle for at least 10 minutes using fiber optic cables paired to the oxygen optode spot mounted inside the bottle (PyroScience).*

L165. Could the fiber optic cables, not the sensors, be really calibrated?

**The fiber optic cables were indeed calibrated with a two-point calibration, using an oxygen sensor spot mounted inside a bottle containing 30 g/L sodium sulfite solution (0% saturation) and a sensor spot mounted inside a bottle containing air-equilibrated seawater**

**(100% saturation). The same two calibration bottles were used for all four of the fiber optic cables, effectively correcting them to the same scale. Differences in detection limit between sensor spots were accounted for by first performing this two-point calibration procedure to correct for differences between fiber optic cables, then measuring the minimum oxygen concentration measured by each sensor spot in helium-purged seawater (purged at 100 mL/min for 90 minutes, equal to 56 volume exchanges). We have added this explanatory text.**

*The fiber optic cables were calibrated with a 2–point measurement of: 1) a sodium sulfite solution (30 g/L in DI, or 0.24 M) and 2) surface seawater saturated with air at 12ºC (270 µM [$O_2$], based on a salinity = 35 psu and temperature = 12ºC) (Garcia and Gordon, 1992). The two calibration bottles, each containing its own optode spot, were used to calibrate all four of the fiber optic cables, effectively correcting them to the same scale. Differences in detection limit between sensor spots were accounted for by first performing this two–point calibration procedure to correct for differences between fiber optic cables, then measuring the minimum oxygen concentration measured by each sensor spot in purged seawater (purged at 100 mL/min. for 90 minutes, equal to 56 volume exchanges). Those detection limits were specific to each optode spot and varied from 146 – 880 nM [$O_2$].*

L177. Which does this optode mean, STOX or chemiluminescent? (see above)

**Again, apologies for the confusion. We refer here to the chemiluminescent optode measurements and have removed any mention of the STOX sensor from the text.**

L233–238. Because the sample for $N_2O$ measurements were poisoned with HgCl (L151), remaining sample could damage the denitrifying bacteria. How did the authors get around this problem?

**Samples are diluted in the denitrifier media (2.0 mL of sample with $HgCl_2$ into 5.0 mL total volume with denitrifying medium), so that the effective concentration of $HgCl_2$ that the denitrifiers experience is lower than typical for poisoning. In addition, the denitrifier method uses a high concentration of bacteria (denitrifiers grow in 440 mL medium for 5-7 days and are concentrated 10 times prior to using them to convert $NO_x$ to $N_2O$); no adverse effects from use of $HgCl_2$ have been observed.**

**In test runs, we found no statistically significant difference in the $\delta(^{15}N)$ of standards (USGS32, USGS34, and USGS35) prepared with and without $HgCl_2$. This was true of standards prepared with 20 nmol $NO_3^-$ and 10 nmol $NO_3^-$.**

L269. Why were not individual uncertainties for $\delta(^{15}N\text{-}NO_2^-)$ measurements estimated? Was there no need to apply the procedure for $\delta(^{15}N\text{-}NO_2^-)$ because of larger peak area obtained?

**Our method of estimating individual uncertainties was developed to deal with low $NH_3$ oxidation rates, which generated low peak areas in $\delta(^{15}N\text{-}NO_3^-)$ samples. Since the rates of $NO_3^-$ reduction were generally much higher than the rates of $NH_3$ oxidation (Table S2), a parallel method was not needed to estimate individual uncertainties in samples measured with the azide method, i.e. $\delta(^{15}N\text{-}NO_2^-)$ measurements. This has been clarified in the text.**

*Our method of estimating individual uncertainties was developed to deal with low $NH_3$ oxidation rates, which generated low peak areas in $\delta(^{15}N\text{–}NO_3^-)$ samples. Since the rates of $NO_3^-$ reduction were generally much higher than the rates of $NH_3$ oxidation (Table S2), a parallel method was not needed to estimate individual uncertainties in samples measured with the azide method, i.e. $\delta(^{15}N\text{–}NO_2^-)$ measurements, so rates of $NO_3^-$ reduction were with an ordinary least squares regression in eqn. (7) instead of a weighted least squares regression.*

L317. In the work by Frey et al. (2023), time course of $N_2O$ production was analyzed with Michaelis-Menten kinetics and Km values of 0.017–0.018 mM were obtained for oxycline at stations PS2 and PS3. In the present study, NH4+ was added at 0.5 mM, two orders of magnitude higher than the Km values. This means the rate of N2O production should reach to the maximum value, irrespective of substrate concentration.

**Please see response above regarding the representation of N₂O production kinetics in our model.**

L336, eq (16). Following the convention used for eq (14), 1/2 of the right-hand side of this equation should correspond to the ammonia consumption rate.

**Eq. (14) contains the factor ½ because that converts the rate of ammonia consumption in nM-N/day to N₂O production in nM-N₂O/day. We have clarified this in the text.**

*J was multiplied by ½ to convert the rate from nM N/day to nM $N_2O$/day, which was then multiplied by eqns. (9–12) to obtain the rates of production of each isotopocule (note that rates are reported in pM/day).*

L566–568. Describe more details about the "different conditions". It seems the location and cruise are identical between the two studies. Were date or time different? What were the differences in other hydrographic/chemical parameters?

**It is important to note that where our samples overlapped with this previous work, we observed similar results (>90% hybrid production). The depths where we observed a smaller proportion of hybrid production had not been sampled in previous work; it is possible that we sampled different microbial communities there, acclimated to different levels of ammonium, nitrite, and dissolved oxygen. This has been clarified in the text.**

*Previous work in the ETNP found that hybrid $N_2O$ production always comprised > 90% of $N_2O$ production from $NH_4^+$ (Frey et al., 2023), and where our samples overlapped with this previous work, we observed similarly high proportions of hybrid production (Fig. 5). The depths where we observed a smaller proportion of hybrid production had not been sampled previously; it is possible that we sampled different microbial communities there, acclimated to different levels of $NH_4^+$, $NO_2^-$, and dissolved oxygen.*

L590. On the basis of which data can this claim be made? Fig. S9 shows a clear deviation from the relationship expected for N2O production from a single substrate pool, but it does not present how the relation would be if NH4+ and NO2- were used in the ratio 1:1.

**That's true. We don't actually present evidence of the 1:1 ratio of $NH_4^+$ to $NO_2^-$; instead, hybrid N2O production is operationally defined in our model as a 1:1 combination of N derived from $NH_4^+$ and $NO_2^-$, which is generally consistent with previous work (Stieglmeier et al., 2014). Any combination of N derived from $NO_2^-$ with a second N derived from $NO_2^-$ would be included in the N2O production from $NO_2^-$ pool; likewise, any combination of N derived from $NH_4^+$ with a second N derived from $NH_4^+$ would be included in the N2O production from solely $NH_4^+$ pool. The question, then, is what reaction would be specific enough to have one N derived from each substrate, but not specific enough to govern $^{15}N$ placement in the resulting N2O? One such reaction could be the combination of $NH_4^+$ and $NO_2^-$ to form an intermediate such as hyponitrite (HONNOH or $^-ONNO^-$ in its deprotonated form), which reacts to form N2O via breakage of one of the N–O bonds, resulting in N2O that contains a 1:1 ratio of $NH_4^+$: $NO_2^-$. With a precursor such as hyponitrite, equal formation of $^{45}N_2O^\alpha$ and $^{45}N_2O^\beta$ could be achieved with non-selective N–O bond breakage. We have revised the discussion accordingly.**

*In our model, hybrid N2O production is operationally defined as a 1:1 combination of N derived from $NH_4^+$ and $NO_2^-$, which is generally consistent with previous work (Stieglmeier et al., 2014). Any combination of N derived from $NO_2^-$ with a second N derived from $NO_2^-$ would be included in the modeled quantity of N2O production from $NO_2^-$; likewise, any combination of N derived from $NH_4^+$ with a second N derived from $NH_4^+$ would be included in the N2O production from solely $NH_4^+$. The question, then, is what reaction would be specific enough to have one N derived from each substrate, but not specific enough to govern $^{15}N$ placement in the resulting N2O? One such reaction could be the combination of $NH_4^+$ and $NO_2^-$ to form a symmetrical intermediate such as hyponitrite (HONNOH or $^-ONNO^-$ in its deprotonated form), which reacts to form N2O via breakage of one of the N–O bonds, resulting in N2O that contains a 1:1 ratio of $NH_4^+$:$NO_2^-$. With a precursor such as hyponitrite, equal formation of $^{45}N_2O^\alpha$ and $^{45}N_2O^\beta$ could be achieved with non–selective N–O bond breakage.*

L614–616. I cannot understand whether the authors consider the N-O bond breakage occur randomly or at specific site regardless of $^{15}N$ distribution in the intermediate containing two N-O bonds. I see that the former case corresponds to $f = 1/2$, and $\delta^{15}N^{sp}$ will become equal to $\varepsilon$ (i.e., $^{14}N$-O bond at one side of the intermediate molecule is more likely to be broken than $^{15}N$-O bond at the other side). In the latter case, however, what happens if the bond cleavage resulting in $N^\beta$ of N2O does not proceed due to the slower rate for $^{15}N$ than $^{14}N$? We cannot rule out the possibility that the intermediate go back to substrate in such a case, but it accompanies N-N bond breakage, which should require more energy than N-O bond breakage. Rather, it appears that all intermediates are eventually converted to N2O. Then we don't need to consider $\varepsilon$ for the $N^\beta$-O bond breakage.

**Here we assume the former case: that either N-O bond could break, not at a specific site.**

L623–625 and 674–677. I agree that denitrification is not likely to proceed in the aerobic water column, but how about the microsites within suspended particles which might provide anaerobic condition?

**Good point — it is also possible that particle-associated denitrification is a potential driver of the $\delta(^{15}N^{sp})$ minimum observed in Popp et al. (2002) (L623-625). While we have removed the discussion of Popp et al. (2002), we added particle-associated denitrification as a potential contributor to our observed N₂O production from denitrification at higher-than-expected dissolved oxygen levels (L674-677).**

*Most surprising were the significant rates of N₂O production via denitrification at [O₂] >3 µM (Fig. 8g–h), which has previously been suggested as the threshold above which denitrification ceases (Dalsgaard et al., 2014). These observations are particularly evident in the plots of N₂O production from NO₃⁻ vs. incubation [O₂] (Fig. 8h), where positive, significant rates of N₂O production from NO₃⁻ were evident in incubations containing [O₂] as high as 19.2±0.8 µM (PS2 Deep ODZ Core experiment). One explanation for N₂O production via denitrification at such high levels of ambient dissolved oxygen is particle–associated denitrification (Bianchi et al., 2018; Smriga et al., 2021; Wan et al., 2023a).*

L632 (caption), It would be helpful if x-axis includes the full range of f (0 to 1).

**We modified the x-axis to include the full range of *f*:**

[Figure]

*Figure 10. Simulated values of $\delta(^{15}N^{sp})$ calculated with a range of f (the proportion of $N^\alpha$ derived from NO₂⁻ during hybrid N₂O production) and $\delta(^{15}N–NH_4^+) – \delta(^{15}N–NO_2^-)$, assuming $\varepsilon = 30.3‰$ (Santoro et al., 2011). Results are shaded by $\delta(^{15}N–NH_4^+) – \delta(^{15}N–NO_2^-)$. When f is less than or greater than ½, there is the potential for $\delta(^{15}N^{sp})$ to depend on the isotopic compositions of each substrate.*

L721–728. It seems that the authors assumes the first case I pointed out above. I cannot follow why the resulting site preference becomes variable.

**Thank you for making this point. We rephrased the conclusions to focus on the fact that we see more or less equal production of $^{45}N_2O^\alpha$ and $^{45}N_2O^\beta$ in most of our experiments, which would imply that hybrid $\delta(^{15}N^{sp})$ does *not* vary.**

*Based on the equal production of $^{45}N_2O^\alpha$ and $^{45}N_2O^\beta$ in the vast majority of our experiments, we posit a two–step process for hybrid N₂O production involving an initial bond–forming step that draws nitrogen atoms from each substrate to form a symmetric intermediate, and a second bond–breaking step that breaks an N–O bond in the symmetric intermediate to form N₂O. From this, we infer that hybrid N₂O production likely has a consistent $\delta(^{15}N^{sp})$, despite drawing from two distinct substrate pools. This has important implications for the interpretation of natural abundance isotopocule measurements, since it implies that it may be possible to define a $\delta(^{15}N^{sp})$ endmember for hybrid N₂O formation. More culture experiments are needed to quantify the $\delta(^{15}N^{sp})$ of N₂O produced by ammonia–oxidizing archaea under different temperatures, oxygen levels, and ratios of $NH_4^+:NO_2^-$.*

L768 (eq A10) and L769. "slope2" and "intercept2" do not appear in eq (A10). Is this equation correct?

**Thank you for catching this error. Eqn. (A10) was indeed written incorrectly. We corrected eqn. (A10) to include *slope₂* and *intercept₂* (now called *m* and *b*).**

$$\delta\left(^{15}N_{corrected_i}\right) = m\left(\frac{A_{sample}}{A_{measured,i}}\right) + b \tag{A10}$$

Table S3. If I understand correctly, f is applicable only to hybrid N2O production. Why values (including 0) are listed even when hybrid production rate is zero?

**Thank you for catching this error. We have removed the *f* values in Table S3 (now table S4) and Fig. S12 (now Fig. S10) for experiments where the hybrid production rate is zero. There are some very small but significant rates that were hidden due to how the numbers were rounded. The rates in Table S4 have been converted to pM/day to fix this issue.**

Technical corrections
L24. O in N2O should not be subscript.
**Corrected.**
*...as well as the isotopic labeling of the central (α) and terminal (β) nitrogen atoms of the N₂O molecule.*

L38. The error for the value "0.85" should be "0.03"?
**Corrected.**
*N₂O has a global warming potential 273 times that of carbon dioxide (Smith et al., 2021), and its atmospheric mixing ratio is increasing at a rate of 0.85±0.03 ppb/year (Tian et al., 2020).*

L43. The "m" in "mmol/kg" must be mu.
**Corrected.**
*ODZs have expanded over the last 60 years (Stramma et al., 2008; Breitburg et al., 2018) and will likely continue to do so as the oceans warm (Oschlies et al., 2018), although fate of the anoxic cores of ODZs ([$O_2$] ≤20 μmol/kg) remains uncertain (Cabré et al., 2015; Bianchi et al., 2018; Busecke et al., 2022).*

L202, eq (3). It seems unnatural to write down $^{18}R_{VSMOW}$ numerically, but not for $^{17}R_{VSMOW}$.
**Corrected.**

$$^{17}R/^{17}R_{\mathrm{VSMOW}} = (^{18}R/^{18}R_{\mathrm{VSMOW}})^{\beta}[\Delta(^{17}O) + 1] \qquad (3)$$

L266. Use a single character for parameters such as rate and slope.
**Corrected.**

$$rate \text{ (nM N/day)} = \frac{m(^{15}F_{product})[P]}{^{15}F_{substrate}} \qquad (7)$$

L293. nitrifier-denitrification using extracelluar NO2-.
**Corrected.**
*3) production from $NO_2^-$, i.e. denitrification or nitrifier–denitrification using extracellular $NO_2^-$ (blue hatched horizontal arrows);*

L302 (eq. 8). Subscripts "i" and "k" in the summation terms should be "n".
**Corrected.**

$$N_{t+1} = N_t + \Delta t \left( \sum_{n=1}^{i} J_n^{source} - \sum_{n=1}^{k} J_n^{sink} \right) \qquad (8)$$

L486 (Caption of Fig. 5). …total N2O production at stations PS1 (a), …
**Corrected (now Fig. 7).**
*Figure 7. $N_2O$ production from solely $NH_4^+$ (yellow bars), hybrid $N_2O$ production (green bars), $N_2O$ production from $NO_2^-$ (blue hatched bars), and $N_2O$ production from $NO_3^-$ (indigo bars) as proportions of total $N_2O$ production at stations PS1 (a), PS2 (b), and PS3 (c). Data are plotted over depth profiles of dissolved [$O_2$] (dashed lines) and [$N_2O$] (solid lines, from Kelly et al., 2021). Note broken y–axes and different x–axis scales for [$O_2$] and [$N_2O$] (top) and proportions (bottom).*

L506 (Caption of Fig. 6). I cannot see "values of a and b in white boxes", but a legend (without box) showing the fitting function in each panel.
**Apologies, the Copernicus system seemed to have removed any transparent objects (including these white boxes) from figures if they are saved as vector files. We removed the transparent objects to fix this issue.**

L529. "0.12 nM N2O/day" seems to correspond to "0.11" in Table S3.
**0.12 was the correct number. We have changed the units of Table S3 (now Table S4) to pM N2O/day to make the numbers easier to read.**

L614. Add equation number to the first equation, or continue the eq (24) from the first line by deleting "d($^{15}N^{sp}$)" in the left-hand side.
**Corrected.**

$$\delta\left({}^{15}N^{sp}\right) = \delta\left({}^{15}N^{\alpha}\right) - \delta\left({}^{15}N^{\beta}\right)$$

$$= \left[f\delta\left({}^{15}N - NO_2^-\right) + (1-f)\delta\left({}^{15}N - NH_4^+\right)\right] - \left[(1-f)\delta\left({}^{15}N - NO_2^-\right) + f\delta\left({}^{15}N - NH_4^+\right) - \varepsilon\right] \quad (24)$$

L754 (eq A2) and L755. It is confusing to use same character "m" and "b" in eq (A2) and the general equation for linear function.
**Changed terms "$m$" and "$b$" to "$A_{measured}$" and "$A_{blank}$".**

$$\delta\left({}^{15}N_{measured}\right) = \delta\left({}^{15}N_{sample}\right)\left(\frac{A_{sample}}{A_{measured}}\right) + \delta\left({}^{15}N_{blank}\right)\left(\frac{A_{blank}}{A_{measured}}\right) \quad (A2)$$

L757 and elsewhere. Parameters in equations A3–A7 and A10 should be written with a single character (and subscripts).
**Corrected.**

*Eqn. (A2) can be expressed as a linear equation y = mx + b, where m is the slope of $\delta\left({}^{15}N_{measured}\right)$ vs. $\delta\left({}^{15}N_{sample}\right)$ and b is the y–intercept. Thus:*

$$m = \left(\frac{A_{sample}}{A_{measured}}\right) \quad (A3)$$

$$b = \delta\left({}^{15}N_{blank}\right)\left(\frac{A_{blank}}{A_{measured}}\right) \quad (A4)$$

*We can obtain the mean blank peak area $A_{blank}$ from the slope and the mean peak area of the measured reference materials ($A_{measured}$):*

$$\left(\frac{A_{blank}}{A_{measured}}\right) = 1 - \left(\frac{A_{sample}}{A_{measured}}\right) = 1 - (m) \quad (A5)$$

$$A_{blank} = [1 - (m)](A_{measured}) \quad (A6)$$

*Finally, we obtain $\delta({}^{15}N_{blank})$ from:*

$$\delta\left({}^{15}N_{blank}\right) = {b}\Big/{\left(\frac{A_{blank}}{A_{measured}}\right)} = \frac{b}{1 - (m)} \quad (A7)$$

L972. Fix the author lists of Prokopiou et al. (2017).
**This reference has been removed.**

Title page of supplement says the file contains 14 figures, but I can see only 12.

**Corrected (there are now 10 supplementary figures).**

Figure S1. Add "a" or "b" to each panel.
**Corrected.**

[Figure]

*Figure S1. [O₂] measured by chemiluminescent optodes mounted inside sample bottles vs. ambient [O₂] measured by a Seabird sensor for the bottles from which samples were taken. Data (circles) are plotted along the full range of [O₂] (a) and zoomed in to 0-20 µM [O₂] (b). The dashed line in each plot is the 1:1 line. High values of optode [O₂] at 0 ambient [O₂] correspond to the two experiments at anoxic depths at station PS2 that were not purged before tracer addition.*

Figure S2. It would be helpful if the region of ambient nitrate between 20 and 50 mM is enlarged because the delta values look significantly higher than natural values.
**We added a panel (b) with values between 20 and 50 µM. They are indeed elevated.**

[Figure]

*Figure S2. δ($^{15}$N-NO$_x$) at t0 vs. ambient [NO$_3^-$] in $^{15}$N-NO$_2^-$ experiments across the full range of ambient [NO$_3^-$] (a) and from 20-50 µM [NO$_3^-$] (b).*

Figure S4. Fix the explanation of panels a–d so that the figures and caption are consistent. **Corrected.**

[Figure]

*Figure S4. Example forward-running model fit through N₂O isotopocule data for the $^{15}N\text{-}NH_4^+$ experiment in the secondary chlorophyll maximum at station PS3. Model output (solid lines) is optimized against the observed $^{44}N_2O$ (a), $^{46}N_2O$ (b), $^{45}N_2O^\alpha$ (c), and $^{45}N_2O^\beta$ (d) at each timepoint in each tracer experiment.*

Figure S7, caption. Fix the typo "bluen".
**Corrected.**

Figure S12, caption. Panel (b) is plotted against sigma theta, not nitrite.
**Corrected.**

[Figure]

*Figure S10. Weighted least squares regressions of f against ambient [$O_2$] (a) and potential density $\sigma_\theta$ (b). Slope, intercept, $R^2$, and p-values are displayed on each plot for the weighted least squares regression through the data. The value of f indicates the proportion of each N atom in $N_2O$ derived from $NH_4^+$ and $NO_2^-$ during hybrid $N_2O$ production; as approaches 1, more of $N^\alpha$ is derived from $NO_2^-$. Separation of $^{45}N_2O^\alpha$ and $^{45}N_2O^\beta$ production indicate values of f less than or greater than ½.*

**Reviewer 2**

This study presents very interesting findings on N2O hybrid production in marine environment. The complex approach applying 15N tracing methods in 3 different treatments with simultaneous measurements of d15N alfa and beta is very innovative and applied here for the first time in a real case study. Authors present the improved method of calculations of d15N alfa and beta in traced experiments, which has been integrated into the isotopomer-calculation software. These points are making this study important in further development of N2O-isotope based research, since the presented approach may broaden our interpretation potential of N2O isotopolocule studies.

However, the manuscript needs minor revision. Due to complexity of the experimental approach and results description, some aspects are difficult to follow by the reader and some information is missing. I suggest some technical corrections for this (below).

**Thank you very much for your insightful comments and suggestions! Please find our point-by-point responses below.**

But more importantly, I disagree with the conclusion that hybrid N2O formation results in incorporation of N atoms from 2 substrates into different positions of N2O molecule (alfa and beta) - because this is not supported by your data. Most of your samples indicate the opposite - that N is located in both position independently of the substrate - which you describe very nicely in section 4.2. Below, in the specific comments, I also explain my points in more detail.

**Very true. We have rephrased this part of the discussion, as well as the conclusions and abstract, to center around the fact that we *do* see equal formation of $^{45}N_2O^\alpha$ and $^{45}N_2O^\beta$ in most of our experiments, which would indicate that hybrid site preference does not vary after all.**

[revised manuscript text omitted]

Specific comments:
L80: Actual definition of delta values is

(Rsample/Rstandard–1)

factor 1000 is just due to expression in permil notation, should be omitted in the definition
**We removed the factor of 1000 from the definition.**
*The isotopic content of the individual nitrogen and oxygen atoms in the N₂O molecule are expressed in delta notation, defined as $\delta(^{15}N)$ or $\delta(^{18}O) = (R_{sample}/R_{standard}–1)$, where $R_{standard}$ for*

$\delta(^{15}N)$ and $\delta(^{18}O)$ are the ratios $^{15}N/^{14}N$ of air and $^{18}O/^{16}O$ of Vienna Standard Mean Ocean Water (VSMOW), respectively (Kim and Craig, 1990; Rahn and Wahlen, 2000; Toyoda and Yoshida, 1999).

L 151: 2%-92% - that wide range? is this correct?

**This is correct. We added 1 µM $^{15}N$-NO$_3^-$ to all of our experimental depths, regardless of the ambient NO$_3^-$ concentration, resulting in a wide range of atom fractions due to the wide range of ambient NO$_3^-$ concentrations. At depths where ambient NO$_3^-$ is high, however, and thus the atom fraction is low, the rate of N$_2$O production from NO$_3^-$ is high enough that we still get a detectable signal in $^{45}N_2O$ and $^{46}N_2O$ (see Figures 4 and 5).**

L194: It should be described more precisely how much was added, depending on the concentration and enrichment level? I understand this was just a dilution procedure for mineral nitrogen isotope measurements? Or also for N2O measurements? It is a bit misleading because this chapter title is N2O isotopocule measurements... so I am not sure if my understanding is correct. Or you have diluted mineral nitrogen forms in your experiment to dilute your produced N2O in the headspace? Why not to dilute the N2O sample with any technical N2O gas to get respective dilutions?

**The first paragraph of Section 2.4 describes the sample preparation procedure, immediately prior to mass spec analysis of liquid samples for nitrous oxide isotopocules. Since we run liquid samples on the purge-and-trap system (see below), we need to protect the purge-and-trap system from highly $^{15}N$-enriched NH$_4^+$, NO$_2^-$, and NO$_3^-$ dissolved in the sample. To accomplish this, 100 µL of $^{14}NH_4Cl$, Na$^{14}NO_2$, or K$^{14}NO_3$ carrier was added to each sample a final concentration of 54 µM, 262 µM, or 27 µM, respectively, to bring $^{15}N$ tracer levels below 5000 ‰. We have clarified the above in the text.**

*Two steps were taken to prepare incubation samples for N$_2$O isotopocule analysis immediately prior to measurement. First, a 5 mL aliquot was removed from each sample by syringe and replaced with He gas. These aliquots were refrigerated until analysis for [NO$_2^-$] and [NH$_4^+$] to check tracer and carrier additions, as mentioned above. After this aliquot was removed, 100 µL of $^{14}NH_4Cl$, Na$^{14}NO_2$, or K$^{14}NO_3$ carrier was added to each sample a final concentration of 54 µM, 262 µM, or 27 µM, respectively, to bring $^{15}N$ tracer levels below 5000‰. Note that these carrier additions were different from the $^{14}N$ carrier added to each incubation alongside $^{15}N$ tracer; the purpose of the later carrier additions was to prevent exposure of the IRMS system to highly $^{15}N$–enriched substrates.*

section 2.4: Actually you do not say how you finally collect your gaseous N2O samples - which volume, which containers, which procedure? Were the N2O samples colleted once only from each bottle or regularly in some time intervals?

**We apologize for any confusion here. The purge-and-trap system completely extracts the dissolved N$_2$O from the sample (incubation) bottle and is described in greater detail in McIlvin and Casciotti (2010). So, one bottle = one sample. Time series are constructed by sacrificing triplicate bottles over a time course, rather than resampling the incubation bottles over time. This time series approach is now stated explicitly in the methods section.**

*Time series were constructed by sacrificing triplicate bottles over a time course, rather than resampling the incubation bottles over time.*

**We describe how liquid samples were collected for incubation in section 2.2, "sample collection."**

Equation 3: what value was assumed for D17O?

**$\Delta(^{17}O)$ was assumed to be 0. We have added this to the text.**

*Here, $\Delta(^{17}O)$ was assumed to be equal to 0.*

Figure 2: should the yellow arrow between NH4 and NO2- go in both directions? since this represents formation of hybrid N2O with cellular NO2-, right?

**We added an arrow representing hybrid N₂O with cellular NO₂⁻. The vertical arrow was between NH₄⁺and NO₂⁻ was a bit confusing since it did not represent an N₂O production processes, only NH₄⁺oxidation to NO₂⁻. We made the vertical arrows colorless to indicate that they are not N₂O production processes.**

[Figure]

| Pathway name | Includes N$_2$O production from… | Corresponding process in Wan et al. (2023) |
|---|---|---|
| Production from solely NH$_2$⁺ | Hydroxylamine oxidation | Pathway 1 |
| | Hybrid production using cellular NO$_2$⁻ | Pathway 2 |
| | Nitrifier–denitrification using cellular NO$_2$⁻ | N/A |
| Hybrid production | Hybrid production using extracellular NO$_2$⁻ | Pathway 3 |
| Production from NO$_2$⁻ | Denitrification or nitrifier–denitrification using extracellular NO$_2$⁻ | N/A |
| Production from NO$_3$⁻ | Denitrification using cellular NO$_2$⁻ | N/A |

*Figure 2. Schematic of the forward–running model used to solve for rates of N₂O production. Horizontal arrows represent processes whose rates are solved for, while vertical arrows represent processes whose rates are prescribed based on our experimental results. The model solves for 2$^{nd}$–order rate constants for four N₂O–producing processes: 1) production from solely NH₄⁺ (yellow horizontal arrows), which includes N₂O from hydroxylamine oxidation (Wan et al., 2023 Pathway 1), hybrid production using cellular NO₂⁻ (Wan et al., 2023 Pathway 2), and nitrifier–denitrification using cellular NO₂⁻; 2) hybrid production using NH₄⁺ and extracellular NO₂⁻ (green arrows, Wan et al., 2023 Pathway 3); 3) production from NO₂⁻, i.e. denitrification or nitrifier–denitrification using extracellular NO₂⁻ (blue hatched horizontal arrows); and 4) production from NO₃, i.e. denitrification*

*or nitrifier–denitrification using cellular $NO_2^-$ (indigo horizontal arrows). The model also solves for f, the proportion of $N^\alpha$ derived from $NO_2^-$ during hybrid $N_2O$ production. $NH_3$ oxidation (yellow vertical arrows), $NO_2^-$ oxidation (blue hatched vertical arrows), and $NO_3^-$ reduction to $NO_2^-$ (indigo vertical arrows) are modeled as first–order rates to account for $^{15}N$ transfer between substrate pools, as described in the main text. Finally, $N_2O$ consumption (black dashed arrow) is modeled as first–order to $N_2O$. It is assumed that while the distribution of $^{15}N$ in each tracer experiment at a given station and depth is different, the overall rates and mechanisms of $N_2O$ production are the same regardless of which substrate is labeled. The model is optimized against the observed $^{46}N_2O$, $^{45}N_2O^\alpha$, $^{45}N_2O^\beta$, and $^{44}N_2O$ at each timepoint in each tracer experiment (black box).*

Equation 18: I think this definition, with some explanation why this is possible should appear in methods section 2.6 This does not fit in results section. Same with Eq. 19

**We respectfully disagree. Section 2.6 describes the modeling framework, and the model does not use equations (18) and (19). Actually, the modeling framework is a much more nuanced way of estimating the rates of hybrid $N_2O$ formation than simply using eqns. (18) and (19). Eqns. (18) and (19) are just a way of showing that hybrid $N_2O$ production is indeed occurring in our experiments, which we do in section 3.3.**

L 550: Why there is such large difference in NH3 oxidation with different studies? - it should be discussed - is this due to different analytical approaches?

**There are several factors that may have contributed to Travis et al. (2023) measuring higher rates of ammonia oxidation than our study or that of Frey et al. (2023). The incubations in Travis et al. (2023) were performed at different depths than ours, so they likely captured different microbial communities, different light levels, different chemical conditions (nitrate, dissolved oxygen, etc.). This is further exaggerated by the fact that the oxycline was moving up and down during the course of our occupation of PS3 (as indicated by oxygen profiles captured by an Argo float near our sampling stations during the time of our cruise, Figure S4 in Sun et al., 2021), so even experiments performed at the same depth on different days would likely sample different biogeochemical conditions. Nitrification rates tend to show a very sharp subsurface maximum (the feature Travis et al focused on) and the resolution of the depth profiles in our study was not optimized to "catch" it. Finally, the incubations performed in Travis et al. (2023) were fully aerobic, whereas ours were generally low-oxygen and gas-tight. For example, the dissolved oxygen in our incubation with the highest rates of ammonia oxidation was 2 μM (see tables S1 and S2).**

[Figure]

**Figure S4. O₂ profiles based on one Argo float at a station (13.1 N, 108.4 W) between our sampling stations PS2 and PS3 in the ETNP OMZ.** These data were collected in the same month that our samples were collected. Color indicates the date and time of data collection. The 20 m interval (100–120 m) varied from nearly 100 μM to below detection within two weeks.

**We also needed to make a correction: the highest rate of ammonia oxidation measured by Travis et al. (2023) was actually 90±2 nM/day, not 48.7 nM/day.**

*$NH_3$ oxidation rates in this study were smaller than those measured on the same cruise by Travis et al. (2023), who measured $NH_3$ oxidation rates as high as 90±2 nM/day in fully oxygenated incubations at station PS3.*

L 600: why, which process can be responsible for this? Very important observation! You could give more details to these points - which processes dominated there, what was the N2O flux (rather high or low) or how it is possible to interpret these data?

**Please see comments in response to L 613.**

L 613: But in the first and second paragraph in this section 4.2 you showed that the values originating from NO2 and NH4 are mixed and finally the formed N2O has randomly situated 15N atoms from NO2 and NH4

I see, below the Eq (24) you explain, in most cases it is equally distributed but in some it is not. But why? The reader is a bit lost here

In the second paragraph you described very precisely how the hybrid formation may function and why we get equal distribution, and this is very convincing. So, the few cases with f unequal

0.5 must be due to some other process, some different mechanism? I understand this is rather an exception than a rule for hybrid formation – but you define this as a rule in Eq.24 (and then repeat this as final conclusion).

This is very important to describe this correctly here because for NA studies we do not know f, hence your conclusions here will be crucial for d15N-SP interpretations in NA studies.

**Thank you for these comments. We revised the discussion in section 4.2 to reflect the fact that the majority of our experiments have equal formation of $^{45}N_2O^\alpha$ and $^{45}N_2O^\beta$ and $f$ within error of 0.5. This is actually a very important finding for the interpretation of natural abundance $N_2O$ isotopocules because it implies that hybrid $N_2O$ would indeed have a constant $\delta(^{15}N^{sp})$, despite being derived from two different sources. We revised section 4.2, the conclusions, and the abstract to reflect the equal formation of $^{45}N_2O^\alpha$ and $^{45}N_2O^\beta$ seen in most of our experiments and the implications of $f$ being equal to 0.5.**

[revised manuscript text omitted]

L 630: ok, but maybe you can sum up what were the conditions for the samples with f unequal 0.5 in your studies

I believe that it is rather not the hybrid process that behaves sometimes like this and sometimes the other way but rather admixture of some other processes, or the issue with the usage of cellular and extracellular NO2-. What about possible fungal co-denitrification that may show different mechanism?

I think you have so much data that maybe some hypotheses can be made?

**The experiments with unequal $^{45}N_2O^\alpha$ and $^{45}N_2O^\beta$ formation spanned a range of oxygen concentrations, depths, and substrate concentrations, and no clear patterns emerged. We do note that significant relationships emerged between $f$ and ambient [O₂] ($R^2 = 0.84$, p < 0.001; Fig. S12a) and potential density anomaly ($\sigma_\theta$) ($R^2 = 0.72$, p < 0.001; Fig. S12b), although both relationships exhibited a large amount of scatter. These oxygen and potential density gradients may be proxies for changing archaeal community compositions at different depths in the water column, which may exhibit different patterns of incorporation of $NO_2^-$-derived N and $NH_4^+$-derived N into $N^\alpha$ and $N^\beta$. We now note this in the text.**

**Thanks for the suggestion that we may have sampled a different "hybrid" process at these depths, such as fungal co-denitrification (Shoun et al., 2012), which may proceed via a**

**different pathway from archaeal hybrid N₂O production. We added this alternative to the text.**

*The unequal production of $^{45}N_2O^\alpha$ and $^{45}N_2O^\beta$ observed at certain depths led to values of f significantly different from 0.5 (Table S4). At these depths, $N^\alpha$ retained a different proportion of nitrogen derived from $NO_2^-$ and $NH_4^+$ than $N^\beta$, causing $^{45}N_2O^\alpha$ and $^{45}N_2O^\beta$ to diverge. The depths with f ≠ 0.5 anchored significant relationships between f and ambient [$O_2$] ($R^2 = 0.84$, p < 0.001; Fig. S10a) and potential density anomaly ($\sigma_\theta$) ($R^2 = 0.72$, p < 0.001; Fig. S10b). The oxygen and potential density gradients may be proxies for changing archaeal community compositions at different depths in the water column, which may exhibit different patterns of incorporation of $NO_2^-$–derived N and $NH_4^+$–derived N into $N^\alpha$ and $N^\beta$. It is also possible that we sampled a different "hybrid" N₂O–producing process at these depths, such as fungal co–denitrification (Shoun et al., 2012), which may proceed via a different pathway from archaeal hybrid N₂O production.*

L 703: Have you observed any activity, any N2O production in HgCl2 poisoned treatments? Would be interesting to report what was the "background" N2O production, since in some studies it appears quite high.

Was this in the expected range of abiotic N2O production?

**We agree with the reviewer that there is a concern about abiotic reactions between $NO_2^-$ and $HgCl_2$. In our $^{15}N$-$NO_2^-$ experiments, the $t_0$ samples did not have $\delta(^{15}N$-$N_2O)$ or $\delta(^{18}O$-$N_2O)$ outside of the natural abundance range, which would have indicated an abiotic reaction between the $^{15}N$-$NO_2^-$ tracer and $HgCl_2$ e.g., during storage of the samples prior to analysis. In comparison, we do see some elevated $\delta(^{15}N$-$NO_x)$ in these samples (Figure S2), indicating that the sulfamic acid treatment may have converted some $^{15}N$-$NO_2^-$ to $^{15}N$-$NO_3^-$, and/or that there was $^{15}N$-$NO_3^-$ contamination in our $^{15}N$-$NO_2^-$ tracer. That is why it is important to measure $t_0$'s in case an abiotic reaction should shift the baseline and it is necessary to account for this shift, as we have done.**

L 724: But this conclusion is not supported by the previous sentence. From the mechanism you describe it is expected that the alfa and beta positions are independent of the substrate origin.

I do not agree with this conclusion since MOST of your samples do not support this, only in few cases you observed differences in alfa and beta position, so rather the opposite conclusion should be given here, with an indication that there are also some exceptions, with not fully understood mechanism (in my opinion resulting from admixture of processes which has not been taken into consideration - e.g. fungal co-denitrification - which you admit in section 4.6, that fungal N2O can be an important source and it is not included in your model). You have actually concluded this at the end of your section 4.2 properly. You cannot simplify this into different direction in the conclusions because people will mostly read only conclusions, and this is very important point impacting the interpretations of natural abundance N2O isotopocule studies very much.

**We revised the conclusions to reflect the fact that we see equal formation of $^{45}N_2O^\alpha$ and $^{45}N_2O^\beta$ in most of our experiments, and thus that hybrid N₂O is *not* likely to have a variable**

$\delta(^{15}N^{sp})$. **This is an equally strong conclusion because it implies that it may be possible to define a $\delta(^{15}N^{sp})$ endmember for hybrid N$_2$O formation.**

*5. Conclusions*

*We applied N$_2$O isotopocule measurements to $^{15}$N tracer incubations to measure N$_2$O production rates and mechanisms in the ETNP. We found that N$_2$O production rates peaked at the oxic–anoxic interface above the ODZ, with the highest rates of N$_2$O production from NO$_3^-$. Hybrid N$_2$O production peaked in both the shallow and deep oxyclines, where NH$_3$ oxidation was also active, and exhibited yields as high as 21% of ammonia oxidation.*

*Based on the equal production of $^{45}$N$_2$O$^\alpha$ and $^{45}$N$_2$O$^\beta$ in the vast majority of our experiments, we posit a two–step process for hybrid N$_2$O production involving an initial bond–forming step that draws nitrogen atoms from each substrate to form a symmetric intermediate, and a second bond–breaking step that breaks an N–O bond in the symmetric intermediate to form N$_2$O. From this, we infer that hybrid N$_2$O production likely has a consistent $\delta(^{15}N^{sp})$, despite drawing from two distinct substrate pools. This has important implications for the interpretation of natural abundance isotopocule measurements, since it implies that it may be possible to define a $\delta(^{15}N^{sp})$ endmember for hybrid N$_2$O formation. More culture experiments are needed to quantify the $\delta(^{15}N^{sp})$ of N$_2$O produced by ammonia–oxidizing archaea under different temperatures, oxygen levels, and ratios of NH$_4^+$:NO$_2^-$.*

733: These observations can be also due to fungal activity since fungal species usually tolerate higher oxygen levels than bacteria.

**Thank you for pointing this out. We added fungal denitrification as a potential explanation for some of the N$_2$O production from denitrification at higher oxygen levels than expected, both in the conclusions and in section 4.4, "Oxygen dependence of N$_2$O production rates and yields".**

*Most surprising were the significant rates of N$_2$O production via denitrification at [O$_2$] >3 μM (Fig. 8g–h), which has previously been suggested as the threshold above which denitrification ceases (Dalsgaard et al., 2014). These observations are particularly evident in the plots of N$_2$O production from NO$_3^-$ vs. incubation [O$_2$] (Fig. 8h), where positive, significant rates of N$_2$O production from NO$_3^-$ were evident in incubations containing [O$_2$] as high as 19.2±0.8 μM (PS2 Deep ODZ Core experiment). One explanation for N$_2$O production via denitrification at such high levels of ambient dissolved oxygen is particle–associated denitrification (Bianchi et al., 2018; Smriga et al., 2021; Wan et al., 2023a). Fungal denitrification may also have contributed to these fluxes, since denitrifying fungi can tolerate a higher level of oxygen than their bacterial counterparts.*

**Reviewer 3**

The authors present an impressively thorough analysis of $N_2O$ isotope systematics from a field study in the oxygen deficient zone of the eastern tropical north Pacific (ETNP) – a region well studied for its redox active nitrogen cycle. Through a suite of $^{15}N$ labeling experiments and the leveraging of those results, the paper lays out a complex yet compelling argument for the ecological distribution of various pathways of $N_2O$ production. Taking the isotopic scrutiny to the next level, the paper presents a powerful and novel analytical model that leverages both the relative formation of singly labeled ($^{45}N_2O$) and doubly labeled ($^{46}N_2O$) as well as the site-specific labeling of the inner (alpha) and outer (beta) N atoms across all experiments (e.g., $^{15}N$ labeled $NH_4^+$, $NO_2^-$ or $NO_3^-$) to solve for relative contribution of $N_2O$ formation pathways. To my knowledge, such a sophisticated analysis has not been braved – and the authors should be commended for it.

The authors also use their results to evaluate the $O_2$ sensitivities of each of the formation pathways under these field incubation conditions, tying the results to both in situ $O_2$ and incubation levels of $O_2$ (which sometimes differed from in situ). These results show that adopted thresholds for $N_2O$ production by denitrification (for example) may not be as hard and fast as previously thought. The data provide quantitative relationships from which models can be built for estimating wider patterns in $N_2O$ production.

Especially unique and thought-provoking was the model analysis interrogating the possible impact on natural abundance site-preference compositions in $N_2O$ as arising from hybrid formation – especially the proposed involvement of a symmetric intermediate. I very much enjoyed Section 4.2 which carefully walks the reader through the logic of the analysis and argues for the hybrid pathway involving formation of a symmetric intermediate (such as hyponitrite). Equation 24 demonstrates how, with a symmetric intermediate (and a 50/50 contribution of $NH_4^+$ and $NO_2^-$ precursors) – the actual composition of the precursors does not impact site preference. However, if this 50/50 proportion varies (as they observe in some incubations) – then this assumption falls apart – and could in fact explain or demonstrate that the site preference values for hybrid $N_2O$ formation may vary under differing ambient conditions. While exceptionally nuanced, I found the arguments laid out in this section to offer real strides forward in our collective understanding.

I also found particularly useful the demonstration of how go about combining probabilistic analysis of $N_2O$ formation (e.g., stochastic distribution of $^{15}R$ between alpha and beta positions) with the $^{15}N$ labeling exercise (where an excess of doubly labeled $N_2O$ (15-15-16) may arise depending on formation pathways). Introduction of this 'excess' term allows for the application of site-specific composition to determine N atom sources under $^{15}N$ labeling circumstances. To my knowledge, this approach has not been leveraged previously – and thus the manuscript contains a wealth of valuable methodological information – which I found laid out very clearly. Thus, the paper should also stand as a useful model for work beyond $N_2O$ dynamics in ODZs – and could provide a model for application to a range of other systems.

Overall, because of the complex nature of the work - this paper is a beast to get through. That being said – it is excellently written and offers a wealth of value for really pulling apart the

complexity of environmental $N_2O$ formation. I provide some minor editorial comments below which hopefully help to highlight some areas that could be clarified. I recommend publication.

**We are sincerely grateful for this positive and thorough evaluation of our work. Thank you for taking the time to work through the many aspects of this paper.**

**Specific Comments:**
There is a lot of complex discussion of $N_2O$ isotope systematics – which are notoriously challenging to understand. I can see that the authors are very careful to be clear in explaining most things and using careful wording for helping the readers follow the logic.

What were isotope effects used for $NH_4^+$ oxidation, etc.? Table? Would variation of these values (for example) impact the error estimates as mentioned in L350-352?

**We added a supplementary table (now Table S3) of the isotope effects used in the model for $NH_4^+$ oxidation, $NO_2^-$ oxidation, $NO_3^-$ reduction, and $N_2O$ reduction. Since we're dealing with tracer-level $^{15}N$, though, natural abundance-level isotope effects are unlikely to affect the model results. No isotope effects were applied to $N_2O$ formation.**

**Table S3. Fractionation factors used the time-dependent numerical model.**

| Process | $^{15}\varepsilon^{bulk}$ (‰) | $^{15}\varepsilon^{\alpha}$ (‰) | $^{15}\varepsilon^{\beta}$ (‰) | Reference |
|---|---|---|---|---|
| $NH_4^+$ oxidation | 22.0 | | | Santoro and Casciotti, 2011 |
| $NO_2^-$ oxidation | -15.0 | | | Casciotti, 2009 |
| $NO_3^-$ reduction to $NO_2^-$ | 5.0 | | | Granger et al., 2008 |
| $N_2O$ reduction to $N_2$ | | 11.8 | 0.0 | Kelly et al., 2021 |

While I recognize here a nomenclature used for isotope ratios (e.g., "$\delta(^{15}N)$") has been adopted to be in line with some recent protocols, I find the use of the extra set of parentheses extremely distracting, unnecessary, and confusing. While I'm sure that the adoption of such conventions was intended to help clarify, the addition of more symbols into these terms does not help the reader and frankly muddies the message. I may very well be a minority here, but simply don't see the logic in these new conventions (especially in the context of $N_2O$ which is already complex enough). I see zero value in adopting the new nomenclature, and though probably futile, would suggest the authors stick to the nomenclature that has been in use for decades (e.g., $\delta^{15}N$).

**The justification for writing $\delta$ values with parentheses, e.g., $\delta(^{15}N)$, is that $\delta$ is the quantity symbol and "$^{15}N$" is the label. I started using this notation in Kelly et al. (2023) in order to reflect the recommendations in the latest SI Brochure (https://www.bipm.org/en/publications/si-brochure/ ) and I continue its use here for consistency and semantic precision. I understand that this is a change from the conventions in the field and is likely to be unpopular, but perhaps the notation will become less confusing if it is more widely adopted.**

The paragraphs starting on Line 610, together with Equation 24 and Figure 8 worked to convince me that when the proportion of $NO_2^-$ and $NH_4^+$ to hybrid $N_2O$ formation is equal (and the intermediate is a symmetric molecule), then the actual $^{15}N$ content (or $\delta^{15}N$ value) of those substrates does not play a role in the emergent site preference value. Why then on line 725 in the conclusion – do the authors state that these values do matter (even if 1:1 contribution)? Is it not true that the hypothetically variable site preference values from hybrid $N_2O$ formation actually emerge from variations in the 50/50 (or 1:1) contribution – and that only in those cases will the values of the substrates play into the site preference of the product $N_2O$ (as in Figure 8)? Please clarify.

**Thank you for this comment. When the contributions of $NO_2^-$ and $NH_4^+$ *to each N position* are equal, hybrid site preference doesn't depend on the isotopic composition of either substrate. You could hypothetically have $N_2O$ containing a 1:1 ratio of $NO_2^-$ and $NH_4^+$, but with $N^\alpha$ always derived from $NO_2^-$ ($f=1$), and in this case site preference would depend strongly on the isotopic composition of each substrate. But in most of our experiments, $N^\alpha$ is equally derived from $NO_2^-$ and $NH_4^+$, which would imply that hybrid site preference does not vary. This means that it may even be possible to identify an isotopic endmember for hybrid $N_2O$ production, which would be very useful to the natural abundance $N_2O$ isotopocule community. We have revised the discussion and throughout the paper to reflect this majority case. This is an important clarification of the results, so we are grateful to you (and the other reviewers) for pointing this out.**

[revised manuscript text omitted]

**Technical Corrections:**
Methods: Perhaps I missed this somewhere. What volume of sample was collected for the N$_2$O analyses? 160ml serum bottles? Foil bags?

**In 2.2, "Sample collection," we state that "Incubation samples were filled directly from Niskin bottles into 160 mL glass serum bottles (Wheaton) using Tygon tubing. Incubation bottles were overflowed three times before being capped and sealed with no headspace using gray butyl rubber septa (National Scientific) and aluminum crimp seals." In response to this comment and a similar comment from Reviewer 2, we added a clarification that time series were constructed by sacrificing triplicate bottles over a time course, rather than resampling the incubation bottles over time.**

*Time series were constructed by sacrificing triplicate bottles over a time course, rather than resampling the incubation bottles over time.*

L24:  N$_2$O formatting
**Corrected**
*Using $^{15}$N–labeled tracer incubations, we measured the rates of N$_2$O production from ammonium (NH$_4^+$), nitrite (NO$_2^-$), and nitrate (NO$_3^-$) in the Eastern Tropical North Pacific ODZ, as well as the isotopic labeling of the central ($\alpha$) and terminal ($\beta$) nitrogen atoms of the N$_2$O molecule.*

L25: 'forward running model' – unclear what this means… numerical model? Analytical model? Is there some terminology you could use here to help clarify?

**Changed to "time-dependent numerical model".**
*Implementing the rates of labeled N$_2$O production in a time–dependent numerical model, we found that N$_2$O production from NO$_3^-$ dominated at most stations and depths, with rates as high as $1600\pm200$ pM N$_2$O/day.*

L 86: instead of 'unlinked to' (which seems a little awkward) maybe consider 'independent from'
**Corrected.**
*In natural abundance studies, $\delta(^{15}N^{sp})$ is particularly useful because if exhibits distinct values for different N$_2$O production processes, independent of the isotopic value of the substrate (Toyoda et al., 2002; Sutka et al., 2003, 2006, 2004; Toyoda et al., 2005; Frame and Casciotti, 2010).*

L134: Was the introduction of this background N$_2$O done as a gas or in dissolved form?
**Gas form. Added to the text.**
*After sparging, 100 µL of 1030 ppm N$_2$O in He (4 nmol N$_2$O) in gaseous form was introduced back into each bottle for a final concentration of 26 nM to provide a constant background of N$_2$O for later isotopic analysis (Fig. S4a).*

L148: … to provide enough total NO$_2^-$ …

**Corrected.**

*The Na$^{14}$NO$_2$ and $^{14}$NH$_4$Cl amendments served two purposes: 1) to provide enough total NO$_2^-$ for isotopic analysis of $^{15}$NO$_2^-$ produced from $^{15}$NH$_4^+$, and 2) to minimize isotope dilution of the substrate pool, which can cause underestimation of rates with low substrate additions.*

L229: Here referring to the precision being lower, but the standard deviations being higher is a little confusing. Perhaps refer to the precision being 'poorer'?
**Corrected.**
*The analytical precision was poorer than that in a similar natural abundance dataset (Kelly et al., 2021) due to minor $^{15}$N carry–over in some of the standards analyzed immediately following highly enriched samples.*

L245: …another explanation would be that the $^{15}$NO$_2^-$ tracer actually may have contained some amount of $^{15}$NO$_3^-$ to begin with.
**Added to the text.**
*Incubations with low ambient [NO$_3^-$] had high $t_0$ $\delta(^{15}$N) values (>1000 ‰; Fig. S2). This is likely because NO$_3^-$ is produced when sulfamic acid is added to NO$_2^-$ (Granger and Sigman, 2009), so the sulfamic treatment probably chemically converted some $^{15}$N–NO$_2^-$ tracer to $^{15}$N–NO$_3^-$; additionally, $^{15}$N–NO$_3^-$ is a possible contaminant of the $^{15}$N–NO$_2^-$ tracer solutions.*

L253: seawater water?
**Corrected.**
*Reference materials were diluted from 200 mM working stocks into 3 mL NO$_2^-$–free seawater in 5 and 10 nmol quantities of NO$_2^-$ to correct for the contribution of a consistent blank to a range of sample sizes.*

L256: …precision for the denitrifier and azide methods is typically better…
**Corrected.**
*The $\delta(^{15}$N) analytical precision for the denitrifier and azide methods is typically better (Sigman et al., 2001; McIlvin and Altabet, 2005), but tracer measurements tend to have lower analytical precision than natural abundance measurements.*

L336: here the word 'exchange' is used to refer to movement of $^{15}$N from one pool to another occurring through biologically mediated processes. I would suggest using the word 'transfer' and not 'exchange' – as exchange is often used to refer to abiotic (or enzyme mediated) equilibration between two distinct pools.
**Corrected here and throughout the text.**
*Rates of $^{15}$N and $^{14}$N transfer between substrate pools via NH$_3$ oxidation, NO$_2^-$ oxidation, and NO$_3^-$ reduction were also included in the model.*

L348: extra comma
**Corrected.**
*The model was optimized using the Nelder–Mead Simplex algorithm (Nelder and Mead, 1965), implemented in the Scipy optimization library (Virtanen et al., 2020), which has been used successfully for natural abundance N$_2$O isotopocule models (Monreal et al., 2022).*

L395: With respect to the apparent negative nitrite oxidation rate – can any explanation here be invoked? Is this a real phenomenon or just some random analytical artifact that can't be easily explained?

**The "negative" nitrite oxidation rates at two depths are likely an artifact of the elevated $t_0$ $\delta(^{15}N)$ values in some of our $^{15}N$-$NO_2^-$ treatments (discussed above). We have added this to the text.**

*In some cases, $NO_2^-$ oxidation rates appeared negative due to a decrease in $^{15}N–NO_3^-$ vs. incubation time (Fig. 3b, h), which was likely an artifact of the elevated $t_0$ $\delta(^{15}N)$ values in some of our $^{15}N–NO_2^-$ treatments (discussed above).*

L456: sediment-water interface?

**This measurement was made at 898 m, which was very close to the bottom depth at station PS3. Clarified in the text.**

*At station PS3, there was also a small, significant rate of $NH_3$ oxidation ($0.303\pm0.005$ nM N/day) at 898 m, which was close to the bottom depth (Fig. 3i).*

L490: $N_2O$ production pathways

**Corrected.**

*The oxygen dependencies of $N_2O$ production pathways were determined by fitting model derived $N_2O$ production pathways vs. $[O_2]$ using the following rate law:*

L725: depends on the $^{15}N$ content of each substrate

**The conclusions have been modified to reflect the fact that we actually see approximately equal placement of $NO_2^-$-derived N and $NH_4^+$-derived N in $N^\alpha$ and $N^\beta$, and thus that hybrid site preference may actually be constant (see response to L610 above).**

**Community Comment, Julie Granger**

The authors present a supremely well executed study of N cycling rates in an oxygen deficient zones from well-controlled tracer incubations, from which they derive the relative contribution of respective processes to N2O production, and from which they document the sensitivity of said production pathways to dissolved oxygen concentrations. Their tracer incubations rely in part on site-preference measurements of isotopocules in order to determine pathways of production. Their data corroborate a dominance of denitrification in N2O production within the anaerobic regions of the water column, whereas multiple pathways operate concurrently in oxyclines. N2O production from ammonium, presumed to be catalyzed by nitrifiers, occurred dominantly through a hybrid pathway reliant on both ammonium and nitrite as substrates, whereas the hydroxylamine pathway (both N's in $N_2O$ from ammonium) was relegated to the well-oxygenated upper water column. The results and interpretation are highly informative, providing important constraints on pathways of N2O production and their respective sensitivity to oxygen.

I found the manuscript generally well written but, perhaps necessarily, a challenging read. I read it multiple times. The "cognitive challenge" arises from the inherent complexity of the topic and study design. It is also exacerbated by some structural elements of the manuscript that would benefit from revision: (a) The motivations for the study are not made clear in the introduction; (b) the general "order of operation" keeps jumping around in the results and discussion (I explain what I mean below), (c) there is a heavy reliance on supplementary materials, requiring a lot of back and forth.

I suggest a number of modifications that I think could improve ease of understanding by readers peripheral to the field of N2O isotopes who want to understand the findings and who also want to have a sense of the limitations of the findings.

**Thank you for taking the time to thoroughly read and understand our paper, and for your constructive feedback. We have restructured the paper according to your suggestions and hope that it is easier to follow as a result.**

The introduction does not effectively motivate the study. This study appears to be a companion to a published study where net rates of N cycling were determined from bulk tracer additions. I suppose that is why the bulk rate estimates figures were relegated to the supplements even though they are highly informative in the current context. Regardless, questions evidently emerged from the previous study that are presumably addressed herein, but these questions are not articulated in the introduction. I suggest the following paragraph sequence, which would make the intro more seamless:

The first paragraph alerts us that the study deals with nitrous oxide in oxygen deficient zones, with a justification of why N2O matters. In the second paragraph, the reader expects to learn where N2O is believed to come from in ODZ's. Instead, the paragraph otherwise begins with what seems a separate (but related) topic, N2O production by archaea, ocean-wide, not necessarily in ODZ's. In lieu, I suggest moving up the third paragraph to the second, to explain the current understanding that most N2O in ODZ's appears produced by denitrification. This would lead into a third paragraph that explains that nonetheless, a significant fraction appears to

be produced by archaeal nitrification. I would present the current evidence that supports this hypothesis, in order to motivate "looking" for hybrid production, which is where this paper ultimately brings us.

**Thank you for this helpful suggestion. We moved up paragraph three of the introduction (N2O production via denitrification) and revised (formerly) paragraph two to focus more on motivating our discussion of hybrid production.**

[revised manuscript text omitted]

The fourth paragraph should be explicit in whether it is referring to naturally occurring isotopes or tracer isotopes, since the subsequent paragraph jumps into tracers. To better motivate the study, perhaps this section can explain what naturally occurring isotopocules have divulged about N2O production in ODZ's specifically, and which questions remain unanswered – in order to link to the last paragraph of the intro.

**We made it more explicit that paragraph four is about natural abundance isotopes. We also revised it to focus on the fact that hybrid $N_2O$ production complicates the interpretation of natural abundance $\delta(^{15}N^{sp})$ because it draws from two different substrate pools.**

*The stable, natural abundance nitrogen and oxygen isotopes of $N_2O$ can provide quantification of – and distinction among – potential $N_2O$ cycling mechanisms (Kim and Craig, 1990; Rahn and Wahlen, 2000; Toyoda and Yoshida, 1999). For example, natural abundance $N_2O$ isotopocule studies have indicated that the high, near–surface $N_2O$ accumulations in the eastern tropical North Pacific (ETNP) ODZ are 80% derived from denitrification and 20% derived from nitrification (Kelly et al., 2021). The isotopic content of the individual nitrogen and oxygen atoms in the $N_2O$ molecule are expressed in delta notation, defined as $\delta(^{15}N)$ or $\delta(^{18}O) = (R_{sample}/R_{standard}–1)$, where $R_{standard}$ for $\delta(^{15}N)$ and $\delta(^{18}O)$ are the ratios $^{15}N/^{14}N$ of air and $^{18}O/^{16}O$ of Vienna Standard Mean Ocean Water (VSMOW), respectively (Kim and Craig, 1990; Rahn and Wahlen, 2000; Toyoda and Yoshida, 1999). In addition to the bulk nitrogen and oxygen isotope ratios in $N_2O$, we can measure the isotopic content of the inner (α) nitrogen atom and an outer (β) nitrogen atom in $N_2O$ (Toyoda and Yoshida, 1999; Brenninkmeijer and Röckmann, 1999). The difference in the $^{15}N$ content of these two atoms is often referred to as the 'site preference' and is defined as $\delta(^{15}N^{sp}) = \delta(^{15}N^{\alpha}) – \delta(^{15}N^{\beta})$. In natural abundance studies, $\delta(^{15}N^{sp})$ is particularly useful because if exhibits distinct values for different $N_2O$ production processes, independent of the isotopic value the substrate (Toyoda et al., 2002; Sutka et al., 2003, 2006, 2004; Toyoda et al., 2005; Frame and Casciotti, 2010). This allows for partitioning between different $N_2O$ sources, and has been used extensively to quantify $N_2O$ cycling in the ocean (Toyoda et al., 2002, 2019, 2021, 2023; Popp et al., 2002; Toyoda et al., 2005; Yamagishi et al., 2007; Westley et al., 2006; Farías et al., 2009; Bourbonnais et al., 2017, 2023; Casciotti et al., 2018; Kelly et al., 2021; Monreal et al., 2022). As we elaborate upon in the discussion, however, the premise that $\delta(^{15}N^{sp})$ exhibits a unique and consistent value depends on the assumption that both N atoms in $N_2O$ are derived from a singular substrate pool. Thus, hybrid $N_2O$ production may complicate traditional interpretations of natural abundance $N_2O$ isotopocules.*

IN the last paragraph, the motivation for measuring site preference on tracer experiments needs clearer articulation. What additional insights can it provide that natural abundance or bulk tracer experiments did not? And your results, as I see them, inform on more than a dependence of oxygen on hybrid production, correct? They (a) corroborate previous findings on relative pathways of N2O production (b) uncover that the hybrid pathway dominates production by nitrification and (c) production from hydroxylamine is not a thing except at the surface. Importantly, do the results confirm inferences from natural abundance tracers in the same system? These can be posed as questions to which the authors can return in the discussion.

**We added a sentence to the last paragraph saying that $^{45}N_2O^{\alpha}$ and $^{45}N_2O^{\beta}$ measurements create an additional constraint on N2O production rates and thus allow us to quantify different source process more precisely and accurately. As per your suggestion, we also detailed more thoroughly the different findings from this study.**

*Previous studies have used $^{15}N$ tracer experiments to measure N2O production rates in ODZs (Ji et al., 2015, 2018; Frey et al., 2020, 2023). These studies used the accumulation of $^{45}N_2O$ and $^{46}N_2O$ resulting from the addition of $^{15}N$–labeled substrates such as $^{15}NH_4^+$ and $^{15}NO_2^-$ to measure N2O production rates. To our knowledge, however, the isotopomer measurement has never been applied to $^{15}N$–tracer experiments to track $^{15}N$ from different substrates into the $\alpha$ and $\beta$ positions of the N2O molecule. Here, we present data showing the production of N2O isotopomers with $^{15}N$ in the $\alpha$ position ($^{45}N_2O^{\alpha}$) and $^{15}N$ in the $\beta$ position ($^{45}N_2O^{\beta}$) from $^{15}N$–labeled NH_4^+, NO_2^-, and NO_3^-. Measuring the production of $^{45}N_2O^{\alpha}$ and $^{45}N_2O^{\beta}$ creates an additional constraint on N2O production mechanisms and thus allows us to quantify different source process more precisely and accurately. We employed these measurements to (a) validate previous $^{15}N$ tracer studies of N2O production rates in the ETNP, (b) uncover that the hybrid pathway dominates production by nitrification, (c) establish the insignificance of production from solely NH_4^+ except the surface, and (d) infer a constant $\delta(^{15}N^{sp})$ for hybrid N2O, despite drawing from two substrate pools. We also use these results to confirm inferences from natural abundance N2O isotopocules measured in the same system (Kelly et al., 2021).*

Methods:
Line 200: I would rephrase to "…. contribution of 15N15NO to masses 46 and 31, which, while negligible at natural abundance, becomes important in tracer experiments."
**Corrected.**
*The number ratios of isotopomers $^{14}N^{15}NO$ and $^{15}N^{14}NO$ were calculated as in Kelly et al., 2023, with the following modifications to account for contribution of $^{15}N^{15}NO$ to the molecular ion number ratios 46/44 ($^{46}R$) and 31/30 ($^{31}R$), which, while negligible at natural abundance, becomes important in tracer experiments.*

Equations 1-4: I think it would be wise to define ALL the terms in equations 1-4, for readers peripheral to this field who may still strive to understand the equations.
**Corrected.**
*In natural abundance samples, pyisotopomer solves the following four equations to obtain $^{15}R^{\alpha}$ and $^{15}R^{\beta}$:*

$$^{45}R = {}^{15}R^{\alpha} + {}^{15}R^{\beta} + {}^{17}R \qquad (1)$$

$$^{46}R = \left({}^{15}R^{\alpha} + {}^{15}R^{\beta}\right){}^{17}R + {}^{18}R + {}^{15}R^{\alpha}\,{}^{15}R^{\beta} \qquad (2)$$

$$^{17}R/{}^{17}R_{VSMOW} = ({}^{18}R/{}^{18}R_{VSMOW})^{\beta}[\Delta({}^{17}O) + 1] \qquad (3)$$

$$^{31}R = \frac{(1-\gamma)\,{}^{15}R^{\alpha} + \kappa\,{}^{15}R^{\beta} + {}^{15}R^{\alpha}\,{}^{15}R^{\beta} + {}^{17}R[1 + \gamma\,{}^{15}R^{\alpha} + (1-\kappa)\,{}^{15}R^{\beta}]}{1 + \gamma\,{}^{15}R^{\alpha} + (1-\kappa)\,{}^{15}R^{\beta}} \qquad (4)$$

*Where $^{45}R$, $^{46}R$, and $^{31}R$ are the molecular ion number ratios 45/44, 46/44, and 31/30. $^{15}R^{\alpha}$, $^{15}R^{\beta}$, $^{17}R$ and $^{18}R$ denote the number ratios of $^{14}N^{15}N^{16}O$, $^{15}N^{14}N^{16}O$, $^{14}N_2{}^{17}O$, and $^{14}N_2{}^{18}O$, respectively, to $^{14}N_2{}^{16}O$. Here, $\Delta(^{17}O)$ was assumed to be equal to 0. In these equations, the term $(^{15}R^{\alpha})(^{15}R^{\beta})$ represents the statistically expected contribution of $^{15}N^{15}N^{16}O$ to the $^{46}R$ and $^{31}R$ ion number ratios, based on the probabilities of forming $^{15}N^{15}N^{16}O$. The probability of getting $^{15}N$ in $N^{\alpha}$ is given by $^{15}R^{\alpha}$ and the probability of getting $^{15}N$ in $N^{\beta}$ is given by $^{15}R^{\beta}$; furthermore, the two probabilities are assumed to be independent, so the probability of getting $^{15}N$ in both positions would be $(^{15}R^{\alpha})(^{15}R^{\beta})$ (Kaiser et al., 2004). Predicting the concentration of $^{15}N^{15}N^{16}O$ from the distribution of $^{15}N$ in the singly–labeled molecules ($^{15}R^{\alpha}$ and $^{15}R^{\beta}$) is a reasonable assumption for natural abundance samples, where the concentration of $^{15}N^{15}N^{16}O$ is extremely low (Magyar et al., 2016; Kantnerová et al., 2022).*

Line 245: Nitrate IS produced from nitrite when sulfamic acid (or any acid) is added to nitrite, due to the acid decomposition of nitrous acid. See Granger and Sigman 2009, Equations 6 and Figure 2. And 15N nitrate is a probable contaminant of the 15N nitrite solutions.
**We revised this section to say that our high t0's are likely because $NO_3^-$ is produced when sulfamic acid is added to $NO_2^-$ (Granger and Sigman, 2009), so the sulfamic treatment probably chemically converted some $^{15}N\text{-}NO_2^-$ tracer to $^{15}N\text{-}NO_3^-$; additionally, $^{15}N\text{-}NO_3^-$ is a probable contaminant of the $^{15}N\text{-}NO_2^-$ tracer solutions.**
*Incubations with low ambient $[NO_3^-]$ had high $t_0$ $\delta(^{15}N)$ values (>1000 ‰; Fig. S2). This is likely because $NO_3^-$ is produced when sulfamic acid is added to $NO_2^-$ (Granger and Sigman, 2009), so the sulfamic treatment probably chemically converted some $^{15}N–NO_2^-$ tracer to $^{15}N–NO_3^-$; additionally, $^{15}N–NO_3^-$ is a possible contaminant of the $^{15}N–NO_2^-$ tracer solutions. Regardless, this would have shifted all three timepoints equally, and thus should not introduce a bias into the slope of $\delta(^{15}N–NO_3^-)$ with time and the rates calculated there from.*

Line 274: what is N exchange between substrates?
**Sorry, "exchange" is probably the wrong word here. We have changed it to N transfer between substrates.**
*While it is possible to calculate rates of hybrid and bacterial $N_2O$ production with linear regressions of $^{45}N_2O$ and $^{46}N_2O$ with time (Trimmer et al., 2016), these calculations cannot take into account $^{15}N$ transfer between substrates, and more importantly, produce separate rate estimates for separate tracer experiments.*

Line 280: These "pathways" were not discovered by Wan et al. 2023. The citations are unclear to me.

**We changed these citations to "referred to as Pathway 1 in Wan et al., 2023…".**
*The model encoded four different $N_2O$ producing pathways: 1)* **production from solely $NH_4^+$**, *which includes $N_2O$ from hydroxylamine oxidation (referred to as Pathway 1 in Wan et al., 2023), hybrid production using cellular $NO_2^-$ (referred to as Pathway 2 in Wan et al., 2023) and nitrifier–denitrification using cellular $NO_2^-$; 2)* **hybrid production** *using extracellular $NO_2^-$ (referred to as Pathway 3 in Wan et al., 2023); 3)* **production from $NO_2^-$**, *i.e. denitrification or nitrifier–denitrification using extracellular $NO_2^-$; and 4)* **production from $NO_3^-$**, *i.e. denitrification using cellular $NO_2^-$ (Fig. 2).*

Results:
I realize some of the data are published elsewhere but they are fundamental to navigating the paper. I suggest moving some of these back to the main text. In particular, the N2O production plots (mass 45 for each 15N substrate).
**To clarify: none of the data included in this study have been published elsewhere. A companion paper (Frey et al., 2023) published rates of ammonia oxidation and $N_2O$ production from ammonium measured in concurrent, but separate, experiments. Nevertheless, we have moved the $^{45}N_2O$ and $^{46}N_2O$ production plots into the main text. They are now figures 4 and 5.**

I suggest presenting the results in order of dominance of rates, and sticking to this pattern in all subsequent text and figures. Denitrification is fastest; detailing it first helps contextualize nitrite oxidation rates, which are also very high, and ammonium oxidation rates, which are puny.
**We changed the order of section 3.2, "Nitrification and nitrate reduction rates," to talk about denitrification first, then nitrite oxidation, then ammonia oxidation.**

*3.2 Nitrification and nitrate reduction rates*
*$NO_3^-$ reduction to $NO_2^-$ occurred at rates ranging from $0.54\pm0.04$ to $33.2\pm0.1$ nM N/day (Table S2). There was a small, significant rate of $NO_3^-$ reduction to $NO_2^-$ in apparently aerobic waters near the surface at station PS1 (Fig. 3a). The highest rates of $NO_3^-$ reduction to $NO_2^-$ occurred in the deep, anoxic waters at station PS2 ($33.24\pm0.01$ nM N/day; Fig. 3d) and in the secondary chlorophyll maximum at station PS3 ($19.2\pm0.1$ nM N/day; Fig. 3g).*

*$NO_2^-$ oxidation rates ranged from $13.05\pm0.08$ nM N/day to $465\pm86$ nM N/day (Table S2). The highest rates of $NO_2^-$ oxidation occurred within apparently oxygen deficient waters, at $81.0\pm0.2$ nM N/day in the secondary chlorophyll a maximum at station PS2 and at $465\pm86$ nM N/day in the secondary $NO_2^-$ maximum at station PS3 (Fig. 3e, h; Table S2). Note that these are potential rates, since the $^{15}N$ addition was generally much greater than the ambient concentration (Lipschultz, 2008). In some cases, $NO_2^-$ oxidation rates appeared negative due to a decrease in $^{15}N–NO_3^-$ vs. incubation time (Fig. 3b, h), which was likely an artifact of the elevated $t_0$ $\delta(^{15}N)$ values in some of our $^{15}N–NO_2^-$ treatments (discussed above). We chose, however, not to left censor the data.*

*$NH_3$ oxidation to $NO_2^-$ occurred at small, but significant rates ranging from $0.19\pm0.0004$ nM N/day to $4.68\pm0.07$ nM N/day (Table S2). At every station, rates of $NH_3$ oxidation peaked near the base of the mixed layer, at the same depth as the near–surface $[N_2O]$ maximum (Fig. 3c, f, i). At station PS2, $NH_3$ oxidation showed a secondary peak at the same depth as the deep $[N_2O]$*

*maximum (Fig. 3f). At station PS3, there was also a small, significant rate of $NH_3$ oxidation (0.303±0.005 nM N/day) at 898 m, which was close to the bottom depth (Fig. 3i). Rates of $NH_3$ oxidation were generally lower than $NO_2^-$ oxidation and undetectable in oxygen deficient waters (Fig. 3c, f, i).*

Stick with one, NH3 or NH4 oxidation. It varies in the text.
**We changed all of these to $NH_3$ oxidation.**

Section 3.3 is very difficult to navigate. I read it multiple times. The term "high rates" is meaningless without context. Rates peak or not, but it can't be argued that rates of 45N2O-alpha are high even in this context, at picomolar per day. In this regard, I suggest using picomolar in lieu of multiple decimals in the text and figures, which are tiresome. And the Figure S8 is nearly impossible to navigate as every panel has a different x axis range. Perhaps homogenize ranges for given isotopocule production? And I'm not sure why these figures are relegated to the supplements. I spent a long time looking at them. A long time…

**You're right, in this section "high rates" is relative. We revised "high rates" to "relatively higher rates."**
*At each station, the observed rates of net $^{46}N_2O$ (Fig. 4), $^{45}N_2O^\alpha$ and $^{45}N_2O^\beta$ (Fig. 5) production from $^{15}N–NH_4^+$, $^{15}N–NO_2^-$, and $^{15}N–NO_3^-$ all peaked at or just below the oxic–anoxic interface, where the near surface $[N_2O]$ maximum was found. There were also relatively higher rates of net $^{46}N_2O$ production from $^{15}N–NO_2^-$ and $^{15}N–NO_3^-$ within the secondary $NO_2^-$ maximum (253 m) at station PS2 (Fig. 4d–e). Relatively high rates of net $^{45}N_2O^\alpha$ and $^{45}N_2O^\beta$ production also occurred in the secondary $NO_2^-$ maximum at stations PS2 (253m; Fig. 5d–e) and PS3 (182 m; Fig. 5g–h). The net rates of $^{45}N_2O^\alpha$ and $^{45}N_2O^\beta$ production varied in concert at almost every station and depth, with a few exceptions (Fig. 5).*

**We changed all of the $N_2O$ production rates from nM/day to pM/day, here and throughout the text.**

*For example, in the secondary $NO_2^-$ maximum (182 m) at station PS3, in the $^{15}N–NO_2^-$ experiment, the production of $^{45}N_2O^\alpha$ was 60±30 pM $N_2O$/day (p = 0.09) and there was no significant production of $^{45}N_2O^\beta$ (Fig. 5h). In the parallel $^{15}N–NH_4^+$ experiment, the production of $^{45}N_2O^\beta$ was 0.7±0.3 pM $N_2O$/day (p = 0.06) and there was no significant production of $^{45}N_2O^\alpha$. At this station and depth, f (the proportion of $N^\alpha$ derived from $NO_2^-$) was equal to 0.9±0.2 (Table S4). The second experiment in which labeling was unequal occurred at the oxic–anoxic interface (92 m) at station PS2, where in the $^{15}N–NH_4^+$ experiment, the production of $^{45}N_2O^\alpha$ was 5±2 pM $N_2O$/day (p = 0.02) and there was no significant production of $^{45}N_2O^\beta$ (Fig. 5f). Here, f was equal to 0.2±0.1. Finally, at the mid–oxycline depth (25 m) at station PS3, in the $^{15}N– NH_4^+$ experiment, the production of $^{45}N_2O^\alpha$ was 0.23±0.8 pM $N_2O$/day (p = 0.02) and there was no significant production of $^{45}N_2O^\beta$. Here, f was statistically indistinguishable from 0.*

**We homogenized the x-axis ranges for Fig. S7 and Fig. S8 as much as possible while still allowing the variation in each panel to be visualized and moved Fig. S7 and Fig. S8 to the main text. They are now Figs. 4 and 5.**

[Figure]

*Figure 4. Net $^{46}N_2O$ production from $^{15}N–NO_3^-$ (a, d, g, indigo), $^{15}N–NO_2^-$ (b, e, h, blue), and $^{15}N–NH_4^+$ (c, f, i, yellow) at stations PS1 (a–c), PS2 (d–f), and PS3 (g–i). $N_2O$ production rates are plotted over depth profiles of dissolved $[O_2]$ (dashed lines) and $[N_2O]$ (solid lines, from Kelly et al., 2021). Error bars are calculated from linear regression slope error of $^{46}N_2O$ vs. incubation time. Note the different x–axis scales for $^{46}N_2O$ production (top) and $[O_2]$ and $[N_2O]$ (bottom).*

[Figure]

*Figure 5. Net $^{45}N_2O^\alpha$ (open symbols) and $^{45}N_2O^\beta$ (closed symbols) production from $^{15}N–NO_3^-$ (a, d, g, indigo), $^{15}N–NO_2^-$ (b, e, h, blue), and $^{15}N–NH_4^+$ (c, f, i, yellow) at stations PS1 (a–c), PS2 (d–f), and PS3 (g–i). $N_2O$ production rates are plotted over depth profiles of dissolved $[O_2]$ (dashed lines) and $[N_2O]$ (solid lines, from Kelly et al., 2021). Error bars are calculated from linear regression slope error of $^{45}N_2O$ vs. incubation time. Note the different x–axis scales for $^{45}N_2O$ production (top) and $[O_2]$ and $[N_2O]$ (bottom).*

The line at 215 belongs with the previous paragraph. And it's not clear whether this will be an example of rates varying in concert or not. Wordsmith accordingly.

**Did you mean a different line? 215 is just after eqn. (6), "where $^{15}N^{15}N^{16}O_{excess}$ represents the amount of $^{15}N^{15}N^{16}O$ produced in the sample over the course of the experiment."**

Equation 13: In the case of nitrite where a higher concentration was added then intended, I would think that the flux derived therefrom, J, is no longer proportional to nitrite (zero order) at these concentrations. Does this matter?

**If we compare the $^{15}N$-labeled ammonium treatment to the $^{15}N$-labeled nitrite treatment at the same experimental depth, the $^{45}N_2O$ and $^{46}N_2O$ production rates in the $^{15}N$-labeled nitrite treatment were far higher than those in the $^{15}N$-labeled ammonium treatment, even when normalized by atom fraction. This is visualized below. In fact, the rates of production of $^{45}N_2O$ and $^{46}N_2O$ in the $^{15}N$-labeled ammonium treatments were so small, comparatively, that they are visually indistinguishable from zero when plotted on the same scale as the rates of production of $^{45}N_2O$ and $^{46}N_2O$ in the $^{15}N$-labeled nitrite treatments.**

[Figure]

**Production of $^{45}N_2O$, divided by atom fraction, in the $^{15}N$-NO₂⁻ treatment vs. $^{15}N$-NH₄⁺ treatment at the same experimental depths. Red diamonds indicate p$^{45}N_2O^{\alpha}$/$^{15}F$ and black diamonds indicate p$^{45}N_2O^{\beta}$/$^{15}F$. b) Production of $^{46}N_2O$, divided by atom fraction squared, in the $^{15}N$-NO₂⁻ treatment vs. $^{15}N$-NH₄⁺ treatment at the same experimental depths. In both plots, the dashed line is the 1:1 line.**

**Since the tracer concentration was much higher in the $^{15}N$-labeled nitrite treatment (5.00 µM) than in the $^{15}N$-labeled ammonium treatment (0.501 µM), this imbalance of $^{45}N_2O$ production supports the idea that there is some dependence of $N_2O$ production rate on substrate concentration.**

Line 337: Wording of sentence is awkward.
**Revised to:**
*The model solves for $N_2O$ production rates, given a set of $NH_3$ oxidation, $NO_2^-$ oxidation, and $NO_3^-$ reduction rates calculated in Sect. 2.5, eqn. (7) (Table S2).*

Line 395: How can nitrite oxidation rates possibly be negative?

**The "negative" nitrite oxidation rates at two depths are likely an artifact of the elevated t₀ δ($^{15}$N) values in some of our $^{15}$N-NO₂⁻ treatments (discussed above). We have added this to the text.**

*In some cases, NO₂⁻ oxidation rates appeared negative due to a decrease in $^{15}$N–NO₃⁻ vs. incubation time (Fig. 3b, h), which was likely an artifact of the elevated t₀ δ($^{15}$N) values in some of our $^{15}$N–NO₂⁻ treatments (discussed above).*

Line 420: Remind the reader what "f" designates.
**Done.**
*At this station and depth, f (the proportion of $N^\alpha$ derived from NO₂⁻) was equal to 0.9±0.2 (Table S4).*

Equation 19: "AP" was designated as "15F" in equations above…
**Changed to $^{15}F$.**

$$p_{excess}^{45} = p^{45} - p_{expected}^{45} = p^{45} - \frac{p^{46}}{^{15}F}2\left(1 - {}^{15}F\right)$$

(19)

Could p45excess result from misestimation of the actual atom percent of substrates the incubations? The rates are very small such having a small error on AP could potentially account for this? Or wrong proportion of carrier? I think Figure S9 may allude to this but the associated uncertainty needs to be better explained in the main text, whether or not the data evince unequal values of "f" beyond a reasonable "doubt"
**Figure S9 (now Fig. S S7) alludes to this. The dashed lines in Figure S9 indicate the range of atom fractions in each type of experiment, which far exceeds the uncertainty in the atom fraction of any one individual experiment. So points above the dashed line indicate excess $^{45}$N₂O production, beyond a reasonable doubt.**

[Figure]

*Figure S7. Net production of $^{45}$N₂O$^\alpha$ (red diamonds) and $^{45}$N₂O$^\beta$ (black triangles) vs. $^{46}$N₂O from $^{15}$N-NH₄⁺ (a) and $^{15}$N-NO₂⁻ (b). The insert in (b) shows a zoomed-in view of the data. The solid black lines indicate the expected production $^{45}$N₂O$^\alpha$ and $^{45}$N₂O$^\beta$ from a process drawing both N atoms in N₂O from the same substrate pool, based on the atom fraction of the labeled substrate (NH₄⁺ or NO₂⁻) and a binomial distribution of N₂O isotopocules. Dashed lines indicate the range of expected values, based on the range of atom fractions in each experiment. Production of $^{45}$N₂O$^\alpha$ and $^{45}$N₂O$^\beta$ above this expected production indicate the presence of a hybrid process.*

Figure 4: Present in order brought up in text, which is N2O production from nitrate first. Is production from NH4+ only necessarily hydroxylamine oxidation? It is called that in some figure captions. If so, it would be much easier for readers if it were called hydroxylamine oxidation throughout.

**The order of this figure (now Figure 6) has been changed. Sorry, "hydroxylamine oxidation" was a mistake — N2O from NH4+ could also include hybrid production using an internal NO2- pool. We have revised the figure captions to "N2O production from solely NH4+".**

[Figure]

*Figure 6. $N_2O$ production from $NO_3^-$ (a, e, i, indigo diamonds), $N_2O$ production from $NO_2^-$ (b, f, j, blue diamonds), hybrid $N_2O$ production (c, g, k, green diamonds), and $N_2O$ production from solely $NH_4^+$ (d, h, l, yellow diamonds) at stations PS1 (a–d), PS2 (e–h), and PS3 (i–l). Panels a, e, and i also show rates of $NO_3^-$ reduction to $NO_2^-$ (open circles). Panels b, f, and j show depth profiles of dissolved $[O_2]$ (dashed lines) and $[N_2O]$ (solid lines, from Kelly et al., 2021). Panels c, g, and k show rates of $NH_3$ oxidation (gray circles). $N_2O$ production rate error bars are calculated from 100 model optimizations, varying key parameters by up to 25%. Note the different x–axis scales for $NO_3^-$ reduction to $NO_2^-$ (a, e, i, bottom), $N_2O$ production (top), $[O_2]$ and $[N_2O]$ (b, f, j, bottom), and $NH_3$ oxidation (c, g, k, bottom).*

Section 3.5: I would start with describing N2O production "as a whole", followed by nitrate reduction (highest flux), etc… Same order of operation as suggested above.
**We changed the order of section 3.5 to discuss N₂O production from nitrate first. We also changed the corresponding section of the discussion (Section 4.4).**

**3.5 Oxygen dependence of N₂O production**
*The oxygen dependencies of N₂O production pathways were determined by fitting model derived N₂O production pathways vs. [O₂] using the following rate law:*

$$rate = \ ae^{-b[O_2]} \qquad\qquad (21)$$

*In this analysis, both ambient [O₂] measured by the Sea–Bird sensor mounted on the rosette ("ambient [O₂]") and [O₂] measured by chemiluminescent optodes mounted inside incubation bottles ("incubation [O₂]") were examined. The rate dependencies on ambient and incubation [O₂] reflect both preconditioning (i.e., the ambient [O₂] in which the microbial community was living before the incubation experiment), and response to perturbation (i.e., the experimental conditions inside the incubation bottles, if different from the environment). Those incubations that had higher incubation [O₂] than the ambient [O₂], had received small oxygen perturbations.*

*N₂O production via denitrification exhibited an exponentially declining relationship with dissolved O₂, where N₂O production from NO₂⁻ was more inhibited by dissolved O₂ than N₂O production from NO₃⁻ (Fig. 8). When looking at the oxygen dependence of denitrification, we found several instances of N₂O production from NO₃⁻ via denitrification with dissolved [O₂] greater than 3 μM (Fig. 8a–b). For example, at the oxic–anoxic interface at station PS2, where ambient [O₂] was 6.49 μM and incubation [O₂] was 6.29±0.07 μM (Table S1), N₂O production from NO₃⁻ was 70±10 pM N₂O/day (Fig. 6e, Table S4). N₂O production from NO₂⁻ at the same station and depth was 8.9±0.2 pM N₂O/day (Fig. 6f, Table S4). Similarly, at the oxic–anoxic interface of station PS3, where ambient [O₂] was 12.48 μM and incubation [O₂] was 6.64±0.03 μM (Table S1), N₂O production from NO₃⁻ was 120±20 pM N₂O/day (Fig. 6i, Table S4). There were also two anoxic depths at station PS2 that were not sparged with He before tracer addition ("base of ODZ" and "deep ODZ core"), where ambient [O₂] was below detection but incubation [O₂] was significantly elevated (17.7±0.1 μM and 19.2±0.8 μM, respectively; Table S1). At these depths, N₂O production from NO₂⁻ was 12±1 pM N₂O/day and 5.2±0.4 pM N₂O/day, respectively (Fig. 6f, Table S4). N₂O production from NO₃⁻ at the "deep ODZ core" depth was 210±40 pM N₂O/day (Table S4).*

[Figure]

*Figure 8. N₂O production from NO₃⁻ via denitrification (a, b) and from NO₂⁻ via denitrification (c, d), measured at a range of [O₂] measured by a Seabird sensor (a, c) or by chemiluminescent optodes mounted inside incubation bottles (b, d). Curves of form* **yield = ae⁻ᴼ²ᵇ** *are fit through the data (black lines); values of a and b are shown in white boxes in each plot.*

*Hybrid N₂O production rates also decreased exponentially with increasing dissolved [O₂] (Fig. 9a–b). Fitting hybrid rates vs. ambient [O₂] produced a rate equation (21) with a = 65.83 and b = 0.17 (Fig. 9a); hybrid rates vs. incubation [O₂] produced fits with a = 76.26 and b = 0.067 (Fig. 9b).*

[Figure]

*Figure 9. Hybrid N₂O production rates (a,b), N₂O yield (%) during hybrid production (c, d), and N₂O yield (%) during production from solely NH₄⁺ (e, f) along a range of ambient [O₂] measured by a Seabird sensor for the Niskin bottles from which samples were taken (a, c, e) and [O₂] measured by chemiluminescent optodes mounted inside incubation bottles (b, d, f). Error bars are calculated from 100 model optimizations, varying key parameters by up to 25%. Yields are only calculated at stations and depths where rates of NH₃ oxidation are greater than 0. Curves of form $\textbf{rate} = \textbf{ae}^{-\textbf{b[O}_\textbf{2}]}$ are fit through the data (black lines); values of a and b are shown in white boxes in each plot.*

*The rate of N₂O production from solely NH₄⁺ also decreased exponentially with increasing dissolved [O₂]. The highest rates of N₂O production from solely NH₄⁺ occurred in the secondary chlorophyll maximum at station PS3 (Table S4), where dissolved oxygen was below detection.*

*$N_2O$ yield during production from solely $NH_4^+$ also exhibited exponentially decreasing relationships with dissolved $[O_2]$ (Fig. 9e–f). To ensure mass balance in terms of $NH_4^+$ consumption (Fig. S9), $N_2O$ yield (%) during production from solely $NH_4^+$ was calculated as:*

$$yield\ (\%) = \frac{2\left[N_2O\ from\ solely\ NH_4^+\left(^{nM\ N_2O}/_{day}\right)\right]}{2\left[N_2O\ from\ solely\ NH_4^+\left(^{nM\ N_2O}/_{day}\right)\right] + hybrid\ N_2O\left(^{nM\ N_2O}/_{day}\right) + NH_3\ oxidation\left(^{nM\ N}/_{day}\right)} \quad (22)$$

*where $N_2O$ production from solely $NH_4^+$ is in units of nM $N_2O$/day, hybrid $N_2O$ production is in units of nM $N_2O$/day, and $NH_3$ oxidation to $NO_2^-$ is in units of nM N/day. This assumes that the formation of $N_2O$ from solely $NH_4^+$ draws two nitrogen atoms from the $NH_4^+$ pool, while hybrid $N_2O$ production and the oxidation of $NH_4^+$ to $NO_2^-$ each draw one atom from the $NH_4^+$ pool (Fig. S9). Following the same convention, $N_2O$ yield (%) during hybrid production was calculated as:*

$$yield\ (\%) = \frac{hybrid\ N_2O\left(^{nM\ N_2O}/_{day}\right)}{2\left[N_2O\ from\ solely\ NH_4^+\left(^{nM\ N_2O}/_{day}\right)\right] + hybrid\ N_2O\left(^{nM\ N_2O}/_{day}\right) + NH_3\ oxidation\left(^{nM\ N}/_{day}\right)} \quad (23)$$

*The maximum $N_2O$ yield from hybrid production was 21±7% (Fig. 9c, d). while the maximum $N_2O$ yield during production from solely $NH_4^+$ was 2.2±0.7% (Fig. 9e, f). $N_2O$ yield during production from solely $NH_4^+$ declined more sharply with increased $O_2$ than $N_2O$ yield during hybrid production (Fig. 9c–f).*

Figure 4 d: the trace for ammonium oxidation differs from the corresponding trace in Figure 3 a. **Thank you for catching this. Figure 3a is correct. Not sure what happened with Figure 4d (now Figure 6d) but we corrected it (see response to comment on Figure 4 above).**

Discussion:
Because the study is very complex, it would be beneficial for the discussion to begin with a paragraph that summarizes the dominant findings, rather than jumping into the deep end form the get go. In this regard, I would also get N2O production from denitrification out of the way first because it was the dominant flux, then discuss hybrid production. I find it interesting as well that production from hydroxylamine was virtually absent except at the surface – I think this merits more emphasis.
**Thank you for this suggestion. We added a summary paragraph at the beginning of the discussion, and we changed the order of the discussion to 1) N₂O production from denitrification, 2) hybrid production, 3) production from solely NH₄⁺.**

**4 Discussion**
*In this study, we found that $N_2O$ production from denitrification was the dominant source of $N_2O$ both within the ODZ and in the upper oxycline. Hybrid $N_2O$ production was a smaller but significant contributor to $N_2O$ in the upper oxycline, and the primary source of $N_2O$ in the deep oxycline. $N_2O$ production from solely $NH_4^+$ – which includes $N_2O$ from hydroxylamine oxidation, hybrid production with cellular $NO_2^-$, and nitrifier–denitrification with cellular $NO_2^-$ – was negligible everywhere except surface waters. Our findings of equal formation of $^{45}N_2O^\alpha$ and $^{45}N_2O^\beta$ in most experiments indicate that $N^\alpha$ retains an equal proportion of $NO_2^-$ and $NH_4^+$– derived N during hybrid production, which may imply that hybrid $N_2O$ production exhibits a constant $\delta(^{15}N^{sp})$. All of the processes measured in this study exhibited a strong dependence on*

*dissolved oxygen, although denitrification was less inhibited by dissolved oxygen than previous work would suggest.*

Section 4.3: I get that MOST N2O is produced by denitrification and 1/5 from hybrid production. Is that what is also inferred from natural abundance measurements, in these proportions? Curious minds want to know

**Yes, this is indeed what we inferred from natural abundance measurements. Based on natural abundance site preference, we found that the near-surface [N₂O] maximum in was likely to be comprised of ~20% N₂O produced via nitrification or archaeal N₂O production and ~80% N₂O produced via denitrification (Kelly et al., 2021). We added this to the beginning of section 4.1 (formerly section 4.3).**

*Based on our rate data, N₂O production from NO₃⁻ is the dominant source of N₂O in both the near–surface [N₂O] maximum and the anoxic ODZ core. This agrees well with natural abundance isotopocule measurements in the ETNP, which indicate that the near surface [N₂O] maximum is likely to be comprised of ~80% N₂O produced via denitrification and ~20% N₂O produced via nitrification or archaeal N₂O production, producing a local minimum in $\delta(^{15}N^{sp})$ (Kelly et al., 2021).*

Line 642: What do you mean by "allowed?" Need better wording.

**Here we're alluding to natural abundance measurements indicating that N₂O production from NO₃⁻ could be an important source of N₂O in the anoxic core of ODZs, as long as it has a positive $\delta(^{15}N^{sp})$. As you know, denitrification is usually assigned $\delta(^{15}N^{sp}) \approx 0‰$ (Sutka et al., 2006), but some strains of denitrifying bacteria can produce N₂O with $\delta(^{15}N^{sp}) > 0‰$ (Toyoda et al., 2005; Wang et al., 2023). And so can denitrifying fungi (Sutka et al., 2008; Rohe et al., 2014; Yang et al., 2014; Lazo-Murphy et al., 2022). So, given that there are several potential sources of N₂O production from NO₃⁻ with a positive $\delta(^{15}N^{sp})$, the importance of N₂O production from NO₃⁻ in this study agrees with natural abundance work.**

*Natural abundance isotopomer work has shown that N₂O production from NO₃⁻ could be an important source of N₂O in the anoxic core of ODZs, as long as it has a positive $\delta(^{15}N^{sp})$ (Casciotti et al., 2018; Kelly et al., 2021; Monreal et al., 2022). While denitrification is generally accepted to produce N₂O with $\delta(^{15}N^{sp}) \approx 0‰$ (Sutka et al., 2006; other refs), some strains of denitrifying bacteria can produce N₂O with $\delta(^{15}N^{sp}) = 10–22‰$ (Toyoda et al., 2005; Wang et al., 2023) and denitrifying fungi produce N₂O with $\delta(^{15}N^{sp}) = 35–37‰$ (Sutka et al., 2008; Rohe et al., 2014; Yang et al., 2014; Lazo-Murphy et al., 2022). Here, the dominance of N₂O production from $^{15}N–NO_3^-$, combined with parallel natural abundance isotopomer studies, suggest that strains of denitrifying bacteria and fungi that produce N₂O with a high site preference may be important contributors to N₂O in the core of ODZs.*

Line 650: qualify "this" , you mean the notion that internal pool are processed, not external…?

**Yes, exactly. We changed "this" to "N₂O production from NO₃⁻ that utilizes an internal NO₂⁻ pool".**

*N₂O production from NO₃⁻ that utilizes an internal NO₂⁻ pool is currently left out of most biogeochemical models of nitrogen cycling in and around oxygen–deficient zones (Bianchi et al., 2023), and modeling work is needed that includes this as a source of N₂O.*

Line 600: Reader is left hanging: What are the implications for mechanisms of production? Need a concluding sentence for the paragraph to bridge it to the next, or simply amalgamate with the following paragraph.

**We re-wrote this paragraph and the following text to reflect the fact that most of our experiments actually have *equal* formation of $^{45}N_2O^\alpha$ and $^{45}N_2O^\beta$, and thus *f*=0.5, which would imply that hybrid $\delta(^{15}N^{sp})$ would *not* vary in most of the tested conditions.**

*Although our data do not allow us to comment directly on the enzymatic machinery of hybrid $N_2O$ formation, our data can be used to theorize hypothetical pathways for hybrid $N_2O$ production. Firstly, we see much higher rates of hybrid production using ambient $NO_2^-$ (Pathway 3 in Wan et al., 2023) than hybrid production using cellular $NO_2^-$ (Pathway 2 in Wan et al., 2023). Again, this agrees with the results of Wan et al. (2023), who see higher rates of hybrid formation from extracellular $NO_2^-$ within the range of $[^{15}N–NH_4^+]/[NO_2^-]$ covered by our experiments. In our model, hybrid $N_2O$ production is operationally defined as a 1:1 combination of N derived from $NH_4^+$ and $NO_2^-$, which is generally consistent with previous work (Stieglmeier et al., 2014). Any combination of N derived from $NO_2^-$ with a second N derived from $NO_2^-$ would be included in the modeled quantity of $N_2O$ production from $NO_2^-$; likewise, any combination of N derived from $NH_4^+$ with a second N derived from $NH_4^+$ would be included in the $N_2O$ production from solely $NH_4^+$. The question, then, is what reaction would be specific enough to have one N derived from each substrate, but not specific enough to govern $^{15}N$ placement in the resulting $N_2O$? One such reaction could be the combination of $NH_4^+$ and $NO_2^-$ to form a symmetrical intermediate such as hyponitrite (HONNOH or $^-ONNO^-$ in its deprotonated form), which reacts to form $N_2O$ via breakage of one of the N–O bonds, resulting in $N_2O$ that contains a 1:1 ratio of $NH_4^+:NO_2^-$. With a precursor such as hyponitrite, equal formation of $^{45}N_2O^\alpha$ and $^{45}N_2O^\beta$ could be achieved with non–selective N–O bond breakage.*

*These findings of equal $^{45}N_2O$ production have important implications for the natural abundance $\delta(^{15}N^{sp})$ of $N_2O$ produced by the hybrid $N_2O$ process. Assuming that hybrid $N_2O$ production proceeds through a symmetrical intermediate in which $NH_4^+$ and $NO_2^-$ are paired in a 1:1 ratio, we can model $\delta(^{15}N^{sp})$ as:*

$$\delta\left(^{15}N^{sp}\right) = \delta\left(^{15}N^\alpha\right) - \delta\left(^{15}N^\beta\right)$$

$$= \left[f\delta\left(^{15}N - NO_2^-\right) + (1-f)\delta\left(^{15}N - NH_4^+\right)\right] - \left[(1-f)\delta\left(^{15}N - NO_2^-\right) + f\delta\left(^{15}N - NH_4^+\right) - \varepsilon\right] \quad (24)$$

*where f is the proportion of the $\alpha$ nitrogen derived from $NO_2^-$ and the proportion of the $\beta$ nitrogen derived from $NH_4^+$, and $\varepsilon$ is the fractionation factor associated with $N^\beta$–O bond breakage. If $f \neq \frac{1}{2}$, hybrid $\delta(^{15}N^{sp})$ retains a dependence on the $\delta(^{15}N)$ of the substrates – or more accurately, the difference in $\delta(^{15}N)$ of the two substrates; if the $\delta(^{15}N)$ of the substrates is equal, it will cancel out regardless of f. If $\delta(^{15}N–NH_4^+) > \delta(^{15}N–NO_2^-)$, as is generally the case in the secondary nitrite maximum (Buchwald et al., 2015; Casciotti, 2016), then low values of f should produce high hybrid $\delta(^{15}N^{sp})$, and high values of f should produce low hybrid $\delta(^{15}N^{sp})$ (Fig. 10). If, however, $f = \frac{1}{2}$, as was the case for most experimental depths in this study, hybrid $\delta(^{15}N^{sp})$ should depend only on $\varepsilon$ and not the isotopic composition of each substrate. This means that a*

*δ($^{15}N^{sp}$) endmember could potentially be established for hybrid N₂O production, even though hybrid N₂O production draws from different substrate pools. More studies are needed to determine the δ($^{15}N^{sp}$) of N₂O produced by ammonia–oxidizing archaea under a range of conditions.*

Paragraph at line 605: Reads like something that should be in results section.
**We moved this text down to our paragraph where we address the unequal production of $^{45}N_2O^\alpha$ and $^{45}N_2O^\beta$ at certain depths, which anchored significant relationships between *f* and ambient [O₂] and potential density anomaly ($\sigma_\theta$). The oxygen and potential density gradients may be proxies for changing archaeal community compositions at different depths in the water column, which may exhibit different patterns of incorporation of NO₂⁻-derived N and NH₄⁺-derived N into N$^\alpha$ and N$^\beta$. It is also possible that we sampled a different "hybrid" N₂O-producing process at these depths, such as fungal co-denitrification (Shoun et al., 2012), which may proceed via a different pathway from archaeal hybrid N₂O production.**

*The unequal production of $^{45}N_2O^\alpha$ and $^{45}N_2O^\beta$ observed at certain depths led to values of f significantly different from 0.5 (Table S4). At these depths, N$^\alpha$ retained a different proportion of nitrogen derived from NO₂⁻ and NH₄⁺ than N$^\beta$, causing $^{45}N_2O^\alpha$ and $^{45}N_2O^\beta$ to diverge. The depths with f ≠ 0.5 anchored significant relationships between f and ambient [O₂] ($R^2 = 0.84$, p < 0.001; Fig. S10a) and potential density anomaly ($\sigma_\theta$) ($R^2 = 0.72$, p < 0.001; Fig. S10b). The oxygen and potential density gradients may be proxies for changing archaeal community compositions at different depths in the water column, which may exhibit different patterns of incorporation of NO₂⁻–derived N and NH₄⁺–derived N into N$^\alpha$ and N$^\beta$. It is also possible that we sampled a different "hybrid" N₂O–producing process at these depths, such as fungal co–denitrification (Shoun et al., 2012), which may proceed via a different pathway from archaeal hybrid N₂O production.*

Line 610: Articulate fully for readers to catch up again "findings of unequal alpha vs. beta production during hybrid pathway have implications for interpretation of the natural abundance isotopes of N2O produced by hybrid process."
**We now write:**
*The equal formation of $^{45}N_2O^\alpha$ and $^{45}N_2O^\beta$ led to values of f within error of 0.5 in most of our experiments (Table S4), and the mean value of f across all stations and depths was 0.5±0.2. This means that during hybrid N₂O production, half of the N$^\alpha$ atoms were derived from NO₂⁻, and half were derived from NH₄⁺ (likewise for N$^\beta$).*

Paragraph at line 670: I don't understand why the results here should be different than cited study.
**(Ji et al., 2018) did not include hybrid N₂O production in their estimates of N₂O yield. We added this to the text.**
*The maximum N₂O yield for hybrid production (21%; Fig. 8c,d) was an order of magnitude higher than previous estimates of N₂O yields during NH₃ oxidation from ETSP and ETNP, which did not include hybrid N₂O production (Ji et al., 2018).*

I remain perplexed by the following: In Figure S8, there is NO production of $^{45}N_2O$ from addition of $^{15}NH_4^+$ at 100 m at station 1, yet there is reportedly 50 nM/day N2O production from

the hybrid pathway at this depth… Am I fundamentally misunderstanding something about the experimental design? The hybrid pathway requires some input from $^{15}NH_4^+$ which should be detected as $^{45}N_2O$?

**We can understand why this would be confusing. The model solves for the same rates of hybrid $N_2O$ production in the $^{15}NH_4^+$ and $^{15}NO_2^-$ experiments. In this case, there is high $^{45}N_2O$ production in the $^{15}NO_2^-$ experiment but very little $^{45}N_2O$ production in the $^{15}NH_4^+$, so the model finds an intermediate value. Given that the $^{15}N$-$NO_2^-$ spike was added at a higher concentration (5 µM) than the $^{15}N$-$NH_4^+$ spike (0.5 µM), it is feasible that the $^{15}N$-$NO_2^-$ generated a greater $^{45}N_2O$ signal than the $^{15}N$-$NH_4^+$ experiment.**

[revised manuscript text omitted]

---

## Author Response (AR2)

**Response to Reviewers 6/4/2024**
**Submission: "Isotopomer labeling and oxygen dependence of hybrid nitrous oxide production" [Manuscript ID egusphere-2023-2642]**

We thank the editor and two reviewers for their positive re-evaluation of our work. We address below the editor's recommendation and a few minor technical corrections suggested by anonymous reviewer #1.

**Editor**

Regarding your conclusions ("we posit a two–step process for hybrid N2O production involving an initial bond–forming step that draws nitrogen atoms from each substrate to form a symmetric intermediate, and a second bond–breaking step that breaks an N–O bond in the symmetric intermediate to form N2O"), I would like to refer your attention to the paper Wei et al. 2019, Geochimica et Cosmochimica Acta 267, 17–32, https://doi.org/10.1016/j.gca.2019.09.018, where in section 4.2 we discussed possible pathways ("end members") of hybrid N2O formation, i.e. via cis-hyponitrous acid, trans-hyponitrous acid and nitramide, all leading to N2O with different SP values. We have cited in this section the relevant papers in which the respective SP values were either measured or predicted. You may want to include these hypothetical pathways in your discussion or conclusions, but do not feel obliged to do so.

**Thank you for pointing out this excellent work on potential $\delta(^{15}N^{sp})$ values of hybrid N2O formation. We have added two references to this work to our discussion of hybrid $\delta(^{15}N^{sp})$:**

*The question, then, is what reaction would be specific enough to have one N derived from each substrate, but not specific enough to govern $^{15}N$ placement in the resulting N2O? One such reaction could be the combination of $NH_4^+$ and $NO_2^-$ to form a symmetrical intermediate such as hyponitrous acid (HONNOH, or hyponitrite $^-ONNO^-$ in its deprotonated form), which has been discussed as a possible intermediate in hybrid nitrous oxide formation (Wei et al., 2019). Hyponitrous acid may react to form N2O via breakage of one of the N–O bonds, resulting in N2O that contains a 1:1 ratio of $NH_4^+$:$NO_2^-$. With a precursor such as hyponitrite or hyponitrous acid, equal formation of $^{45}N_2O^\alpha$ and $^{45}N_2O^\beta$ could be achieved with non–selective N–O bond breakage.*
*[…]*
*This means that a $\delta(^{15}N^{sp})$ endmember could potentially be established for hybrid N2O production, even though hybrid N2O production draws from different substrate pools. Wei et al. (2019) discuss possible pathways or end members of hybrid N2O formation, i.e. via cis-hyponitrous acid, trans-hyponitrous acid and nitramide, all leading to N2O with different $\delta(^{15}N^{sp})$ values. More studies are needed to determine the $\delta(^{15}N^{sp})$ of N2O produced by ammonia–oxidizing archaea under a range of conditions.*

**Reviewer 1**

L198. "… was added to each sample a final concentration of …" I think "a" between "sample" and "final" should be "at".
**Corrected.**

L682. "Fig. 8g-h" must be "Fig. 8a-d" or "Fig. 8a-b".
**Corrected.**

L684. "Fig. 8h" should be "Fig. 8b".
**Correct.ed**